# Local Curvature Descent: Squeezing More Curvature out of Standard and Polyak Gradient Descent

## Abstract

We contribute to the growing body of knowledge on more powerful and adaptive stepsizes for convex optimization, empowered by *local curvature information*. We do not go the route of fully-fledged second-order methods which require the expensive computation of the Hessian. Instead, our key observation is that, for some problems (e.g., when minimizing the sum of squares of absolutely convex functions), certain local curvature information is readily available, and can be used to obtain surprisingly powerful matrix-valued stepsizes, and meaningful theory. In particular, we develop three new methods—LCD1, LCD2 and LCD3—where the abbreviation stands for *local curvature descent*. While LCD1 generalizes gradient descent with fixed stepsize, LCD2 generalizes gradient descent with Polyak stepsize. Our methods enhance these classical gradient descent baselines with local curvature information, and our theory recovers the known rates in the special case when no curvature information is used. Our last method, LCD3, is a variable-metric version of LCD2; this feature leads to a closed-form expression for the iterates. Our empirical results are encouraging, and show that the local curvature descent improves upon gradient descent.

## 1 Introduction

In this work we revisit the standard optimization problem

$$\min_{x \in \mathbb{R}^d} f(x), \tag{1}$$

where $f : \mathbb{R}^d \to \mathbb{R}$ is a continuous convex function with a nonempty set of minimizers $\mathcal{X}_\star$. Further, we denote the optimal function value by $f_\star := f(x_\star)$, where $x_\star \in \mathcal{X}_\star$.

### 1.1 First-order methods

First-order methods of the Gradient Descent (GD) and Stochastic Gradient Descent (SGD) variety have been widely adopted to solve problems of type (1) (Polyak, 1963; Robbins & Monro, 1951). Due to their simplicity and relatively low computational cost, these methods have seen great success across many machine learning applications, and beyond. Nonetheless, GD, performing iterations of the form

$$x_{k+1} = x_k - \gamma_k \nabla f(x_k), \tag{2}$$

where $\gamma_k > 0$ is a learning rate (stepsize), suffers from several well-known drawbacks. For example, for convex and $L$-smooth objectives, GD converges provided that[1]

$$\sum_{k=0}^{\infty} \gamma_k = +\infty \qquad \text{and} \qquad \gamma_k \leq \tfrac{1}{L} \quad \forall k \geq 0 \tag{3}$$

(Nesterov, 2004). For many problems, $L$ is very large and/or unknown, and estimating its value is a non-trivial task. Overestimation of the smoothness constant leads to unnecessarily small stepsizes, which degrades performance, both in theory and in practice.

---

[1]It is theoretically possible to use slightly larger stepsizes, by at most a factor of 2, but this is does not play a role in our narrative.

**Polyak stepsize.** When the optimal value $f_\star$ is known, a very elegant solution to the above-mentioned problems was provided by Polyak (1987), who proposed the use of what is now known as the Polyak stepsize:

$$\gamma_k := \frac{f(x_k) - f_\star}{\|\nabla f(x_k)\|^2}. \tag{4}$$

It is known that if $f$ is convex and $L$-smooth, then $\gamma_k \geq \frac{1}{2L}$ for all $k \geq 0$. So, unlike strategies based on the recommendation provided by (3), Polyak stepsize can never be too small compared to the upper bound from (3). In fact, it is possible for $\gamma_k$ to be larger than $\frac{1}{L}$, which leads to practical benefits. Moreover, this is achieved without having to know or estimate $L$, which is a big advantage. Since the function value $f(x_k)$ and the gradient $\nabla f(x_k)$ are typically known, the only price for these benefits is the knowledge of the optimal value $f_\star$. This may or may not be a large price to pay, depending on the application.

**Malitsky-Mishchenko stepsize.** In the case of convex and locally smooth objectives, Malitsky & Mishchenko (2020) recently proposed an ingenious adaptive stepsize rule that iteratively builds an estimate of the inverse *local* smoothness constant from the information provided by the sequence of iterates and gradients. Furthermore, they prove their methods achieve the same or better rate of convergence as GD, without the need to assume global smoothness. For a review of further approaches to adaptivity, we refer the reader to Malitsky & Mishchenko (2020), and for several extensions of this line of work, we refer to Zhou et al. (2024).

**Adaptive stepsizes in deep learning.** When training neural networks and other machine learning models, issues related to the appropriate selection of stepsizes are amplified even further. Optimization problems appearing in deep learning are not convex and may not even be $L$-smooth, or $L$ is prohibitively large, and tuning the learning rate usually requires the use of schedulers or a costly grid search. In this domain, adaptive stepsizes have played a pivotal role in the success of first-order optimization algorithms. Adaptive methods such as Adam, RMSProp, AMSGrad, and Adagrad scale the stepsize at each iteration based on the gradients (Kingma & Ba, 2017; Hinton, 2014; Reddi et al., 2019; Duchi et al., 2011). Although Adam has seen great success empirically when training deep learning models, there is very little theoretical understanding of why it works so well. On the other hand, Adagrad converges at the desired rate for smooth and Lipschitz objectives but is not as successful in practice as Adam (Duchi et al., 2011).

## 1.2 SECOND-ORDER METHODS

When $f$ is twice differentiable and $L$-smooth, $L$ can be seen as a global upper bound on the largest eigenvalue of the Hessian of $f$. So, there are close connections between the way a learning rate should be set in GD-type methods and the curvature of $f$.

**Newton's method.** Perhaps the most well-known second-order algorithm is Newton's method:

$$x_{k+1} = x_k - \left(\nabla^2 f(x_k)\right)^{-1} \nabla f(x_k).$$

When it works, it converges in a few iterations only. However, it may fail to converge even on convex objectives[2]. It needs to be modified in order to converge from any starting point, say by adding a damping factor (Hanzely et al., 2022) or regularization (Mishchenko, 2023). However, under suitable assumptions, Newton's method converges quadratically when started close enough to the solution. The key difficulty in performing a Newton's step is the computation of the Hessian and a performing a linear solve. In analogy with (2), it is possible to think of $\left(\nabla^2 f(x_k)\right)^{-1}$ as a matrix-valued stepsize.

**Quasi-Newton methods.** To reduce the computational cost, quasi-Newton methods such as L-BFGS utilize an approximation of the inverse Hessian that can be computed from gradients and iterates only, typically using the approximation $\nabla^2 f(x_{k+1})(x_{k+1} - x_k) \approx \nabla f(x_{k+1}) - \nabla f(x_k)$, which makes sense under appropriate assumptions when $\|x_{k+1} - x_k\|$ is small Nocedal & Wright (2006); Al-Baali et al. (2014); Al-Baali & Khalfan (2007); Dennis & Moré (1977). Until very recently, quasi-Newton methods were merely efficient heuristics, with very weak theory beyond quadratics (Kovalev et al., 2021; Rodomanov & Nesterov, 2021).

**Polyak stepsize with second-order information.** Li et al. (2022) recently proposed extensions of the Polyak stepsize, named SP2 and SP2+, that incorporate second-order information. SP2 can also

---

[2]A well-known example is the function $f(x) = \ln(e^{-x} + e^x)$ with $x_0 = 1.1$.

be derived similarly to the Polyak stepsize. While SP2 can be utilized in the non-convex stochastic setting, it only has convergence theory for quadratic functions and can often be very unstable in practice. Furthermore, the quadratic constraint defined for SP2 may not even be a localization set. Instead, we propose an assumption similar to earlier works Karimireddy et al. (2018); Gower et al. (2019), with the aim of using second-order information rigorously.

### 1.3  NOTATION

All vectors are in $\mathbb{R}^d$ unless explicitly stated otherwise. We use $\mathcal{X}_\star$ to denote the set of minimizes of $f$. Matrices are uppercase and bold (e.g., $\mathbf{A}, \mathbf{C}$), the $d \times d$ zero (resp. identity) matrix is denoted by $\mathbf{0}$ (resp. $\mathbf{I}$), and  is the set of $d \times d$ positive semi-definite matrices. The standard Euclidean inner product is denoted with $\langle \cdot, \cdot \rangle$. For $\mathbf{A} \in$, we let $\|x\|_{\mathbf{A}}^2 := \langle \mathbf{A}x, x \rangle$. By $\|x\|_p := (\sum_{i=1}^d |x_i|^p)^{1/p}$ we denote the $L_p$ norm in $\mathbb{R}^d$. The Löwner order for positive semi-definite matrices is denoted with $\preceq$.

## 2  SUMMARY OF CONTRIBUTIONS

In this work we contribute to the growing body of knowledge on more powerful and adaptive step-sizes, empowered by *local curvature information*. We do not go the route of fully-fledged second-order methods which require the expensive computation of the Hessian.

> Instead, our key observation is that, for some problems, certain local curvature information is readily available, and can be used to obtain surprisingly powerful matrix-valued stepsizes.

The examples mentioned above, and discussed in detail in Sections 6 and 7 lead to the following abstract assumption, which at the same time defines what we mean by the term *local curvature*:

**Assumption 2.1** (Convexity and smoothness with local curvature). *There exists a curvature mapping/metric/matrix* $\mathbf{C} : \mathbb{R}^d \rightarrow$ *and a constant* $L_{\mathbf{C}} \geq 0$ *such that the inequalities*

$$\underbrace{f(y) + \langle \nabla f(y), x - y \rangle + \tfrac{1}{2}\|x - y\|_{\mathbf{C}(y)}^2}_{M_{\mathbf{C}}^{\mathrm{low}}(x;y)} \leq f(x), \tag{5}$$

$$f(x) \leq \underbrace{f(y) + \langle \nabla f(y), x - y \rangle + \tfrac{1}{2}\|x - y\|_{\mathbf{C}(y)+L_{\mathbf{C}}\cdot\mathbf{I}}^2}_{M_{\mathbf{C}}^{\mathrm{up}}(x;y)} \tag{6}$$

*hold for all* $x, y \in \mathbb{R}^d$.

Assumption 2.1 defines a new class of functions. Note that with the specific choice $\mathbf{C}(y) \equiv \mathbf{0}$, (5) reduces to convexity, and (6) reduces to $L$-smoothness, with $L = L_{\mathbf{C}}$. Note that any function satisfying (5) is necessarily convex, and similarly, any $L$-smooth function satisfies (6) with any curvature mapping $\mathbf{C}$ and $L_{\mathbf{C}} = L$. However, the converse is not true: a function satisfying (6) is not necessarily $L$-smooth for any finite $L$. Further, note that if $f$ is $\mu$-strongly convex, then it satisfies (5) with curvature mapping $\mathbf{C}(y) \equiv \mu\mathbf{I}$. The class of convex and $L$-smooth functions is one of the most studied functional classes in optimization. Our new class is a strict and, as we shall see, useful generalization.

We now provide a brief overview of our theoretical and empirical contributions:

### 2.1  LOCAL CURVATURE AND A NEW FUNCTION CLASS

We define a new function class, described by Assumption 2.1, extending the classical class of convex and $L$-smooth functions. Further, we show that there are problems which satisfy Assumption 2.1 with nontrivial and easy-to-compute curvature mapping $\mathbf{C}$ (see Section 6 and Section 7).

### 2.2  THREE NEW ALGORITHMS

We propose three novel algorithms for solving problem (1) for function $f$ satisfying Assumption 2.1: Local Curvature Descent 1 (LCD1), Local Curvature Descent 2 (LCD2) and Local Curvature Descent

3 (LCD3). First, LCD1 generalizes GD with constant stepsize: one moves from point $y$ to the point obtained by minimizing the upper bound (6) on $f$ in $x$. Indeed, if $\mathbf{C}(y) \equiv \mathbf{0}$, this algorithmic design strategy leads to gradient descent with stepsize $1/L$, where $L = L_{\mathbf{C}}$. Second, LCD2 generalizes GD with Polyak stepsize: one moves from point $y$ to the Euclidean projection of $y$ onto the ellipsoid

$$\mathcal{L}_{\mathbf{C}}(y) := \{x \in \mathbb{R}^d \mid f(y) + \langle \nabla f(y), x - y \rangle + \tfrac{1}{2}\|x - y\|^2_{\mathbf{C}(y)} \leq f_\star\}.$$

Indeed, if $\mathbf{C}(y) \equiv \mathbf{0}$, this algorithmic design leads to GD with stepsize (4). Computing the projection involves finding the unique root of a scalar equation in variable, which can be executed efficiently. Third, LCD3 is obtained from LCD2 by replacing the Euclidean projection with the projection defined by the local curvature matrix $\mathbf{C}$. The projection problem then has a closed-form solution.

## 2.3 THEORY

We prove convergence theorems for LCD1 (Theorem 4.1) and LCD2 (Theorem 4.1), with the same $\mathcal{O}(1/k)$ worst case rate of GD with constant and Polyak stepsize, respectively. Previous work on preconditioned Polyak stepsize (Abdukhakimov et al., 2023) fails to provide convergence theory and uses matrix stepsizes based on heuristics. In contrast, LCD2 utilizes local curvature from Assumption 2.1, and enjoys strong convergence guarantees.

## 2.4 EXPERIMENTS

We demonstrate superior empirical behavior of LCD2 over the GD with Polyak stepsize across several standard machine learning problems to which our theory applies. The presence of local curvature in our algorithms boosts their empirical performance when compared to their counterparts *not* taking advantage of local curvature.

# 3 THREE FLAVORS OF LOCAL CURVATURE DESCENT

We now describe our methods.

## 3.1 LOCAL CURVATURE DESCENT 1

Our first method, LCD1 is obtained by minimizing the upper bound from Assumption 2.1 where $y = x_k$, and letting $x_{k+1}$ be the minimizer:

$$\boxed{x_{k+1} = x_k - [\mathbf{C}(x_k) + L_{\mathbf{C}} \cdot \mathbf{I}]^{-1} \nabla f(x_k)} \tag{LCD1}$$

The derivation is routine; nevertheless, the detailed steps behind Equation (LCD1) can be found in Appendix B.1. If $\mathbf{C}(x) \equiv \mathbf{0}$ and we let $L = L_{\mathbf{C}}$, we recover GD with the constant stepsize $\gamma_k = \frac{1}{L}$. Note that just like GD, LCD1 is not adaptive to the smoothness parameter $L_{\mathbf{C}}$; this parameter is needed to perform a step.

## 3.2 LOCAL CURVATURE DESCENT 2

Given any $y \in \mathbb{R}^d$, let us define the *localization set*

$$\mathcal{L}_{\mathbf{C}}(y) := \left\{ x \in \mathbb{R}^d \ : \ M_{\mathbf{C}}^{\text{low}}(x, y) \leq f_\star \right\}. \tag{7}$$

Due to (5), we have $\mathcal{X}_\star \subset \mathcal{L}_{\mathbf{C}}(y)$, which justifies the use of the word "localization". Furthermore, $y \in \mathcal{X}_\star$ if and only if $y \in \mathcal{L}_{\mathbf{C}}(y)$. Therefore, $\mathcal{L}_{\mathbf{C}}(x_k)$ separates $\mathbb{R}^d$ in two regions: one containing $\mathcal{X}_\star$, the other the current iterate $y = x_k$. This allows us to design our second algorithm, LCD2: we simply project the current iterate $x_k$ into the localization set $\mathcal{L}_{\mathbf{C}}(x_k)$, bringing it closer to the set of optimal points $\mathcal{X}_\star$:

$$\boxed{x_{k+1} = \underset{x \in \mathcal{L}_{\mathbf{C}}(x_k)}{\arg\min} \ \tfrac{1}{2}\|x - x_k\|^2} \tag{LCD2}$$

It turns out that this projection problem has an implicit parametric solution of the form

$$x_{k+1} = x_k - [\mathbf{C}(x_k) + \beta_k \cdot \mathbf{I}]^{-1} \nabla f(x_k), \tag{LCD2}$$

where $\beta_k > 0$. Importantly, we show in Appendix C.1 that the structure of the problem is easy: the parameter $1/\beta_k$ is the unique root of a scalar equation, solvable efficiently. Moreover, if $\mathbf{C}(x_k)$ is a rank-one matrix or a multiple of $\mathbf{I}$, a closed-form solution exists. We present the details in Appendix C.3.

Note that when $\mathbf{C}(x) \equiv \mathbf{0}$, LCD2 becomes GD with Polyak stepsize. In general, LCD2 can be seen as a variant of GD with Polyak stepsize, enhanced with *local curvature*. The method no longer points in the negative gradient direction anymore, of course. We argue that one step of LCD2 improves on one step of GD with Polyak stepsize. Indeed, since $\mathcal{L}_{\mathbf{C}}(x_k) \subseteq \mathcal{L}_{\mathbf{0}}(x_k)$, with equality if and only if $\mathbf{C}(x_k) = \mathbf{0}$, the point $x_{k+1}$ obtained by LCD2 is closer to $\mathcal{X}_\star$ than what is achieved by a single step of GD with Polyak stepsize.

### 3.3 LOCAL CURVATURE DESCENT 3

Our last method, LCD3, was born out of the desire to remove the need for the univariate root-finding subroutine in order to execute the projection defining LCD2. This can be achieved by projecting using the norm given by the local curvature matrix $\mathbf{C}(x_k)$ instead:

$$\boxed{x_{k+1} = \underset{x \in \mathcal{L}_{\mathbf{C}}(x_k)}{\arg\min} \ \frac{1}{2} \|x - x_k\|^2_{\mathbf{C}(x_k)}} \tag{LCD3}$$

If $\mathbf{C}$ is invertible[3], this projection problem admits the closed-form solution

$$x_{k+1} = x_k - \left(1 - \sqrt{1 - \frac{2(f(x_k) - f_\star)}{\|\nabla f(x_k)\|^2_{\mathbf{C}^{-1}(x_k)}}}\right) \mathbf{C}^{-1}(x_k) \nabla f(x_k). \tag{LCD3}$$

The full derivation of this fact can be found in Appendix D.1. Although LCD3 uses the same localization set as LCD2, we do not provide any convergence theorem for this method. The variable metric nature of the projection makes it technically difficult to provide a meaningful analysis of this method. Nevertheless, we justify the introduction of LCD3 via its promising experimental behavior in Section 8 and Appendix G.

## 4 CONVERGENCE RATES

Having described the methods, this appears to be the right moment to present our main convergence results for LCD1 and LCD2.

**Theorem 4.1** (Convergence of LCD1)**.** *Let Assumption 2.1 be satisfied. For all $k \geq 1$, the iterates of LCD1 satisfy*

$$f(x_k) - f_\star \leq \frac{L_{\mathbf{C}} \|x_0 - x_\star\|^2}{2k}.$$

**Theorem 4.2** (Convergence of LCD2)**.** *Let Assumption 2.1 be satisfied. For all $k \geq 1$, the iterates of LCD2 satisfy*

$$\min_{1 \leq t \leq k} \ f(x_t) - f_\star \leq \frac{L_{\mathbf{C}} \|x_0 - x_\star\|^2}{2k}.$$

The proofs of these results can be found in Appendix B.2 and Appendix C.2, respectively. It is possible to derive linear convergence results under the assumption that $\mathbf{C}(x) \succeq \mu\mathbf{I}$ for all $x \in \mathbb{R}^d$ and some $\mu > 0$; however, we refrain from listing these for brevity reasons.

If $\mathbf{C}(x) \equiv \mathbf{0}$, and we let $L = L_{\mathbf{C}}$, these theorems recover the standard rates known for GD with the stepsize $1/L$ and GD with Polyak stepsize, respectively. So, we generalize these earlier results. However, it is possible for a function to satisfy Assumption 2.1 and not be $L$-smooth. In this sense, our results extend the reach of the classical theorems beyond the class of convex and $L$-smooth functions. On the other hand, if $f$ *is* convex and $L$-smooth, it may be possible that it satisfies

---

[3]We assume this for simplicity only.

Assumption 2.1 with some nonzero local curvature mapping $\mathbf{C}$, in which case we can choose $L_{\mathbf{C}}$ such that $L_{\mathbf{C}} \leq L$. Indeed,

$$\inf_{x \in \mathbb{R}^d} \lambda_{\min}(\mathbf{C}(x)) \leq L - L_{\mathbf{C}} \leq \sup_{x \in \mathbb{R}^d} \lambda_{\max}(\mathbf{C}(x)),$$

where $\lambda_{\min}(\cdot)$ (resp. $\lambda_{\max}(\cdot)$) represents the smallest (resp. largest) eigenvalue of the argument, confirming $L_{\mathbf{C}} \leq L$. However, it may be that $L_{\mathbf{C}} \ll L$, in which case our result leads to improved complexity. Nevertheless, the main allure of our methods is their attractive empirical behavior.

**Convex quadratics.** For convex quadratics, Assumption 2.1 is satisfied with $\mathbf{C}(x) = \nabla^2 f(x)$ and $L_{\mathbf{C}} = 0$. In this case, both LCD1 and LCD2 reduce to Newton's method, and converge in a single step. Moreover, Theorem 4.1 and Theorem 4.2 predict this one-step convergence behavior.

To validate our theoretical setting, we will show that functions satisfying Assumption 2.1 are easy to construct, well-behaved, and practically interesting.

## 5 LOCAL CURVATURE CALCULUS

We now mention a couple basic properties of functions that satisfy Inequalities (5) and (6).

**Lemma 5.1.** *Let $\alpha, \beta \in \mathbb{R}$ with $\beta \geq 0$. Suppose functions $f$ and $g$ satisfy inequality (5) with curvature mappings $\mathbf{C}_1$ and $\mathbf{C}_2$ respectively. Then:*

$$f + \alpha, \qquad \beta f, \quad and \quad f + g,$$

*satisfy Inequality (5) with curvature mappings $\mathbf{C}_1$, $\beta \mathbf{C}_1$, and $\mathbf{C}_1 + \mathbf{C}_2$ respectively.*

The proof of the lemma can be found in Appendix E.1. A particularly useful instantiation of Lemma 5.1 is presented in the following corollary.

**Corollary 5.1.** *If $f$ satisfies (5) and $g$ is convex, then $h := f + g$ also satisfies (5).*

Corollary 5.1 enables us to derive a variety of examples of functions satisfying inequality (5) by summing convex functions with instances from our class. Moreover, we can also show that inequality (5) is preserved under pre-composition with linear functions. Additional results for functions satisfying Assumption 2.1 can be found in Appendix E.

## 6 EXAMPLES OF FUNCTIONS SATISFYING ASSUMPTION 2.1

We first list three examples that satisfy both inequalities in Assumption 2.1. Firstly, observe that if a function is $L$-smooth, then it satisfies inequality (6) since $\mathbf{C}(x)$ is assumed to be a positive semi-definite matrix. We aim to find convex functions that satisfy our assumption in a non-trivial manner, i.e., $\mathbf{C}(x) \not\equiv \mathbf{0}$ and $\mathbf{C}(x) \not\equiv \mu \mathbf{I}$ for some $\mu > 0$.

**Example 6.1** (Huber loss). *Let $\delta > 0$ and consider the Huber loss function $h : \mathbb{R} \to \mathbb{R}$ given by*

$$h(x) = \begin{cases} \frac{1}{2} x^2 & |x| \leq \delta \\ \delta(|x| - \frac{1}{2}\delta) & |x| > \delta \end{cases}.$$

*Then $f = h^2$ satisfies Assumption 2.1 with constant $L_{\mathbf{C}} = 2\delta^2$ and curvature mapping*

$$\mathbf{C}(x) = \begin{cases} x^2 & |x| \leq \delta \\ \delta^2 & |x| > \delta \end{cases}.$$

Example 6.1 is particularly interesting because $\mathbf{C}(x) + 2\delta^2 \leq 3\delta^2$ for any $x \in \mathbb{R}$. By computing the second derivative of $f$, we can obtain the tightest $L$-smoothness constant; it is equal to $3\delta^2$. Therefore, the variable bound we derived is at least as good as the $L$-smoothness bound.

**Example 6.2** (Squared $p$ norm). *Let $p \geq 2$ and define $f : \mathbb{R}^d \to \mathbb{R}$ as $f(x) = \|x\|_p$. Then $f^2$ satisfies Assumption 2.1 with either of the two curvature mappings,*

$$\mathbf{C}(x) = \frac{2}{\|x\|_p^{p-2}} \operatorname{Diag}\left(|x_1|^{p-2}, \ldots, |x_d|^{p-2}\right), \quad \mathbf{C}(x) = 2 \nabla f(x) \nabla f(x)^\top,$$

*and constant $L_{\mathbf{C}} = 2(p-1)$.*

**Example 6.3** ($L_p$ regression). *Suppose $\mathbf{A} \in \mathbb{R}^{n \times d}$ and $b \in \mathbb{R}^n$. For $p \geq 2$, the function $f(x) = \|\mathbf{A}x - b\|_p^2$, satisfies Assumption 2.1 as a precomposition of Example 6.2 with an affine function.*

Therefore, linear regression in the squared $L_p$ norm satisfies our assumption. The $L_p$ regression problem has several applications in machine learning (Dasgupta et al., 2009; Musco et al., 2022; Yang et al., 2018). This includes low-rank matrix approximation, sparse recovery, data clustering, and learning tasks (Adil et al., 2023). In general, convex optimization in non-Euclidean geometries is a well-studied and important research direction. This motivates us to study $L_p$ norms further and understand how they can fit within our assumptions.

We can perform other simple modifications of $L_p$ norm that satisfy only inequality (5).

**Example 6.4.** *Let $p \geq 2$. Then $f(x) = \|x\|_p^p$ satisfies (5) with either of the curvature mappings*

$$\mathbf{C}_1(x) = \frac{1}{p-1}\nabla^2 f(x), \qquad \mathbf{C}_2(x) = \frac{1}{pf(x)}\nabla f(x)\nabla f(x)^\top.$$

We postpone comments to Appendix E.3. Using Corollary 5.1 and the above examples, we can construct regularized convex problems that satisfy our assumptions. For instance, we can add the square of an $L_p$ norm to the logistic loss function to obtain an objective function that satisfies (5), with the mapping from the regularizer. The objective function will be $L$-smooth, so it also satisfies inequality (6).

# 7 ABSOLUTELY CONVEX FUNCTIONS

In addition to the examples from Section 6, we now introduce the class of *absolutely convex* functions, and the problem of minimizing the sum of squares of absolutely convex functions. In this setting, as we shall show, the curvature mapping $\mathbf{C}$ satisfying Inequality (5) is readily available.

## 7.1 ABSOLUTE CONVEXITY

Absolutely convex functions are defined as follows.

**Definition 7.1** (Absolute convexity). A function $\phi : \mathbb{R}^d \to \mathbb{R}$ is absolutely convex if

$$\phi(x) \geq |\phi(y) + \langle \nabla\phi(y),\, x - y\rangle| \quad \forall x, y \in \mathbb{R}^d. \tag{8}$$

Above, $\nabla\phi(y)$ refers to a subgradient of $\phi$ at $y$. Geometrically, (8) means that linear approximations of $\phi$ are always above the graph of $-\phi$ in addition to being below the graph of $\phi$ (same as convexity),

$$-\phi(x) \leq \phi(y) + \langle \nabla\phi(y), x - y\rangle \leq \phi(x).$$

Thus, any absolutely convex function is necessarily convex and non-negative. A constant function is absolutely convex if and only if it is non-negative. A linear function is absolutely convex if and only if it is constant and non-negative. Moreover, the absolute value of any affine function is absolutely convex; that is, $\phi(x) = |\langle a, x\rangle + b|$ is absolutely convex. We avoid stating basic calculus rules as in Lemma 5.1, and opt to present only one interesting property, and one notable example. Many others can be found in Appendix F.

**Lemma 7.1.** *Absolutely convex functions have bounded subgradients.*

**Example 7.1.** *If $p \geq 1$, then $\phi(x) = \|x\|_p$ is absolutely convex.*

## 7.2 MINIMIZING THE SUM OF SQUARES OF ABSOLUTELY CONVEX FUNCTIONS

To conclude, we present the derivation of the curvature mapping $\mathbf{C}$ for the sum of squares of absolutely convex functions. Consider the optimization problem

$$x_\star = \underset{x \in \mathbb{R}^d}{\arg\min}\left\{f(x) := \frac{1}{n}\sum_{i=1}^n \phi_i^2(x)\right\}, \tag{9}$$

where each $\phi_i$ is absolutely convex and a solution, $x_\star$, is assumed to exist. Let $f_i := \phi_i^2$, so that $\nabla f_i(x) = 2\phi_i(x)\nabla\phi_i(x)$. The gradient of $f$ is given by

$$\nabla f(x) = \frac{1}{n}\sum_{i=1}^n \nabla f_i(x) = \frac{2}{n}\sum_{i=1}^n \phi_i(x)\nabla\phi_i(x).$$

Since $\phi_i$ is absolutely convex, $f_i$ is necessarily convex. Indeed, by squaring both sides of the defining inequality (8), we get

$$f_i(y) + \langle \nabla f_i(y), x - y \rangle + \langle \nabla \phi_i(y) \nabla \phi_i(y)^\top (x - y), x - y \rangle \leq f_i(x), \qquad \forall x, y \in \mathbb{R}^d.$$

Summing these inequalities across $i$ and taking the average, we find that the curvature mapping can be set to

$$\mathbf{C}(x) = \tfrac{2}{n} \sum_{i=1}^{n} \nabla \phi_i(x) \nabla \phi_i(x)^\top.$$

In Appendix G, we provide experiments on objective functions that are in this class.

## 8 EXPERIMENTS

To illustrate practical performance of the presented methods, we run a series of experiments on MacBook Pro with Apple M1 chip and 8GB of RAM. We use datasets from LibSVM (Chang & Lin, 2011). We implemented all algorithms in Python.

Let us focus on solving

$$f(x) = \tfrac{1}{n} \sum_{i=1}^{n} \log(1 + e^{-b_i a_i x}) + \lambda \|x\|_p^p,$$

where $a_i \in \mathbb{R}^d$ and $b_i \in \{-1, 1\}$ are the data samples associated with a binary classification problem. The regularization weight $\lambda$ is set proportionally to the $L$-smoothness constant of the logistic regression instance.

In the first experiment, we use $L_2$ regularization. Therefore, $f$ is $L$-smooth and $\mu$-strongly-convex, so $\mathbf{C}(x) \equiv \mu \mathbf{I}$. As mentioned previously, in this setting, LCD1 recovers GD and LCD2 has a closed-form solution coinciding with LCD3.

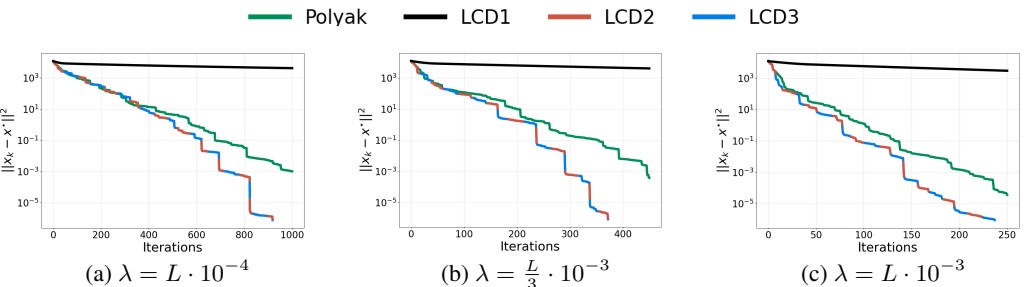

(a) $\lambda = L \cdot 10^{-4}$  (b) $\lambda = \frac{L}{3} \cdot 10^{-3}$  (c) $\lambda = L \cdot 10^{-3}$

Figure 1: Logistic regression on `a2a` dataset with $L_2$ regularization.

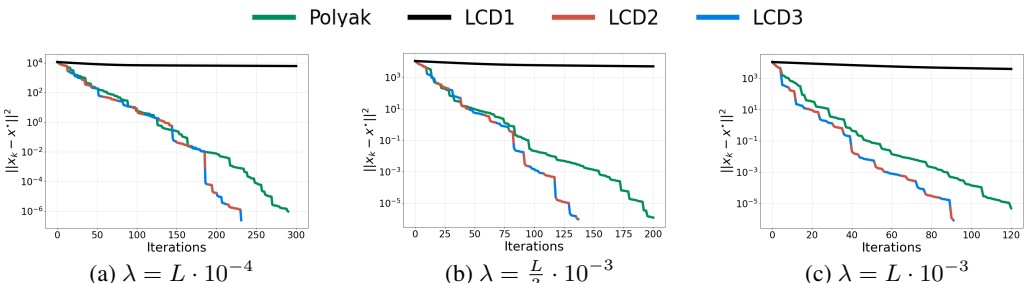

(a) $\lambda = L \cdot 10^{-4}$  (b) $\lambda = \frac{L}{3} \cdot 10^{-3}$  (c) $\lambda = L \cdot 10^{-3}$

Figure 2: Logistic regression on `mushrooms` dataset with $L_2$ regularization.

Figures 1–2 show that LCD2 consistently outperforms Polyak. As expected, the gap increases with $\lambda$ because $\mathbf{C}(x)$ only stores information about the regularizer. Thus, increasing $\lambda$ shrinks the localization set of LCD2 so its improvement over Polyak grows. Importantly, since LCD2 has a closed form solution, its cost-per-iteration is the same as Polyak.

In the next experiment, we use $L_3$ regularization. In Example 6.4 we propose two $\mathbf{C}(x)$ matrix candidates for $\|x\|_p^p$. Here we decide on the diagonal variant $\mathbf{C}_1(x)$. The objective function is no longer $L$-smooth, due to the non-smooth regularizer. As a result, we run LCD1 with the smallest $L_{\mathbf{C}}$ such that the method converges. Additionally, LCD2 no longer has a closed form solution, so the projection algorithm must be deployed. To perform a fair comparison of our algorithms, we show both time and iteration plots.

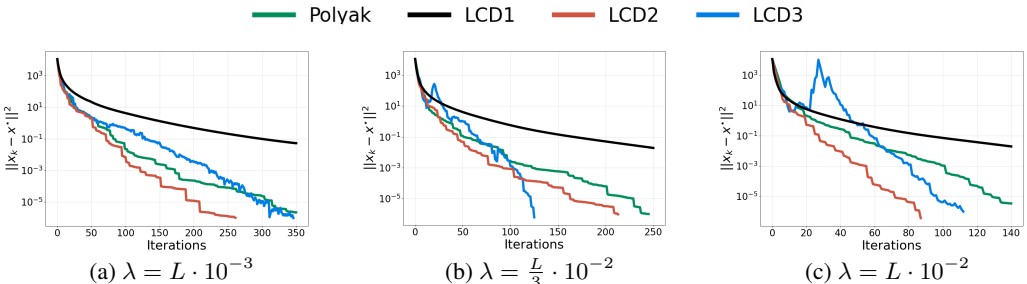

(a) $\lambda = L \cdot 10^{-3}$     (b) $\lambda = \frac{L}{3} \cdot 10^{-2}$     (c) $\lambda = L \cdot 10^{-2}$

Figure 3: Logistic regression on `mushrooms` dataset with $L_3$ regularization - iteration convergence.

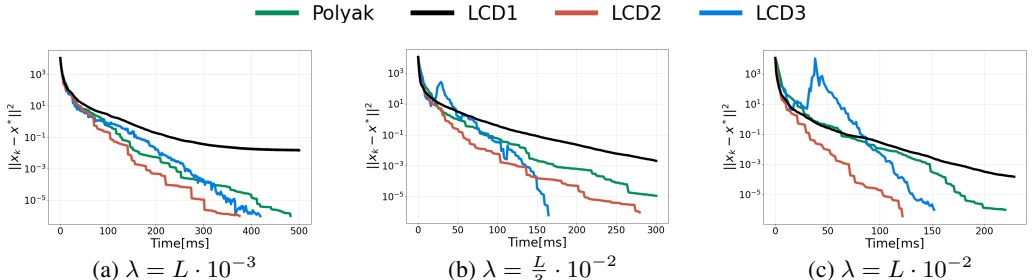

(a) $\lambda = L \cdot 10^{-3}$     (b) $\lambda = \frac{L}{3} \cdot 10^{-2}$     (c) $\lambda = L \cdot 10^{-2}$

Figure 4: Logistic regression on `mushrooms` dataset with $L_3$ regularization - time convergence.

Figure 3 displays similar to the $L_2$ case improvement of LCD2 over Polyak, which grows with $\lambda$. Our heuristic LCD3 can produce satisfying results, experimentally. However, its convergence cannot be guaranteed. In fact, as $\lambda$ increases it becomes unstable. LCD1 converges at comparable pace with the other three methods at initial steps, yet the limited adaptiveness slows it down later on.

Figure 4 shows convergence of our methods in time. One may point that the plots look almost identical to the iteration counterpart. The main reason is the cost of computing the gradient, which is $\mathcal{O}(nd)$. All other operations performed by LCD3 and LCD1 are $\mathcal{O}(d)$. The method with the most expensive update rule is LCD2. At every step it performs around 5 rounds of the projection algorithm, each costing $\mathcal{O}(d)$. We conclude that all the methods have comparable computational cost per iteration, as the main expense is the gradient evaluation. While the complexities discussed above are for diagonal matrices, we remark that the general $\mathcal{O}(d^3)$ cost is bearable when $n \gg d$. Moreover, our examples usually allow cheap diagonal matrix methods.

Further experiments with ridge regression and sum of squared Huber losses are in Appendix G.

## 9 CONCLUSION

We explored adaptive matrix-valued stepsizes under novel assumptions that reinforce convexity and $L$-smoothness with extra curvature information. Under our assumptions, we proposed LCD1 and LCD2, which generalize GD with constant stepsize and Polyak stepsize, respectively. Moreover, we provided convergence theorems for both of these algorithms. We also proposed LCD3 which displays promising experimental behavior. Our key insight is that, for some problems, we have certain local curvature information that can be readily exploited. We tested the methods on these problems using a variety of realistic datasets, demonstrating good empirical performance.

The main limitation of our analysis is the restriction to a deterministic setting. We also acknowledge that the assumption is yet to explore in its entirety. As a matter of fact, the most natural extension of the present work is including stochasticity and understanding the full potential of Assumption 2.1.

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

# A APPENDIX

## CONTENTS

# B  LOCAL CURVATURE DESCENT 1 (LCD1)

## B.1  DERIVATION

Suppose that the upper bound in (6) from Assumption 2.1) holds. Then, at a given point $x_{k+1} \in \mathbb{R}^d$, we have:

$$f(x_{k+1}) \le f(x_k) + \langle \nabla f(x_k), x_{k+1} - x_k \rangle + \frac{1}{2} \|x_{k+1} - x_k\|^2_{\mathbf{C}(x_k)+L_{\mathbf{C}}\mathbf{I}} \qquad \forall x_k \in \mathbb{R}^d.$$

Minimizing the right hand side with respect to $x_{k+1}$ we find that:

$$x_{k+1} = x_k - [\mathbf{C}(x_k) + L_{\mathbf{C}}\mathbf{I}]^{-1} \nabla f(x_k). \tag{10}$$

In particular, the matrix that pre-multiplies the vector is always invertible, since $\mathbf{C}(x)$ is positive semi-definite for each $x \in \mathbb{R}^d$.

## B.2  CONVERGENCE PROOF

**Lemma B.1.** *Let Assumption 2.1 hold. For all $k \ge 0$, the sequence $(x_k)_{k \in \mathbb{N}}$ of* LCD1 *is such that:*

$$\|x_{k+1} - x_\star\|^2_2 - \|x_k - x_\star\|^2_2 \le -\frac{2}{L_{\mathbf{C}}}\left(f(x_{k+1}) - f(x_\star)\right), \quad \forall k \in \mathbb{N}. \tag{11}$$

*Proof.* The proof is achieved by carefully bounding terms. For this reason, we split it into three steps.

We seek a connection between the two distances in the geometry induced by $\tilde{\mathbf{C}}(x_k) := \mathbf{C}(x_k) + L_{\mathbf{C}}\mathbf{I}$:

$$\begin{aligned}
\|x_k - x_\star\|^2_{\tilde{\mathbf{C}}(x_k)} &= \|x_k - x_{k+1} + x_{k+1} - x_\star\|^2_{\tilde{\mathbf{C}}(x_k)} \\
&= \|x_k - x_{k+1}\|^2_{\tilde{\mathbf{C}}(x_k)} + 2\left\langle [\tilde{\mathbf{C}}(x_k)](x_k - x_{k+1}), x_{k+1} - x_\star \right\rangle \\
&\qquad + \|x_{k+1} - x_\star\|^2_{\tilde{\mathbf{C}}(x_k)} \\
&= \|x_k - x_{k+1}\|^2_{\tilde{\mathbf{C}}(x_k)} + 2\langle \nabla f(x_k), x_{k+1} - x_\star \rangle + \|x_{k+1} - x_\star\|^2_{\tilde{\mathbf{C}}(x_k)} \\
&= \|x_{k+1} - x_k\|^2_{\tilde{\mathbf{C}}(x_k)} + 2\langle \nabla f(x_k), x_{k+1} - x_\star \rangle + \|x_{k+1} - x_\star\|^2_{\tilde{\mathbf{C}}(x_k)}.
\end{aligned}$$

Rearranging the terms we obtain

$$\begin{aligned}
\|x_{k+1} - x_\star\|^2_{\tilde{\mathbf{C}}(x_k)} - \|x_k - x_\star\|^2_{\tilde{\mathbf{C}}(x_k)} &= -\|x_{k+1} - x_k\|^2_{\tilde{\mathbf{C}}(x_k)} - 2\langle \nabla f(x_k), x_{k+1} - x_\star \rangle \\
&= -\|x_{k+1} - x_k\|^2_{\tilde{\mathbf{C}}(x_k)} \\
&\qquad - 2\langle \nabla f(x_k), x_{k+1} - x_k + x_k - x_\star \rangle \\
&= -\|x_{k+1} - x_k\|^2_{\tilde{\mathbf{C}}(x_k)} - 2\langle \nabla f(x_k), x_{k+1} - x_k \rangle \\
&\qquad + 2\langle \nabla f(x_k), x_\star - x_k \rangle.
\end{aligned}$$

In particular, we wish to bound the inner products.

Rearranging the lower bound (5) in Assumption 2.1 for the pair $(x_k, x_\star)$:

$$2\langle \nabla f(x_k), x_\star - x_k \rangle \le 2(f(x_\star) - f(x_k)) - \|x_k - x_\star\|^2_{\mathbf{C}(x_k)}.$$

In a similar way, massaging the upper bound (6) of Assumption 2.1 for the pair $(x_{k+1}, x_k)$ one can derive:

$$-2\langle \nabla f(x_k), x_{k+1} - x_k \rangle \le \|x_{k+1} - x_k\|^2_{\tilde{\mathbf{C}}(x_k)} + 2(f(x_k) - f(x_{k+1})).$$

Combining two previous steps we find:

$$\begin{aligned}
\|x_{k+1} - x_\star\|^2_{\tilde{\mathbf{C}}(x_k)} - \|x_k - x_\star\|^2_{\tilde{\mathbf{C}}(x_k)} &\le -\|x_{k+1} - x_k\|^2_{\tilde{\mathbf{C}}(x_k)} + \|x_{k+1} - x_k\|^2_{\tilde{\mathbf{C}}(x_k)} \\
&\qquad + 2(f(x_k) - f(x_{k+1})) + 2(f(x_\star) - f(x_k)) - \|x_k - x_\star\|^2_{\mathbf{C}(x_k)} \\
&= 2(f(x_\star) - f(x_{k+1})) - \|x_k - x_\star\|^2_{\mathbf{C}(x_k)}.
\end{aligned}$$

The term $\|x_k - x_\star\|^2_{\mathbf{C}(x_k)}$ is on both sides of the inequality, so we can cancel it out:

$$\|x_{k+1} - x_\star\|^2_{\tilde{\mathbf{C}}(x_k)} - \|x_k - x_\star\|^2_{L_{\mathbf{C}}\mathbf{I}} \leq 2(f(x_\star) - f(x_{k+1})).$$

Having almost removed all the $\mathbf{C}(x_k)$ norms, it suffices to apply the crude bound:

$$L_{\mathbf{C}}\|x_{k+1} - x_\star\|^2 = \|x_{k+1} - x_\star\|^2_{L_{\mathbf{C}}\mathbf{I}} \leq \|x_{k+1} - x_\star\|^2_{\tilde{\mathbf{C}}(x_k)},$$

which holds since $L_{\mathbf{C}}\mathbf{I} \preceq \mathbf{C}(x_k) + L_{\mathbf{C}}\mathbf{I} = \tilde{\mathbf{C}}(x_k)$.

Therefore, we obtain

$$\|x_{k+1} - x_\star\|^2_{L_{\mathbf{C}}\mathbf{I}} - \|x_k - x_\star\|^2_{L_{\mathbf{C}}\mathbf{I}} \leq 2(f(x_\star) - f(x_{k+1})) = -2(f(x_{k+1}) - f(x_\star)).$$

Reordering gives the claim. □

**Lemma B.2.** *For any $k \in \mathbb{N}$, the iterations of* LCD1 *satisfy:*

$$f(x_{k+1}) - f(x_k) \leq -\frac{1}{2}\|\nabla f(x_k)\|^2_{(\mathbf{C}(x_k)+L_{\mathbf{C}}\mathbf{I})^{-1}} \leq 0. \tag{12}$$

*Proof.* Let us remind the form of the updates for each $k \in \mathbb{N}$

$$x_{k+1} = x_k - \left[\tilde{\mathbf{C}}(x_k)\right]^{-1}\nabla f(x_k),$$

where $\tilde{\mathbf{C}}(x_k) = \mathbf{C}(x_k) + L_{\mathbf{C}}\mathbf{I}$.

By Assumption 2.1, we know that

$$f(x_{k+1}) \leq f(x_k) - \left\langle\nabla f(x_k), [\tilde{\mathbf{C}}(x_k)]^{-1}\nabla f(x_k)\right\rangle + \frac{1}{2}\left\langle\nabla f(x_k), [\tilde{\mathbf{C}}(x_k)]^{-1}\nabla f(x_k)\right\rangle$$

$$= f(x_k) - \frac{1}{2}\|\nabla f(x_k)\|^2_{[\tilde{\mathbf{C}}(x_k)]^{-1}},$$

and the claim follows by simple rearrangement. □

Having the lemmas established, let us proceed to the proof of Theorem 4.1. We want to show that if $f : \mathbb{R}^d \to \mathbb{R}$ satisfies Assumption 2.1 then for any $k \in \mathbb{N}$, the iterates of LCD1 are such that:

$$f(x_k) - f(x_\star) \leq \frac{L_{\mathbf{C}}}{2k}\|x_0 - x_\star\|^2. \tag{13}$$

*Proof.* We use a standard Lyapunov function proof technique. For completeness, let us report it.

By Lemma B.2, function values get closer to $f_\star$ across iterations. Lemma B.1, the vectors get closer in norm to an optimum.

Then, we can combine the two positive decreasing terms $L_{\mathbf{C}}\|x_k - x_\star\|^2$ and $f(x_k) - f(x_\star)$ into a Lyapunov energy function:

$$\mathcal{E}_k := L_{\mathbf{C}}\|x_k - x_\star\|^2 + 2k(f(x_k) - f(x_\star)), \qquad \forall k \in \mathbb{N}.$$

In particular, $\mathcal{E}_0 = L_{\mathbf{C}}\|x_0 - x_\star\|^2$, and we claim that $\mathcal{E}_k$ is a decreasing function. To see this, we start by rewriting the difference:

$$\mathcal{E}_{k+1} - \mathcal{E}_k = 2(k+1)(f(x_{k+1}) - f(x_\star)) - 2k(f(x_k) - f(x_\star))$$
$$+ L_{\mathbf{C}}\|x_{k+1} - x_\star\|^2 - L_{\mathbf{C}}\|x_k - x_\star\|^2$$
$$= 2(f(x_{k+1}) - f(x_\star)) + 2k(f(x_{k+1}) - f(x_\star) - f(x_k) + f(x_\star))$$
$$+ L\|x_{k+1} - x_\star\|^2 - L_{\mathbf{C}}\|x_k - x_\star\|^2$$
$$= 2(f(x_{k+1}) - f(x_\star)) + 2k(f(x_{k+1}) - f(x_k))$$
$$+ L_{\mathbf{C}}\|x_{k+1} - x_\star\|^2 - L_{\mathbf{C}}\|x_k - x_\star\|^2.$$

It is evident that we can apply our Lemmas as follows:

$$f(x_{k+1}) - f(x_k) \leq 0 \qquad \text{by Lemma B.2;} \quad (14)$$

$$L_{\mathbf{C}}\|x_{k+1} - x_\star\|^2 - L_{\mathbf{C}}\|x_k - x_\star\|^2 \leq f(x_{k+1}) - f(x_\star) \qquad \text{by Lemma B.1.} \quad (15)$$

Putting everything together:

$$\mathcal{E}_{k+1} - \mathcal{E}_k \leq 2(f(x_{k+1}) - f(x_\star)) - 2(f(x_{k+1}) - f(x_\star)) = 0,$$

showing that $\mathcal{E}_k$ is decreasing. As a particular case, we then find:

$$2k(f(x_k) - f(x_\star)) \leq \mathcal{E}_k \leq \mathcal{E}_0 = L_{\mathbf{C}}\|x_0 - x_\star\|^2,$$

which reordered recovers the rate of GD with stepsize $\frac{1}{L_{\mathbf{C}}}$, i.e.

$$f(x_k) - f(x_\star) \leq \frac{L_{\mathbf{C}}\|x_0 - x_\star\|^2}{2k}.$$

$\square$

**Remark.** For quadratic functions Assumption 2.1 is satisfied with $\mathbf{C}(x)$ equal to the Hessian, and $L_{\mathbf{C}} = 0$. Thus, LCD1 convergences in one step for this class of functions.

## C  LOCAL CURVATURE DESCENT 2 (LCD2)

### C.1  DERIVATION

Consider the minimization problem for the update step of LCD2:

$$\min_{x \in \mathcal{L}_{\mathbf{C}}(x_k)} \frac{1}{2}\|x - x_k\|^2, \tag{16}$$

where

$$\mathcal{L}_{\mathbf{C}}(x_k) = \left\{ x \in \mathbb{R}^d \mid f(x_k) + \langle \nabla f(x_k), x - x_k \rangle + \frac{1}{2}\|x - x_k\|_{\mathbf{C}(x_k)}^2 \leq f_\star \right\}. \tag{17}$$

If $\mathbf{C}(x_k)$ is the zero matrix, we know this problem has a closed-form solution. Therefore, we focus on the case where $\mathbf{C}(x_k)$ is a non-zero matrix. Moreover, we assume that $x_k \notin \mathcal{X}_\star$. The Lagrangian of this problem is:

$$\mathscr{L}(x, \beta) = \frac{1}{2}\|x - x_k\|^2 + \beta\left(\frac{1}{2}\|x - x_k\|_{\mathbf{C}(x_k)}^2 + \langle \nabla f(x_k), x - x_k \rangle + f(x_k) - f_\star\right),$$

where $\beta \geq 0$. For optimal $\bar{x}$ and $\bar{\beta}$ we have that $\nabla_x \mathscr{L}(\bar{x}, \bar{\beta}) = 0$. Therefore,

$$\bar{x} - x_k + \bar{\beta}\left(\|\bar{x} - x_k\|_{\mathbf{C}(x_k)}^2 + \nabla f(x_k)\right) = 0.$$

Isolating for $\bar{x}$, we find that:

$$\bar{x} = x_k - \bar{\beta}\left[\mathbf{I} + \bar{\beta}\mathbf{C}(x_k)\right]^{-1}\nabla f(x_k).$$

We can see $\bar{\beta} \neq 0$ so the constraint is tight. The next step would be to substitute $\bar{x}$ into the constraint and solve for $\bar{\beta}$:

$$\frac{1}{2}\|\bar{x} - x_k\|_{\mathbf{C}(x_k)}^2 + \langle \nabla f(x_k), \bar{x} - x_k \rangle + f(x_k) - f_\star = 0.$$

Despite the left-hand side being a scalar function of $\bar{\beta}$, we cannot obtain a closed-form solution for $\bar{\beta}$. However, we can use an iterative root-finding sub-routine such as Newton's method to get an approximation of $\bar{\beta}$ cheaply and effectively. By substituting in the value of $\bar{x}$, we see that we need to find the root of the following function:

$$H(\beta) := \frac{\beta^2}{2} g_k^\top \left[\mathbf{I} + \beta\mathbf{C}(x_k)\right]^{-\top} \mathbf{C}(x_k) \left[\mathbf{I} + \beta\mathbf{C}(x_k)\right]^{-1} g_k \tag{18}$$
$$- \beta g_k^\top \left[\mathbf{I} + \beta\mathbf{C}(x_k)\right]^{-1} g_k + \Delta_k.$$

To simplify notation, let $\mathbf{C} := \mathbf{C}(x_k)$, $g := \nabla f(x_k)$ and $\Delta_k := f(x_k) - f_\star$.

In the following proposition, we confirm that $H$ has a root in the interval $[0, \infty)$. We also show that $H$ is convex and monotonically decreasing on that interval. Therefore, Newton's method is guaranteed to converge to the root of $H$ at a quadratic rate. In particular, we do not need $H$ to be monotonically decreasing; nonetheless, it is an interesting property of the problem.

**Proposition C.1** (Properties of $H$). *Let $H$ be defined as in Equation* (18). *Then, for $\beta \geq 0$:*

$$H(0) > 0, \quad H'(\beta) < 0, \quad H''(\beta) > 0, \quad \lim_{\beta \to \infty} H(\beta) < 0. \tag{19}$$

*Proof.* W.l.o.g. assume that $\mathbf{C}$ is a symmetric matrix. As a result, $\mathbf{C}$ is orthogonally diagonalizable so we let $\mathbf{C} = \mathbf{Q}\mathbf{D}\mathbf{Q}^\top$ where $\mathbf{D}$ is a diagonal matrix, and $\mathbf{Q}$ is an orthogonal matrix such that $\mathbf{Q}\mathbf{Q}^\top = \mathbf{I}$. Manipulating the inverse matrix in the definition of $H$, we find:

$$\left[\mathbf{I} + \beta\mathbf{C}(x_k)\right]^{-1} = \left[\mathbf{Q}\mathbf{Q}^\top + \beta\mathbf{Q}\mathbf{D}\mathbf{Q}^\top\right]^{-1} = \left[\mathbf{Q}(\mathbf{I} + \beta\mathbf{D})\mathbf{Q}^\top\right]^{-1} = \mathbf{Q}\left[\mathbf{I} + \beta\mathbf{D}\right]^{-1}\mathbf{Q}^\top.$$

Let $\tilde{g} := \mathbf{Q}^\top g$. Let $D_i$ represent the $i^{\text{th}}$ entry of the diagonal of $\mathbf{D}$ and $\tilde{g}_i$ represent the $i^{\text{th}}$ entry in $\tilde{g}$. We rewrite $H$ as:

$$
\begin{aligned}
H(\beta) &= \frac{\beta^2}{2} g^\top \mathbf{Q} \left[\mathbf{I} + \beta\mathbf{D}\right]^{-1} \mathbf{D} \left[\mathbf{I} + \beta\mathbf{D}\right]^{-1} \mathbf{Q}^\top g - \beta g^\top \mathbf{Q} \left[\mathbf{I} + \beta\mathbf{D}\right]^{-1} \mathbf{Q}^\top g + \Delta_k \\
&= \frac{\beta^2}{2} \tilde{g}^\top \left[\mathbf{I} + \beta\mathbf{D}\right]^{-1} \mathbf{D} \left[\mathbf{I} + \beta\mathbf{D}\right]^{-1} \tilde{g} - \beta \tilde{g}^\top \left[\mathbf{I} + \beta\mathbf{D}\right]^{-1} \tilde{g} + \Delta_k \\
&= \frac{\beta^2}{2} \sum_{i=1}^{d} \frac{\tilde{g}_i^2 D_i}{(1 + \beta D_i)^2} - \beta \sum_{i=1}^{d} \frac{\tilde{g}_i^2}{1 + \beta D_i} + \Delta_k.
\end{aligned}
$$

By inspection, $H$ is a rational function and the derivative is easily found;

$$
\begin{aligned}
H'(\beta) &= \beta \sum_{i=1}^{d} \frac{\tilde{g}_i^2 D_i}{(1 + \beta D_i)^2} - \beta^2 \sum_{i=1}^{d} \frac{\tilde{g}_i^2 D_i^2}{(1 + \beta D_i)^3} - \sum_{i=1}^{d} \frac{\tilde{g}_i^2}{1 + \beta D_i} + \beta \sum_{i=1}^{d} \frac{\tilde{g}_i^2 D_i}{(1 + \beta D_i)^2} \\
&= - \sum_{i=1}^{d} \frac{\tilde{g}_i^2}{(1 + \beta D_i)^3}.
\end{aligned}
$$

Since $\mathbf{C}$ is a positive semi-definite matrix, $D_i \geq 0$ for all $i$. Thus for $\beta \geq 0$, we have $H'(\beta) \leq 0$. Since the null-space of an orthogonal matrix is the singleton of the zero vector, the product $\tilde{g} = \mathbf{Q}^\top g$ is different than zero when $g \neq 0$, which holds by the assumption $x_k \notin \mathcal{X}_\star$. Therefore, there is at least one $\tilde{g}_i$ that is non-zero and thus, $H'(\beta) \neq 0$. The second derivative of $H$ is,

$$
H''(\beta) = 3 \sum_{i=1}^{d} \frac{\tilde{g}_i^2 D_i}{(1 + \beta D_i)^4}.
$$

By similar arguments used for the first derivative, we can show that $H''(\beta) > 0$.
To conclude, we will show that $\lim_{\beta \to \infty} H(\beta) < 0$. We discuss two cases separately.
Suppose $\mathbf{C}$ is not invertible. Then, there exists an entry $i$ of $\mathbf{D}$ such that $D_i = 0$. Without loss of generality, suppose that the last entry $D_d$, is equal to $0$. The same reasoning will apply if more than one entry is equal to $0$. Taking the limit:

$$
\begin{aligned}
\lim_{\beta \to \infty} H(\beta) &= \lim_{\beta \to \infty} \frac{\beta^2}{2} \sum_{i=1}^{d} \frac{\tilde{g}_i^2 D_i}{(1 + \beta D_i)^2} - \beta \sum_{i=1}^{d} \frac{\tilde{g}_i^2}{1 + \beta D_i} + \Delta_k \\
&= \lim_{\beta \to \infty} \frac{\beta^2}{2} \sum_{i=1}^{d-1} \frac{\tilde{g}_i^2 D_i}{(1 + \beta D_i)^2} - \beta \sum_{i=1}^{d-1} \frac{\tilde{g}_i^2}{1 + \beta D_i} - \beta \tilde{g}_i^2 + \Delta_k \\
&= \frac{1}{2} \sum_{i=1}^{d-1} \frac{\tilde{g}_i^2}{D_i} - \sum_{i=1}^{d-1} \frac{\tilde{g}_i^2}{D_i} + \Delta_k + \lim_{\beta \to \infty} -\beta \tilde{g}_i^2 \\
&= -\infty \\
&< 0.
\end{aligned}
$$

Now suppose that $\mathbf{C}$ is invertible. Then, $D_i > 0$ for all $i$. Differently from before:

$$
\begin{aligned}
\lim_{\beta \to \infty} H(\beta) &= \lim_{\beta \to \infty} \frac{\beta^2}{2} \sum_{i=1}^{d} \frac{\tilde{g}_i^2 D_i}{(1 + \beta D_i)^2} - \beta \sum_{i=1}^{d} \frac{\tilde{g}_i^2}{1 + \beta D_i} + \Delta_k \\
&= \frac{1}{2} \sum_{i=1}^{d} \frac{\tilde{g}_i^2}{D_i} - \sum_{i=1}^{d} \frac{\tilde{g}_i^2}{D_i} + \Delta_k \\
&= \frac{1}{2} \tilde{g}^\top \mathbf{D}^{-1} \tilde{g} - \tilde{g}^\top \mathbf{D}^{-1} \tilde{g} + \Delta_k \\
&= -\frac{1}{2} g^\top \mathbf{Q} \mathbf{D}^{-1} \mathbf{Q}^\top g + \Delta_k \\
&= -\frac{1}{2} g^\top \mathbf{C}^{-1} g + \Delta_k.
\end{aligned}
$$

Recalling our definitions, the right hand side is:

$$\lim_{\beta \to \infty} H(\beta) = -\frac{1}{2} \|\nabla f(x_k)\|^2_{[\mathbf{C}(x_k)]^{-1}} + f(x_k) - f_\star.$$

By Lemma E.5, $\lim_{\beta \to \infty} H(\beta) \leq 0$. The inequality is strict when $f(x_k) - f_\star \neq \frac{1}{2} \|\nabla f(x_k)\|^2_{\mathbf{C}^{-1}}$. In the case where equality holds, we have that $\lim_{\beta \to \infty} H(\beta) = 0$. Therefore, $H$ does not have a root in the interval $[0, \infty)$ but the solution to the optimization problem is obtain when $\beta = \infty$. This corresponds to the following optimal solution $\bar{x}$:

$$\bar{x} = x_k - \mathbf{C}^{-1}(x_k)\nabla f(x_k). \tag{20}$$

Interestingly, under the same condition, LCD3 takes a step in the form $x_{k+1} = x_k - \mathbf{C}^{-1}(x_k)\nabla f(x_k)$. An example of a setting where the equality condition holds is when $f$ is a convex quadratic and $\mathbf{C}$ is the Hessian of $f$. One can see that the update step of LCD3 and LCD2 are equivalent to Newton's method for that case so they both converge in one iteration. $\square$

It may seem that using Newton's root finding method is impractical because computing $H$ defined in Equation (18) for a given $\beta$ requires performing a matrix inversion. However, this can be avoided by computing the eigendecomposition of $\mathbf{C}(x_k)$ at the beginning of each step of LCD2. Then each subsequent evaluation of $H$ done by Newton's method sub-routine only requires inverting a diagonal matrix and not the full matrix. Thus, the main cost at each step of LCD2 is computing the eigendecomposition of $\mathbf{C}(x_k)$ once, which in practice is much faster than computing the inverse. Furthermore, if $\mathbf{C}(x_k)$ is a diagonal matrix, the eigendecomposition of $\mathbf{C}(x_k)$ is itself so each step of LCD2 becomes even cheaper. Also, in practice, Newton's method for root-finding is terminated when $|H| < \epsilon$. Therefore, in the case where

$$f(x_k) - f_\star = \frac{1}{2} \|\nabla f(x_k)\|^2_{\mathbf{C}^{-1}(x_k)},$$

the method will run until a large enough $\beta$ is obtained and the step will become numerically equivalent to $\bar{x} = x_k - \mathbf{C}^{-1}(x_k)\nabla f(x_k)$.

On a related note, Newton's method is used to solve a similar constrained optimization problem for trust region methods, namely, the trust region sub-problem (Nocedal & Wright, 2006). Practical versions of such algorithms do not iterate until convergence but are content with an approximate solution that can be obtained in two or three iterations.

### C.2 CONVERGENCE PROOF

**Lemma C.1.** *Let Assumption 2.1 hold. For all $k \geq 0$, the sequence $(x_k)_{k \in \mathbb{N}}$ of LCD2 obeys the recursion:*

$$\|x_{k+1} - x_\star\|^2 \leq \|x_k - x_\star\|^2 - \|x_{k+1} - x_k\|^2.$$

*Hence, for any $k \geq 1$, we have*

$$\min_{0 \leq t \leq k-1} \|x_{t+1} - x_t\|^2 \leq \frac{\|x_0 - x_\star\|^2}{k}. \tag{21}$$

*Proof.* Let us write down the first-order optimality conditions for the optimization problem at Step 3 of LCD2:

$$\langle x_k - x_{k+1}, x_{k+1} - y \rangle \geq 0, \qquad \forall y \in \mathcal{L}_{\mathbf{C}}(x_k). \tag{22}$$

Since $x_\star \in \mathcal{L}_{\mathbf{C}}(x_k)$, for any $k \geq 0$ we have

$$\begin{aligned}
\|x_{k+1} - x_\star\|^2 &= \|x_k - x_\star\|^2 - 2\langle x_k - x_{k+1}, x_k - x_\star \rangle + \|x_{k+1} - x_k\|^2 \\
&= \|x_k - x_\star\|^2 - 2\langle x_k - x_{k+1}, x_{k+1} - x_\star \rangle \\
&\quad - 2\langle x_k - x_{k+1}, x_k - x_{k+1} \rangle + \|x_{k+1} - x_k\|^2 \\
&= \|x_k - x_\star\|^2 - 2\langle x_k - x_{k+1}, x_{k+1} - x_\star \rangle - \|x_{k+1} - x_k\|^2 \\
&\overset{(22)}{\leq} \|x_k - x_\star\|^2 - \|x_{k+1} - x_k\|^2.
\end{aligned}$$

Summing up these inequalities for $k = 0, \ldots, K-1$, we obtain (21). $\square$

Let us proceed to the proof of Theorem 4.2 for LCD2. We show that any if $f : \mathbb{R}^d \to \mathbb{R}$ satisfies Assumption 2.1 for any $k \geq 1$ the iterates of LCD2 are such that:

$$\min_{1 \leq t \leq k} f(x_t) - f_\star \leq \frac{L_\mathbf{C} \|x_0 - x_\star\|^2}{2k}.$$

*Proof.* Since $x_{k+1} \in \mathcal{L}_\mathbf{C}(x_k)$, we have

$$f(x_{k+1}) \stackrel{(6)}{\leq} f(x_k) + \langle \nabla f(x_k), x_{k+1} - x_k \rangle + \frac{1}{2} \|x_{k+1} - x_k\|^2_{\mathbf{C}(x_k) + L_\mathbf{C} \mathbf{I}}$$

$$= f(x_k) + \langle \nabla f(x_k), x_{k+1} - x_k \rangle + \frac{1}{2} \|x_{k+1} - x_k\|^2_{\mathbf{C}(x_k)}$$

$$- \frac{1}{2} \|x_{k+1} - x_k\|^2_{\mathbf{C}(x_k)} + \frac{1}{2} \|x_{k+1} - x_k\|^2_{\mathbf{C}(x_k) + L_\mathbf{C} \mathbf{I}}$$

$$= f(x_k) + \langle \nabla f(x_k), x_{k+1} - x_k \rangle + \frac{1}{2} \|x_{k+1} - x_k\|^2_{\mathbf{C}(x_k)} + \frac{1}{2} \|x_{k+1} - x_k\|^2_{L_\mathbf{C} \mathbf{I}}$$

$$\stackrel{(17)}{\leq} f_\star + \frac{1}{2} \|x_{k+1} - x_k\|^2_{L_\mathbf{C} \mathbf{I}}$$

$$= f_\star + \frac{L_\mathbf{C} \|x_{k+1} - x_k\|^2}{2}.$$

By rearranging the above inequality and applying (21) from Lemma C.1, we get

$$\min_{0 \leq t \leq k-1} f(x_{t+1}) - f_\star \stackrel{(6)}{\leq} \min_{0 \leq t \leq k-1} \frac{L_\mathbf{C} \|x_{t+1} - x_t\|^2}{2}$$

$$\stackrel{(21)}{\leq} \frac{L_\mathbf{C} \|x_0 - x_\star\|^2}{2k}.$$

$\square$

**Remark.** For quadratic functions Assumption 2.1 is satisfied with $\mathbf{C}(x)$ equal to the Hessian, and $L_\mathbf{C} = 0$. Thus, LCD2 convergences in one step for this class of functions.

### C.3 CLOSED-FORM SOLUTIONS

In the main text, we argued that the update step of LCD2 has a closed-form solution in certain special cases. One interesting case is when $\mathbf{C}(x_k)$ is a rank one matrix. In the setting of minimizing the sum of squares of absolutely convex functions, we present a special rank one matrix and the corresponding update step of LCD2. For general rank one matrices, the update step is not as interpretable or insightful so we leave out the computation.

Let $f(x) = \sum_{i=1}^d \phi_i^2(x)$ where $\phi_i : \mathbb{R}^d \to \mathbb{R}$ is absolutely convex. Then $f$ satisfies inequality (5) with the following curvature mapping:

$$\mathbf{C}(y) = \frac{1}{2f(y)} \nabla f(y) \nabla f(y)^\top.$$

If we use the localization set defined by this curvature mapping, we can obtain a closed-form solution to the LCD2 update step. To simplify notation, let $g_k := \nabla f(x_k)$, $f_k := f(x_k)$, $\Delta_k := f(x_k) - f_\star$, $\mathbf{D} := \mathbf{D}(x_k)$ and $\mathbf{Q} = \mathbf{Q}(x_k)$. Consider the orthogonal decomposition of $\mathbf{C}(x_k)$:

$$\mathbf{D}(x_k) = \text{Diag}\left( \frac{g_k^\top g_k}{2f_k}, \, 0, \, \ldots, \, 0 \right) \qquad \mathbf{Q}(x_k) = \left[ \frac{g_k}{\|g_k\|_2} \quad \hat{g}_{k,1} \quad \ldots \quad \hat{g}_{k,d-1} \right],$$

where $\hat{g}_{k,1}, \ldots, \hat{g}_{k,d-1}$ are $d - 1$ orthogonal eigenvectors that are all also orthogonal to $g_k$.

From Appendix C, we know that to obtain a closed-form solution of LCD2, we must find the positive root of the following function:

$$H(\alpha) = \frac{\alpha^2}{2} g_k^\top \mathbf{Q} (\mathbf{I} + \alpha \mathbf{D})^{-1} \mathbf{D} (\mathbf{I} + \alpha \mathbf{D})^{-1} \mathbf{Q}^\top g_k - \alpha g_k^\top \mathbf{Q} (\mathbf{I} - \alpha \mathbf{D})^{-1} \mathbf{Q}^\top g_k + \Delta_k$$

$$= \alpha^2 \frac{f_k (g_k^\top g_k)^2}{(2f_k + \alpha g_k^\top g_k)^2} - \alpha \frac{2 f_k g_k^\top g_k}{2f_k + \alpha g_k^\top g_k} + \Delta_k.$$

The second equality comes from simplifying the matrix multiplications and observing that,

$$(\mathbf{I} + \alpha\mathbf{D})^{-1} = \text{Diag}\left(\frac{2f_k}{2f_k + \alpha g_k^\top g_k}, \; 0, \; \ldots, \; 0\right) \qquad \mathbf{Q}^\top g_k = [\|g_k\|_2 \quad 0 \quad \ldots \quad 0]^\top$$

Let $v = g_k^\top g_k$. Since $\mathbf{C}$ is a rank-one matrix, we can simplify $H$ and realize that it is a quadratic function of $\alpha$,

$$H(\alpha) = \frac{f_k v^2}{(2f_k + \alpha v)^2}\alpha^2 - \frac{2f_k v}{2f_k + \alpha v}\alpha + \Delta_k = 0.$$

Therefore, we will have at most two roots and there must exist a unique positive root that corresponds to the solution of the problem (the optimalLagrange multiplier). We can solve this quadratic to obtain an interpretable update step for LCD2. To start, we multiply the entire equation by $(2f_k + \alpha v)^2$ and simplify to get,

$$(v^2\Delta_k - f_k v^2)\alpha^2 + \alpha(4f_k v\alpha - 4f_k^2 v) + 4f_k^2\alpha = 0.$$

By observing that $\Delta_k = f_k - f_\star$ we can simplify the expression further,

$$f_\star v^2\alpha^2 + 4f_k f_\star v\alpha - 4f_k^2\Delta_k = 0.$$

Therefore,

$$\alpha = \frac{-4f_k f_\star v \pm \sqrt{(4f_k f_\star v)^2 + 4(f_\star v)(4f_k^2\Delta_k)}}{2f_\star v^2} = \frac{-2f_k f_\star \pm 2f_k\sqrt{f_k f_\star}}{f_\star v}.$$

To determine which root is positive we can rearrange the terms to see that,

$$\alpha = \frac{2f_k\sqrt{f_\star}(-\sqrt{f_\star} \pm \sqrt{f_k})}{f_\star v}.$$

For $\alpha$ to be positive we must select the positive sign. Now recall from Appendix C, that the update step of LCD2 is defined as follows,

$$\begin{aligned} x_{k+1} &= x_k - \alpha(\mathbf{I} + \alpha\mathbf{C})^{-1}g_k \\ &= x_k - \alpha\mathbf{Q}(\mathbf{I} + \alpha\mathbf{D})^{-1}\mathbf{Q}^\top g_k \\ &= x_k - \gamma_k g_k, \end{aligned}$$

where $\gamma_k = \alpha\frac{2f_k}{2f_k + \alpha v}$. We substitute

$$\alpha = \frac{-2f_k f_\star + 2f_k\sqrt{f_k f_\star}}{f_\star v}$$

into $\gamma_k$ to get

$$\gamma_k = \frac{2f_k\left(\frac{-2f_k f_\star + 2f_k\sqrt{f_k f_\star}}{f_\star v}\right)}{2f_k + v\left(\frac{-2f_k f_\star + 2f_k\sqrt{f_k f_\star}}{f_\star v}\right)} = \frac{2f_k\left(\frac{-2f_k f_\star + 2f_k\sqrt{f_k f_\star}}{f_\star v}\right)}{\frac{2f_k\sqrt{f_k f_\star}}{f_\star}} = \frac{2f_k - 2\sqrt{f_k f_\star}}{v}.$$

Therefore, we conclude that the update step has the following form:

$$x_{k+1} = x_k - \frac{2\left(f(x_k) - \sqrt{f(x_k)f_\star}\right)}{\|\nabla f(x_k)\|_2^2}\nabla f(x_k).$$

The update step of LCD2 differs from the classic Polyak stepsize in that we have $\sqrt{f(x_k)f_\star}$ instead of $f_\star$ and we multiply by 2.

**Remark.** In case $\mathbf{C}(x) = c\mathbf{I}$, for $c > 0$, LCD2 reduces to LCD3. Thus, the closed-form solution exists.

# D  LOCAL CURVATURE DESCENT 3 (LCD3)

## D.1  DERIVATION

Suppose the optimal value $f_\star$ is known. While the update step of LCD2 does not have a closed-form solution, the following update step does:

$$x_{k+1} = \arg \min_{x \in \mathcal{L}_{\mathbf{C}}(x_k)} \frac{1}{2} \|x - x_k\|^2_{\mathbf{C}(x_k)}, \tag{23}$$

where $\mathcal{L}_{\mathbf{C}}(x_k)$ is the same localization set defined in (17). Instead of using the $L_2$ norm, we can use the norm induced by $\mathbf{C}(x_k)$. Hence, this algorithm is referred to as LCD3. The benefit of using a different norm is that we can obtain a closed-form solution to this constrained optimization algorithm.

The Lagrangian of this problem is

$$\mathscr{L}(x, \alpha) := \frac{1}{2} \|x - x_k\|^2_{\mathbf{C}(x_k)} + \alpha \left( f(x_k) + \langle \nabla f(x_k), x - x_k \rangle + \frac{1}{2} \|x - x_k\|^2_{\mathbf{C}(x_k)} - f_\star \right).$$

If $\alpha$ is the optimal multiplier, then for optimal $\bar{x}$ we get $\nabla_x \mathscr{L}(\bar{x}, \bar{\alpha}) = 0$. The gradient is:

$$\nabla_x \mathscr{L}(\bar{x}, \bar{\alpha}) = \mathbf{C}(x_k)(\bar{x} - x_k) + \bar{\alpha} \left( \nabla f(x_k) + \mathbf{C}(x_k)(\bar{x} - x_k) \right) = 0.$$

Isolating $\bar{x}$, we get:

$$\bar{x} = x_k - \frac{\bar{\alpha}}{1 + \bar{\alpha}} \left[ \mathbf{C}(x_k) \right]^{-1} \nabla f(x_k). \tag{24}$$

Let $t := \frac{\bar{\alpha}}{1 + \bar{\alpha}}$. If $\bar{\alpha} = 0$, $x_k \in \mathcal{L}_{\mathbf{C}}(x_k)$, which means that the algorithm converged since $x_k \in \mathcal{L}_{\mathbf{C}}(x_k)$ if and only if $x_k \in \mathcal{X}_\star$. Then, for a generic update, we will have $\bar{\alpha} \neq 0$. Imposing $\nabla_\alpha \mathscr{L}(\bar{x}, \bar{\alpha}) = 0$, which means that the constraint must be tight:

$$f(x_k) + \langle \nabla f(x_k), \bar{x} - x_k \rangle + \frac{1}{2} \|x - x_k\|^2_{\mathbf{C}(x_k)} - f_\star = 0.$$

Plugging $\bar{x} \equiv \bar{x}(t) \equiv \bar{x}(\bar{\alpha})$ into the equation gives

$$-t \left\langle \nabla f(x_k), \left[ \mathbf{C}(x_k) \right]^{-1} \nabla f(x_k) \right\rangle + \frac{t^2}{2} \left\langle \nabla f(x_k), \left[ \mathbf{C}(x_k) \right]^{-1} \nabla f(x_k) \right\rangle = f_\star - f(x_k).$$

The two inner products are norms of the form $\|\nabla f(x_k)\|^2_{[\mathbf{C}(x_k)]^{-1}}$, and with more compact notation we can write:

$$t^2 - 2t + \frac{2(f(x_k) - f_\star)}{\|\nabla f(x_k)\|^2_{[\mathbf{C}(x_k)]^{-1}}} = 0.$$

This equation has two roots summing up to 2, but only one of them can be of the form $t = \frac{\bar{\alpha}}{1 + \bar{\alpha}}$ since only one of them can be smaller than 1, with expression:

$$t = 1 - \sqrt{1 - \frac{2(f(x_k) - f_\star)}{\|\nabla f(x_k)\|^2_{[\mathbf{C}(x_k)]^{-1}}}}. \tag{25}$$

Substituting back $t = \frac{\bar{\alpha}}{1 + \bar{\alpha}}$ into (24), where $t$ is given by (25), leads to the method

$$x_{k+1} = x_k - \left( 1 - \sqrt{1 - \frac{2(f(x_k) - f_\star)}{\|\nabla f(x_k)\|^2_{[\mathbf{C}(x_k)]^{-1}}}} \right) \left[ \mathbf{C}(x_k) \right]^{-1} \nabla f(x_k).$$

To realize that the scalar component of the stepsize is well-defined, it suffices to show that:

$$1 - \frac{2(f(x_k) - f_\star)}{\|\nabla f(x_k)\|^2_{[\mathbf{C}(x_k)]^{-1}}} \geq 0, \tag{26}$$

which follows by reordering the result of Lemma E.5 for the pair $(x_k, x_\star)$.

Then the update rule has a closed-form solution:

$$x_{k+1} = x_k - \gamma_k^{\mathsf{LCD3}} \left[\mathbf{C}(x_k)\right]^{-1} \nabla f(x_k), \quad \gamma_k^{\mathsf{LCD3}} := \left(1 - \sqrt{1 - \frac{2(f(x_k) - f_\star)}{\|\nabla f(x_k)\|_{\mathbf{C}_k^{-1}}^2}}\right). \tag{27}$$

In particular, the argument of the square root is always positive, making LCD3 well-defined. Routine (27) is promising: we apply a scalar stepsize $\gamma_k^{\mathsf{LCD3}}$ that is similar in spirit to Polyak's in Equation (4), and "reorient" the gradient according to $\mathbf{C}_k^{-1} = \left[\mathbf{C}(x_k)\right]^{-1}$.

Moreover, at each step, we aim to be as close as possible according to local upper-lower bounds on $f$. Experiments in section 8 and G, show that the algorithm converges, but is slower than LCD2.

### D.2 CONVERGENCE FOR QUADRATICS

Despite not converging in general, in special cases LCD3 reduces to Newton's method. Below, we show that the update rule (27) takes the form of Hessian times gradient.

Let $\phi_i(x) = |a_i^\top x - b_i|$, where $a_i \in \mathbb{R}^d$ and $b_i \in \mathbb{R}$, for $i \in \{1, \ldots, n\}$. We know from Example 7.1 that $\phi_i$ is absolutely convex. Then problem (9) becomes

$$\min_{x \in \mathbb{R}^d} \left\{ f(x) := \frac{1}{n} \sum_{i=1}^n \left(a_i^\top x - b_i\right)^2 \right\}.$$

If $x$ is such that $\phi_i(x) \neq 0$ for all $i$, then $\nabla\phi_i(x) = \frac{a_i^\top x - b_i}{\phi_i(x)} a_i$. Therefore, in view of the computation in Section 7 we get

$$\mathbf{C}(x) = \frac{2}{n} \sum_{i=1}^n \nabla\phi_i(x)\nabla\phi_i(x)^\top = \frac{2}{n} \sum_{i=1}^n \frac{a_i^\top x - b_i}{\phi_i(x)} a_i \left(\frac{a_i^\top x - b_i}{\phi_i(x)} a_i\right)^\top$$

$$= \frac{2}{n} \sum_{i=1}^n \frac{(a_i^\top x - b_i)^2}{\phi_i^2(x)} a_i a_i^\top = \frac{2}{n} \sum_{i=1}^n a_i a_i^\top = \nabla^2 f(x).$$

Therefore, for least-squares problems, the LCD3 method of (27) moves in Newton's direction. Furthermore, $\gamma_k^{\mathsf{LCD3}} = 1$ since for quadratics we have the identity

$$f(x_k) - f_\star = \frac{1}{2}\|\nabla f(x_k)\|_{[\mathbf{C}(x_k)]^{-1}}^2.$$

Indeed, this follows from Lemma E.5 and the fact that for quadratics, equation (5) is an identity. So, for least-squares problems, LCD3 reduces to Newton's method, and converges in one step.

## E    PROPERTIES & EXAMPLES

### E.1    ON THE LOWER BOUND

For clarity, the statements are repeated, but correspond to Lemma 5.1 and Corollary 5.1.

**Lemma E.1.** *Suppose* $f, f_1, f_2 : \mathbb{R}^d \to \mathbb{R}$ *satisfy Equation* (5) *with curvature mappings* $\mathbf{C}, \mathbf{C}_1, \mathbf{C}_2 :$ $\mathbb{R}^d \to$, *respectively. Then, the following functions satisfy Equation* (5):

*(1)* $f + \alpha$ *for* $\alpha \in \mathbb{R}$, *with* $\mathbf{C}(\cdot)$;

*(2)* $\alpha f$ *for* $\alpha \geq 0$, *with* $\alpha \mathbf{C}(\cdot)$;

*(3)* $f_1 + f_2$, *with* $\mathbf{C}_1(\cdot) + \mathbf{C}_2(\cdot)$.

*Proof.* We prove each statement separately.
(1) For any $x, y \in \mathbb{R}^d$, it holds:

$$g(x) := f(x) + \alpha \geq f(y) + \langle \nabla f(y), x - y \rangle + \frac{1}{2}\|x - y\|^2_{\mathbf{C}(y)} + \alpha$$

$$= g(y) + \langle \nabla g(y), x - y \rangle + \frac{1}{2}\|x - y\|^2_{\mathbf{C}(y)}.$$

(2) Similarly, for all $x, y \in \mathbb{R}^d$, one has:

$$g(x) := \alpha f(x) \geq \alpha \left( f(y) + \langle \nabla f(y), x - y \rangle + \frac{1}{2}\|x - y\|^2_{\mathbf{C}(y)} \right)$$

$$= \alpha f(y) + \langle \alpha \nabla f(y), x - y \rangle + \frac{1}{2}\|\alpha(x - y)\|^2_{\mathbf{C}(y)}$$

$$= g(y) + \langle \nabla g(y), x - y \rangle + \frac{1}{2}\|\alpha(x - y)\|^2_{\mathbf{C}(y)}.$$

(3) Concluding, for arbitrary vectors:

$$g(x) := f_1(x) + f_2(x) \geq f_1(y) + \langle \nabla f_1(y), x - y \rangle + \frac{1}{2}\|x - y\|^2_{\mathbf{C}_1(y)}$$

$$+ f_2(y) + \langle \nabla f_2(y), x - y \rangle + \frac{1}{2}\|x - y\|^2_{\mathbf{C}_2(y)}$$

$$= g(y) + \langle \nabla g(y), x - y \rangle + \frac{1}{2}\|x - y\|^2_{\mathbf{C}_1(y) + \mathbf{C}_2(y)}.$$

$\square$

**Corollary E.1.** *Suppose* $f : \mathbb{R}^d \to \mathbb{R}$ *satisfies the lower bound of Equation* (5) *with curvature mapping* $\mathbf{C} : \mathbb{R}^d \to$. *Let* $g : \mathbb{R}^d \to \mathbb{R}$ *be a convex function. Then,* $h(x) = f(x) + g(x)$ *satisfies the lower bound with matrix* $\mathbf{C}(y)$.

*Proof.* Since $g$ is convex it satisfies the lower bound with matrix $\mathbf{C}(y) \equiv \mathbf{0}$. By Lemma E.1, $h$ satisfies the lower bound with $\mathbf{C}(y) + \mathbf{0} = \mathbf{C}(y)$. $\square$

Another lemma used to construct functions that satisfy the lower bound is the following.

**Lemma E.2.** *Suppose* $f : \mathbb{R}^d \to \mathbb{R}$ *satisfies Equation* (5) *with the curvature mapping* $\mathbf{C} : \mathbb{R}^d \to$. *Let* $\mathbf{A} \in \mathbb{R}^{d \times m}$ *and* $b \in \mathbb{R}^d$. *Then* $g : \mathbb{R}^m \to \mathbb{R}$ *where* $g(x) := f(\mathbf{A}x + b)$ *satisfies Equation* (5) *with curvature mapping* $\tilde{\mathbf{C}}(y) = \mathbf{A}^\top \mathbf{C}(\mathbf{A}y + b)\mathbf{A}$.

*Proof.* Without loss of generality, we can assume that $\mathbf{C}(y)$ is symmetric. Then,

$$g(x) := f(\mathbf{A}x + b) \geq f(\mathbf{A}y + b) + \langle \nabla f(\mathbf{A}y + b), \mathbf{A}x + b - (\mathbf{A}y + b) \rangle$$

$$+ \frac{1}{2} \langle \mathbf{C}(\mathbf{A}y + b)(\mathbf{A}x + b - (\mathbf{A}y + b)), \mathbf{A}x + b - (\mathbf{A}y + b) \rangle$$

$$= f(\mathbf{A}y + b) + \langle \nabla f(\mathbf{A}y + b), \mathbf{A}(x - y) \rangle$$

$$+ \frac{1}{2} \langle \mathbf{C}(\mathbf{A}y + b)(\mathbf{A}(x - y)), \mathbf{A}(x - y) \rangle$$

$$= f(\mathbf{A}y + b) + \langle \mathbf{A}^\top \nabla f(\mathbf{A}y + b), x - y \rangle$$

$$+ \frac{1}{2} \langle \mathbf{A}^\top \mathbf{C}(\mathbf{A}y + b)\mathbf{A}(x - y), x - y \rangle$$

$$= g(y) + \langle \nabla g(y), x - y \rangle + \frac{1}{2} \langle \tilde{\mathbf{C}}(y)(x - y), x - y \rangle .$$

Considering the right hand side and the left hand side, we have recovered the claimed expression. The only missing detail is proving that $\tilde{\mathbf{C}}(\cdot)$ is positive semi-definite. Let $z \in \mathbb{R}^m$. Since $\mathbf{C}(y)$ is symmetric and positive semi-definite then $\mathbf{C}(y) = \mathbf{C}(y)^{\frac{1}{2}} \mathbf{C}(y)^{\frac{1}{2}}$ and $\mathbf{C}(y)^{\frac{1}{2}}$ is also symmetric. Therefore:

$$z^\top \tilde{\mathbf{C}}(y) z = z^\top \mathbf{A}^\top \mathbf{C}(\mathbf{A}y + b) \mathbf{A}z$$

$$= z^\top \mathbf{A}^\top \mathbf{C}(\mathbf{A}y + b)^{\frac{1}{2}} \mathbf{C}(\mathbf{A}y + b)^{\frac{1}{2}} \mathbf{A}z$$

$$= ((\mathbf{C}(\mathbf{A}y + b)^{\frac{1}{2}})^\top \mathbf{A}z)^\top (\mathbf{C}(\mathbf{A}y + b)^{\frac{1}{2}} \mathbf{A}z)$$

$$= (\mathbf{C}(\mathbf{A}y + b)^{\frac{1}{2}} \mathbf{A}z)^\top (\mathbf{C}(\mathbf{A}y + b)^{\frac{1}{2}} \mathbf{A}z)$$

$$= \left\| \mathbf{C}(\mathbf{A}y + b)^{\frac{1}{2}} \mathbf{A}z \right\|^2 \geq 0.$$

By the arbitrariness of $z$, the matrix is positive semi-definite. $\qquad\square$

**Lemma E.3.** *Suppose $f : \mathbb{R}^d \to \mathbb{R}$ satisfies Equation (5) with curvature mapping $\mathbf{C} : \mathbb{R}^d \to$. Then the following inequalities hold for any $x, y \in \mathbb{R}^d$,*

*(1)* $\langle \nabla f(y) - \nabla f(x), x - y \rangle \geq \frac{1}{2} \| x - y \|^2_{\mathbf{C}(y) + \mathbf{C}(x)}$

*(2)* $\langle \nabla f(y) - \nabla f(x), x - y \rangle \geq \frac{1}{2} \| x - y \|^2_{\mathbf{C}(y)}$

*(3)* $\langle \nabla f(y) - \nabla f(x), x - y \rangle \geq \frac{1}{2} \| x - y \|^2_{\mathbf{C}(x)}$

*Proof.* Let us present one proof in detail. The other two follow trivially.
(1) By the definition of Bregman divergence:

$$\frac{1}{2} \| x - y \|^2_{\mathbf{C}(x) + \mathbf{C}(y)} = \frac{1}{2} \| x - y \|^2_{\mathbf{C}(y)} + \frac{1}{2} \| y - x \|^2_{\mathbf{C}(x)}$$

$$\leq D_f(x, y) + D_f(y, x)$$

$$= \langle \nabla f(x) - \nabla f(y), x - y \rangle .$$

(2) Start with (1) and note that $\frac{1}{2} \| x - y \|^2_{\mathbf{C}(x) + \mathbf{C}(y)} \geq \frac{1}{2} \| x - y \|^2_{\mathbf{C}(y)}$.
(2) Same as above but use that $\frac{1}{2} \| x - y \|^2_{\mathbf{C}(x) + \mathbf{C}(y)} \geq \frac{1}{2} \| x - y \|^2_{\mathbf{C}(x)}$. $\qquad\square$

**Lemma E.4.** *Suppose $f : \mathbb{R}^d \to \mathbb{R}$ satisfies*

$$\| x - y \|^2_{\mathbf{C}(y)} \leq \langle \nabla f(x) - \nabla f(y), x - y \rangle$$

*for all $x, y \in \mathbb{R}^d$ with curvature mapping $\mathbf{C} : \mathbb{R}^d \to$. Then $f$ satisfies Equation (5) with curvature mapping $\mathbf{C}(\cdot)$.*

*Proof.* By the fundamental theorem of calculus:

$$f(x) - f(y) = \int_0^1 \langle \nabla f(y + t(x - y)), x - y \rangle \, dt$$

$$= \langle \nabla f(y), x - y \rangle + \int_0^1 \langle \nabla f(y + t(x - y)) - \nabla f(y), x - y \rangle \, dt$$

$$= \langle \nabla f(y), x - y \rangle + \int_0^1 \frac{1}{t} \langle \nabla f(y + t(x - y)) - \nabla f(y), t(x - y) \rangle \, dt$$

$$\geq \langle \nabla f(y), x - y \rangle + \int_0^1 \frac{1}{t} \| t(x - y) \|^2_{\mathbf{C}(y)} dt$$

$$= \langle \nabla f(y), x - y \rangle + \| x - y \|^2_{\mathbf{C}(y)} \int_0^1 t \, dt$$

$$= \langle \nabla f(y), x - y \rangle + \frac{1}{2} \| x - y \|^2_{\mathbf{C}(y)}.$$

Rearranging the terms we obtain that $D_f(x, y) \geq \frac{1}{2} \| x - y \|^2_{\mathbf{C}(y)}$ as desired. $\square$

**Lemma E.5.** *Suppose that $f : \mathbb{R}^d \to \mathbb{R}$ satisfies Equation (5) with curvature mapping $\mathbf{C} : \mathbb{R}^d \to$. Suppose that $f$ is differentiable and $\mathbf{C}$ is non-singular. Then*

$$\frac{1}{2} \| \nabla f(x) - \nabla f(y) \|^2_{\mathbf{C}(x)^{-1}} \geq D_f(x, y), \qquad \forall x, y \in \mathbb{R}^d.$$

*Proof.* Fix $x \in \mathbb{R}^d$. Suppose $y \in \mathbb{R}^d$ is arbitrary. Let $\varphi(y) := f(y) - \langle \nabla f(x), y \rangle$. By construction, $\nabla \varphi(y) = \nabla f(y) - \nabla f(x)$. Using this fact, it can be shown that for any $u, v \in \mathbb{R}^d$, $D_f(u, v) \geq \frac{1}{2} \| u - v \|^2_{\mathbf{C}(v)}$. Therefore, $\varphi$ satisfies Equation (5) with curvature mapping $\mathbf{C}$. Hence, for $v \in \mathbb{R}^d$ we have that $\varphi(y) \geq G(y)$ where $G(y)$ is defined as

$$G(y) := \varphi(v) + \langle \nabla \varphi(v), y - v \rangle + \frac{1}{2} \| y - v \|^2_{\mathbf{C}(v)}.$$

Observe that $\nabla \varphi(x) = 0$. Since $\varphi$ is convex, $x$ is a minimizer of $\varphi$, and we the inequality below holds:

$$\varphi(x) = \inf_y \varphi(y) \geq \inf_y G(y). \tag{28}$$

By computing the gradient of $G$ and setting it to zero, we find $\overline{y} = -\mathbf{C}(v)^{-1} \nabla \varphi(v) + v$ such that $\nabla G(\overline{y}) = 0$. Therefore,

$$f(x) - \langle \nabla f(x), x \rangle = \varphi(x) \geq G(\overline{y})$$

$$= \varphi(v) - \langle \nabla \varphi(v), \mathbf{C}(v)^{-1} \nabla \varphi(v) \rangle + \frac{1}{2} \| \mathbf{C}(v)^{-1} \nabla \varphi(v) \|^2_{\mathbf{C}(v)}$$

$$= \varphi(v) - \| \nabla \varphi(v) \|_{\mathbf{C}^{-1}(v)} + \frac{1}{2} \| \nabla \varphi(v) \|_{\mathbf{C}^{-1}(v)}$$

$$= f(v) - \langle \nabla f(x), v \rangle - \frac{1}{2} \| \nabla \varphi(v) \|_{\mathbf{C}^{-1}(v)}$$

$$= f(v) - \langle \nabla f(x), v \rangle - \frac{1}{2} \| \nabla f(v) - \nabla f(x) \|_{\mathbf{C}^{-1}(v)}.$$

By rearranging the terms, we find our result since $x$ and $v$ are arbitrary:

$$\frac{1}{2} \| \nabla f(v) - \nabla f(x) \|_{\mathbf{C}^{-1}(v)} \geq f(v) - f(x) + \langle \nabla f(x), x \rangle - \langle \nabla f(x), v \rangle$$

$$= f(v) - f(x) + \langle \nabla f(x), x - v \rangle$$

$$= f(v) - f(x) - \langle \nabla f(x), v - x \rangle$$

$$= D_f(v, x).$$

$\square$

**Lemma E.6.** *Suppose that $f : \mathbb{R}^d \to \mathbb{R}$ satisfies Equation (5) with curvature mapping $\mathbf{C} : \mathbb{R}^d \to$. If $f$ is twice continuously differentiable, then*

$$\mathbf{C}(x) \preceq \nabla^2 f(x),$$

*for all $x \in \mathbb{R}^d$.*

*Proof.* Let $x, y' \in \mathbb{R}^d$ and $\lambda > 0$. Since $f$ satisfies Equation (5) we can substitute $x + \lambda(y' - x)$ and $\lambda(y' - x)$ into the first inequality described in Lemma E.3, to find:

$$\langle \nabla f(x + \lambda(y' - x)) - \nabla f(x), \lambda(y' - x) \rangle \geq \frac{1}{2}\|\lambda(y' - x)\|^2_{\mathbf{C}(x) + \mathbf{C}(x + \lambda(y' - x))}$$

$$= \frac{\lambda^2}{2}\|y' - x\|^2_{\mathbf{C}(x) + \mathbf{C}(x + \lambda(y' - x))}.$$

By the Fundamental Theorem of Calculus, we further have that:

$$\langle \nabla f(x + \lambda(y' - x)) - \nabla f(x), y' - x \rangle = \int_0^1 \langle \nabla^2 f(x + t\lambda(y' - x))(\lambda(y' - x)), y' - x \rangle \, dt.$$

Dividing the first inequality by $\lambda^2$ on both sides we obtain an intermediate inequality:

$$\frac{1}{2}\|y' - x\|_{\mathbf{C}(x) + \mathbf{C}(x + \lambda(y' - x))} \leq \frac{1}{\lambda} \langle \nabla f(x + \lambda(y' - x)) - \nabla f(x), y' - x \rangle$$

$$= \frac{1}{\lambda} \int_0^1 \langle \nabla^2 f(x + t\lambda(y' - x))(\lambda(y' - x)), y' - x \rangle \, dt$$

$$= \int_0^1 \langle \nabla^2 f(x + t\lambda(y' - x))(y' - x), y' - x \rangle \, dt,$$

from which we take $\lambda \to 0$ of both sides to get an inequality between norms,

$$\frac{1}{2}\|y' - x\|^2_{2\mathbf{C}(x)} \leq \int_0^1 \langle \nabla^2 f(x)(y' - x), y' - x \rangle \, dt$$

$$= \langle \nabla^2 f(x)(y' - x), y' - x \rangle.$$

Thus, $\|y' - x\|^2_{\mathbf{C}(x)} \leq \langle \nabla^2 f(x)(y' - x), y' - x \rangle$. Since $x, y'$ are arbitrary this implies that $\mathbf{C}(x) \preceq \nabla^2 f(x)$. □

### E.2 ON THE UPPER BOUND

We provide analogous lemmas involving functions that satisfy Equation (6).

**Lemma E.7.** *Suppose $f : \mathbb{R}^d \to \mathbb{R}$ satisfies Equation (6). Then for all $x, y \in \mathbb{R}^d$ we have,*

$$\langle \nabla f(y) - \nabla f(x), y - x \rangle \leq \frac{1}{2}\|x - y\|^2_{\mathbf{C}(x) + \mathbf{C}(y)} + L_{\mathbf{C}}\|x - y\|^2.$$

*Proof.* Take the sum of the two Bregmann divergences:

$$\langle \nabla f(y) - \nabla f(x), y - x \rangle = D_f(x, y) + D_f(y, x)$$

$$\leq \frac{1}{2}\|x - y\|^2_{\mathbf{C}(x) + L_{\mathbf{C}}\mathbf{I}} + \frac{1}{2}\|x - y\|^2_{\mathbf{C}(y) + L_{\mathbf{C}}\mathbf{I}}$$

$$= \frac{1}{2}\|x - y\|^2_{\mathbf{C}(x) + \mathbf{C}(y)} + L_{\mathbf{C}}\|x - y\|^2.$$

□

**Lemma E.8.** *Suppose $f : \mathbb{R}^d \to \mathbb{R}$ satisfies the following inequality with constant $L_{\mathbf{C}} > 0$ and curvature mapping $\mathbf{C} : \mathbb{R}^d \to$,*

$$\langle \nabla f(x) - \nabla f(y), x - y \rangle \leq \|x - y\|^2_{\mathbf{C}(y) + L_{\mathbf{C}}\mathbf{I}} \qquad \forall x, y \in \mathbb{R}^d.$$

*Then $f$ satisfies Equation (6) with curvature mapping $\mathbf{C}$ and constant $L_{\mathbf{C}}$.*

*Proof.* We invoke the Fundamental Theorem of Calculus:

$$f(x) - f(y) = \int_0^1 \langle \nabla f(y + t(x - y)), x - y \rangle \, \mathrm{d}t$$

$$= \langle \nabla f(y), x - y \rangle + \int_0^1 \langle \nabla f(y + t(x - y)) - \nabla f(y), x - y \rangle \, \mathrm{d}t$$

$$= \langle \nabla f(y), x - y \rangle + \int_0^1 \frac{1}{t} \langle \nabla f(y + t(x - y)) - \nabla f(y), t(x - y) \rangle \, \mathrm{d}t$$

$$\leq \langle \nabla f(y), x - y \rangle + \int_0^1 \frac{1}{t} \| t(x - y) \|_{\mathbf{C}(y) + L_{\mathbf{C}}\mathbf{I}}^2 \mathrm{d}t$$

$$= \langle \nabla f(y), x - y \rangle + \| x - y \|_{\mathbf{C}(y) + L_{\mathbf{C}}\mathbf{I}}^2 \int_0^1 t \, \mathrm{d}t$$

$$= \langle \nabla f(y), x - y \rangle + \frac{1}{2} \| x - y \|_{\mathbf{C}(y) + L_{\mathbf{C}}\mathbf{I}}^2.$$

Rearranging the inequality above we get our result, $D_f(x, y) \leq \frac{1}{2} \| x - y \|_{\mathbf{C}(y) + L\mathbf{I}}^2$. $\qquad \square$

**Lemma E.9.** *Suppose that $f : \mathbb{R}^d \to \mathbb{R}$ is convex and satisfies Equation (6). Also, assume that $f$ is differentiable. Then,*

$$\frac{1}{2} \| \nabla f(x) - \nabla f(y) \|_{(\mathbf{C}(x) + L_{\mathbf{C}}\mathbf{I})^{-1}}^2 \leq D_f(x, y)$$

*Proof.* Fix $x \in \mathbb{R}^d$. Suppose $y \in \mathbb{R}^d$. Let $\varphi(y) := f(y) - \langle \nabla f(x), y \rangle$. By construction, $\nabla \varphi(y) = \nabla f(y) - \nabla f(x)$.

Using the above fact, we can show that $\varphi$ is convex and that for any $u, v \in \mathbb{R}^d$, we have $D_\varphi(u, v) \leq \frac{1}{2} \| u - v \|_{\mathbf{C}(v) + L_{\mathbf{C}}\mathbf{I}}^2$. Therefore $\varphi$ satisfies Equation (6). Now, let $v \in \mathbb{R}^d$ be arbitrary. Since $\varphi$ satisfies Equation (6), $\varphi(y) \leq G(y)$ where

$$G(y) := \varphi(v) + \langle \nabla \varphi(v), y - v \rangle + \frac{1}{2} \| y - v \|_{\mathbf{C}(v) + L_{\mathbf{C}}\mathbf{I}}^2.$$

Moreover, $x$ is a minimizer of $\varphi$ because $\nabla \varphi(x) = 0$ and $\varphi$ is convex. Combining the last two facts,

$$\varphi(x) = \inf_y \varphi(y) \geq \inf_y G(y).$$

We minimize $G$ with respect to $y$ by finding a $\bar{y} \in \mathbb{R}^d$ such that $\nabla G(\bar{y}) = 0$. Since $\mathbf{C}(v)$ is positive semi-definite, $\mathbf{C}(v) + L_{\mathbf{C}}\mathbf{I}$ is non-singular. Then $\bar{y} = v - (\mathbf{C}(v) + L_{\mathbf{C}}\mathbf{I})^{-1} \varphi(v)$. Therefore,

$$f(x) - \langle \nabla f(x), x \rangle = \varphi(x) \leq G(\bar{y})$$

$$= \varphi(v) - \langle \nabla \varphi(v), (\mathbf{C}(v) + L_{\mathbf{C}}\mathbf{I})^{-1} \nabla \varphi(v) \rangle$$

$$+ \frac{1}{2} \| (\mathbf{C}(v) + L_{\mathbf{C}}\mathbf{I})^{-1} \nabla \varphi(v) \|_{\mathbf{C}(v) + L_{\mathbf{C}}\mathbf{I}}^2$$

$$= \varphi(v) - \| \nabla \varphi(v) \|_{(\mathbf{C}(v) + L_{\mathbf{C}}\mathbf{I})^{-1}}^2 + \frac{1}{2} \| \nabla \varphi(v) \|_{(\mathbf{C}(v) + L_{\mathbf{C}}\mathbf{I})^{-1}}^2$$

$$= \varphi(v) - \frac{1}{2} \| \nabla \varphi(v) \|_{(\mathbf{C}(v) + L_{\mathbf{C}}\mathbf{I})^{-1}}^2$$

$$= f(v) - \langle \nabla f(x), v \rangle - \frac{1}{2} \| \nabla f(v) - \nabla f(x) \|_{(\mathbf{C}(v) + L_{\mathbf{C}}\mathbf{I})^{-1}}^2.$$

Rearranging the terms we obtain

$$\frac{1}{2} \| \nabla f(v) - \nabla f(x) \|_{(\mathbf{C}(v) + L_{\mathbf{C}}\mathbf{I})^{-1}}^2 \leq f(v) - f(x) - \langle \nabla f(x), v \rangle + \langle \nabla f(x), x \rangle$$

$$= f(v) - f(x) - \langle \nabla f(x), v \rangle - \langle \nabla f(x), -x \rangle$$

$$= f(v) - f(x) - \langle \nabla f(x), v - x \rangle$$

$$= D_f(v, x).$$

Since $v, x$ were arbitrary, the claim is true. $\qquad \square$

**Lemma E.10.** *Suppose that $f : \mathbb{R}^d \to \mathbb{R}$ is convex and satisfies Equation* (6). *Also, assume that $f$ is twice differentiable. Then,*

$$\nabla^2 f(x) \preceq \mathbf{C}(x) + L\mathbf{I}.$$

*Proof.* Suppose $x, y' \in \mathbb{R}^d$ and $\lambda > 0$. Since $f$ satisfies Equation (6), we can substitute $x + \lambda(y' - x)$ and $\lambda(y' - x)$ into Lemma E.7,

$$\langle \nabla f(x + \lambda(y' - x)) - \nabla f(x), \lambda(y' - x) \rangle \leq \frac{1}{2} \|\lambda(y' - x)\|^2_{\mathbf{C}(x) + \mathbf{C}(x + \lambda(y' - x)) + 2L_\mathbf{C}\mathbf{I}}$$

$$= \frac{\lambda^2}{2} \|y' - x\|^2_{\mathbf{C}(x) + \mathbf{C}(x + \lambda(y' - x)) + 2L_\mathbf{C}\mathbf{I}}.$$

The following equality is a direct application of the fundamental theorem of calculus:

$$\langle \nabla f(x + \lambda(y' - x)) - \nabla f(x), y' - x \rangle = \int_0^1 \left\langle \nabla^2 f(x + t\lambda(y' - x))(\lambda(y' - x)), y' - x \right\rangle dt$$

$$= \int_0^1 \lambda \left\langle \nabla^2 f(x + t\lambda(y' - x))(y' - x), y' - x \right\rangle dt$$

Dividing the inequality by $\lambda^2$ on both sides:

$$\frac{1}{2} \|y' - x\|^2_{\mathbf{C}(x) + \mathbf{C}(x + \lambda(y' - x)) + 2L_\mathbf{C}\mathbf{I}} \geq \frac{1}{\lambda^2} \langle \nabla f(x + \lambda(y' - x)) - \nabla f(x), \lambda(y' - x) \rangle$$

$$= \frac{1}{\lambda} \langle \nabla f(x + \lambda(y' - x)) - \nabla f(x), y' - x \rangle$$

$$= \frac{1}{\lambda} \int_0^1 \lambda \left\langle \nabla^2 f(x + t\lambda(y' - x))(y' - x), y' - x \right\rangle dt$$

$$= \int_0^1 \left\langle \nabla^2 f(x + t\lambda(y' - x))(y' - x), y' - x \right\rangle dt.$$

It suffices to take limits $\lambda \to 0$ to get that:

$$\frac{1}{2} \|y' - x\|^2_{\mathbf{C}(x) + \mathbf{C}(x) + 2L_\mathbf{C}\mathbf{I}} \geq \int_0^1 \left\langle \nabla^2 f(x)(y' - x), y' - x \right\rangle dt$$

$$= \left\langle \nabla^2 f(x)(y' - x), y' - x \right\rangle,$$

allowing us to conclude with:

$$\frac{1}{2} \|y' - x\|^2_{\mathbf{C}(x) + \mathbf{C}(x) + 2L_\mathbf{C}\mathbf{I}} = \frac{1}{2} \|y' - x\|^2_{2\mathbf{C}(x) + 2L_\mathbf{C}\mathbf{I}}$$

$$= \langle (\mathbf{C}(x) + L_\mathbf{C}\mathbf{I})(y' - x), y' - x \rangle$$

$$\geq \left\langle \nabla^2 f(x)(y' - x), y' - x \right\rangle.$$

Since $y', x \in \mathbb{R}^d$ are arbitrary, we proved the claim: $\nabla^2 f(x) \preceq \mathbf{C}(x) + L_\mathbf{C}\mathbf{I}$. $\qquad\square$

### E.3 LOWER BOUND EXAMPLES

**Lemma E.11.** *Suppose $p \geq 2$. Let $f : \mathbb{R}^d \to \mathbb{R}$ where $f(x) = \|x\|^p_p$. Then $f$ satisfies Equation* (5) *in Assumption 2.1 with curvature mapping*

$$\mathbf{C}(y) = p \operatorname{Diag}\left(|y_1|^{p-2}, \ldots, |y_d|^{p-2}\right) = \frac{1}{p-1} \nabla^2 f(y).$$

*Proof.* When $p = 2$, we have that $\mathbf{C}(y) = 2\mathbf{I}$. Then $f$ satisfies Equation (5) because $\|x\|^2$ is 2-strongly-convex.
Now suppose $p > 2$. For arbitrary $x, y \in \mathbb{R}^d$, an application of Young's Inequality yields

$$\frac{(\|x\|^2_p)^{\frac{p}{2}}}{\frac{p}{2}} + \frac{(\|y\|^{p-2}_p)^{\frac{p}{p-2}}}{\frac{p}{p-2}} \geq \|x\|^2_p \|y\|^{p-2}_p.$$

Rearranging, we obtain:

$$\|x\|_p^p - \frac{p}{2}\|x\|_p^2 \|y\|_p^{p-2} + \left(\frac{p}{2} - 1\right)\|y\|_p^p \geq 0. \tag{29}$$

By applying Hölder's inequality, we get:

$$\sum_{i=1}^d |x_i|^2 |y_i|^{p-2} \leq \|x\|_p^2 \|y\|_p^{p-2}, \tag{30}$$

and thus,

$$-\frac{p}{2}\|x\|_p^2 \|y\|_p^{p-2} \leq -\frac{p}{2}\sum_{i=1}^d |x_i|^2 |y_i|^{p-2}.$$

By adding $\|x\|_p^p + \left(\frac{p}{2} - 1\right)\|y\|_p^p$ to both sides of Equation (30) and using Equation (29) we get,

$$\|x\|_p^p - \frac{p}{2}\sum_{i=1}^d |x_i|^2 |y_i|^{p-2} + \left(\frac{p}{2} - 1\right)\|y\|_p^p \geq 0.$$

To derive the result, we begin by rearranging the above inequality:

$$\|x\|_p^p \geq \frac{p}{2}\sum_{i=1}^d |x_i|^2 |y_i|^{p-2} - \left(\frac{p}{2} - 1\right)\|y\|_p^p$$

$$= \|y\|_p^p - p\|y\|_p^p + \frac{p}{2}\|y\|_p^p + \frac{p}{2}\sum_{i=1}^d |x_i|^2 |y_i|^{p-2}$$

$$= \|y\|_p^p - p\|y\|_p^p + p\sum_{i=1}^d y_i |y_i|^{p-2} x_i - p\sum_{i=1}^d y_i |y_i|^{p-2} x_i + \frac{p}{2}\|y\|_p^p + \frac{p}{2}\sum_{i=1}^d |x_i|^2 |y_i|^{p-2}.$$

After reordering, we find:

$$\|x\|_p^p \geq \|y\|_p^p + p\sum_{i=1}^d y_i |y_i|^{p-2} x_i - p\sum_{i=1}^d y_i^2 |y_i|^{p-2}$$

$$- p\sum_{i=1}^d y_i |y_i|^{p-2} x_i + \frac{p}{2}\|y\|_p^p + \frac{p}{2}\sum_{i=1}^d |x_i|^2 |y_i|^{p-2}.$$

By performing some basic algebra and observing that $\frac{\partial f}{\partial y_i} = p y_i |y_i|^{p-2}$, we obtain our result:

$$\|x\|_p^p = \|y\|_p^p + \sum_{i=1}^d p y_i |y_i|^{p-2} (x_i - y_i) - p\sum_{i=1}^d y_i |y_i|^{p-2} x_i + \frac{p}{2}\|y\|_p^p + \frac{p}{2}\sum_{i=1}^d |x_i|^2 |y_i|^{p-2}$$

$$= \|y\|_p^p + \sum_{i=1}^d p y_i |y_i|^{p-2} (x_i - y_i) + \frac{p}{2}\sum_{i=1}^d |y_i|^p - p\sum_{i=1}^d y_i |y_i|^{p-2} x_i + \frac{p}{2}\sum_{i=1}^d |x_i|^2 |y_i|^{p-2}$$

$$= \|y\|_p^p + \sum_{i=1}^d p y_i |y_i|^{p-2} (x_i - y_i) + \frac{1}{2}\langle \mathbf{C}(y)y, y\rangle - \langle \mathbf{C}(y)y, x\rangle + \frac{1}{2}\langle \mathbf{C}(y)x, x\rangle$$

$$= \|y\|_p^p + \sum_{i=1}^d p y_i |y_i|^{p-2} (x_i - y_i) + \frac{1}{2}\langle \mathbf{C}(y)(x - y), (x - y)\rangle$$

$$= \|y\|_p^p + \langle \nabla f(y), x - y\rangle + \frac{1}{2}\|x - y\|_{\mathbf{C}(y)}^2.$$

$\square$

**Lemma E.12.** *Suppose $p \geq 2$. Let $f : \mathbb{R}^d \to \mathbb{R}$ where $f(x) = \|x\|_p^p$. Then $f$ satisfies Equation (5) in Assumption 2.1 with curvature mapping $\mathbf{C}(y)$ were the $(i, j)^{th}$ entry of $\mathbf{C}(y)$ is*

$$\mathbf{C}_{i,j}(y) = \frac{p}{\|y\|_p^p} y_i y_j |y_i|^{p-2} |y_j|^{p-2},$$

*or alternatively, in matrix form:*

$$\mathbf{C}(y) = \frac{1}{p\|y\|_p^p}\nabla f(y)\nabla f(y)^\top.$$

*Proof.* When $p = 2$ we get that $\mathbf{C}(y) = \frac{2}{\|y\|^2}yy^\top$. Since $\|x\|^2$ is the square of $\|x\|$ which is absolutely convex, $f(x) = \|x\|^2$ satisfies Equation (5) because the curvature mapping $\mathbf{C}$ corresponds to the mapping obtained from absolute convexity.

Now suppose $p > 2$. Again by Hölder's Inequality, we have that

$$\sum_{i=1}^d |x_i|\,|y_i|^{p-1} \leq \|x\|_p\|y\|_p^{p-1}. \tag{31}$$

We can lower bound the left-hand side in the following manner:

$$\sum_{i=1}^d |x_i|\,|y_i|^{p-1} = \sum_{i=1}^d \left|x_i y_i^{p-1}\right| = \sum_{i=1}^d \left|x_i y_i y_i^{p-2}\right| = \sum_{i=1}^d \left|x_i y_i\,|y_i|^{p-2}\right| \geq \left|\sum_{i=1}^d x_i y_i\,|y_i|^{p-2}\right|.$$

Combining this inequality with Inequality (31) and squaring both sides we get,

$$\|x\|_p^2\|y\|_p^{2p-2} \geq \left(\sum_{i=1}^d |x_i|\,|y_i|^{p-1}\right)^2 \geq \left(\left|\sum_{i=1}^d x_i y_i\,|y_i|^{p-2}\right|\right)^2 = \left(\sum_{i=1}^d x_i y_i\,|y_i|^{p-2}\right)^2.$$

Then we multiply both sides by $-\frac{p}{2}$,

$$-\frac{p}{2}\|x\|_p^2\|y\|_p^{2p-2} \leq -\frac{p}{2}\left(\sum_{i=1}^d x_i y_i\,|y_i|^{p-2}\right)^2. \tag{32}$$

From Lemma E.11, we know an application of Young's Inequality with some rearranging yields the following:

$$\|x\|_p^p + \left(\frac{p}{2}-1\right)\|y\|_p^p - \frac{p}{2}\|x\|_p^2\|y\|_p^{p-2} \geq 0.$$

Now multiply both sides by $\|y\|_p^p$,

$$\|x\|_p^p\|y\|_p^p + \left(\frac{p}{2}-1\right)\|y\|_p^{2p} - \frac{p}{2}\|x\|_p^2\|y\|_p^{2p-2} \geq 0. \tag{33}$$

Adding $\|x\|_p^p\|y\|_p^p + \left(\frac{p}{2}-1\right)\|y\|_p^{2p}$ to both sides of Equation (32) and together with Equation (33) we have that,

$$\|x\|_p^p\|y\|_p^p + \left(\frac{p}{2}-1\right)\|y\|_p^{2p} - \frac{p}{2}(\sum_{i=1}^d x_i y_i\,|y_i|^{p-2})^2 \geq 0.$$

Rearranging this inequality and proceeding with the following steps we obtain the claim.

$$\|x\|_p^p \geq \left(1-\frac{p}{2}\right)\|y\|_p^p + \frac{p}{2\|y\|_p^p}\left(\sum_{i=1}^d x_i y_i\,|y_i|^{p-2}\right)^2$$

$$= \|y\|_p^p - p\|y\|_p^p + \frac{p}{2}\|y\|_p^p + \frac{p}{2\|y\|_p^p}\left(\sum_{i=1}^d x_i y_i\,|y_i|^{p-2}\right)\left(\sum_{j=1}^d x_j y_j\,|y_j|^{p-2}\right)$$

$$= \|y\|_p^p - p\|y\|_p^p + \frac{p}{2}\|y\|_p^p + \frac{p}{2\|y\|_p^p}\sum_{i=1}^d\sum_{j=1}^d x_i y_i\,|y_i|^{p-2}\,x_j y_j\,|y_j|^{p-2}$$

$$= \|y\|_p^p - p\|y\|_p^p + \frac{p}{2}\|y\|_p^p + \frac{p}{2\|y\|_p^p}\sum_{i=1}^d\sum_{j=1}^d x_i y_i y_j\,|y_i|^{p-2}\,|y_j|^{p-2}\,x_j$$

$$= \|y\|_p^p - p\|y\|_p^p + \frac{p}{2\|y\|_p^p}\|y\|_p^{2p} + \frac{1}{2}\left\langle\mathbf{C}(y)x, x\right\rangle,$$

The last term seems complicated but can be expressed as a matrix inner product. Continuing,

$$\|x\|_p^p \geq \|y\|_p^p - p\|y\|_p^p + \frac{p}{2\|y\|_p^p}\left(\sum_{i=1}^d |y_i|^p\right)\left(\sum_{j=1}^d |y_j|^p\right) + \frac{1}{2}\langle \mathbf{C}(y)x, x\rangle$$

$$= \|y\|_p^p - p\|y\|_p^p + \frac{p}{2\|y\|_p^p}\left(\sum_{i=1}^d\sum_{j=1}^d |y_i|^p |y_j|^p\right) + \frac{1}{2}\langle \mathbf{C}(y)x, x\rangle$$

$$= \|y\|_p^p - p\|y\|_p^p + \frac{p}{2\|y\|_p^p}\left(\sum_{i=1}^d\sum_{j=1}^d y_i y_i y_j |y_i|^{p-2} |y_j|^{p-2} y_j\right) + \frac{1}{2}\langle \mathbf{C}(y)x, x\rangle$$

$$= \|y\|_p^p - p\|y\|_p^p + \frac{1}{2}\langle \mathbf{C}(y)y, y\rangle + \frac{1}{2}\langle \mathbf{C}(y)x, x\rangle$$

$$= \|y\|_p^p - p\|y\|_p^p + p\sum_{i=1}^d x_i y_i |y_i|^{p-2} - p\sum_{i=1}^d x_i y_i |y_i|^{p-2}$$

$$\quad + \frac{1}{2}\langle \mathbf{C}(y)y, y\rangle + \frac{1}{2}\langle \mathbf{C}(y)x, x\rangle$$

$$= \|y\|_p^p - p\|y\|_p^p + p\sum_{i=1}^d x_i y_i |y_i|^{p-2} - \frac{p}{\|y\|_p^p}\left(\sum_{i=1}^d x_i y_i |y_i|^{p-2}\right)\left(\sum_{j=1}^d |y_j|^p\right)$$

$$\quad + \frac{1}{2}\langle \mathbf{C}(y)y, y\rangle + \frac{1}{2}\langle \mathbf{C}(y)x, x\rangle .$$

To finalize, we proceed with the last few equalities:

$$\|x\|_p^p \geq \|y\|_p^p - p\|y\|_p^p + p\sum_{i=1}^d x_i y_i |y_i|^{p-2} - \frac{p}{\|y\|_p^p}\left(\sum_{i=1}^d x_i y_i |y_i|^{p-2}\right)\left(\sum_{j=1}^d y_j y_j |y_j|^{p-2}\right)$$

$$\quad + \frac{1}{2}\langle \mathbf{C}(y)y, y\rangle + \frac{1}{2}\langle \mathbf{C}(y)x, x\rangle$$

$$= \|y\|_p^p - p\|y\|_p^p + p\sum_{i=1}^d x_i y_i |y_i|^{p-2} - \frac{p}{\|y\|_p^p}\sum_{i=1}^d\sum_{j=1}^d x_i y_i |y_i|^{p-2} y_j |y_j|^{p-2} y_j$$

$$\quad + \frac{1}{2}\langle \mathbf{C}(y)y, y\rangle + \frac{1}{2}\langle \mathbf{C}(y)x, x\rangle$$

$$= \|y\|_p^p - p\|y\|_p^p + p\sum_{i=1}^d x_i y_i |y_i|^{p-2} - \langle \mathbf{C}(y)x, y\rangle + \frac{1}{2}\langle \mathbf{C}(y)y, y\rangle + \frac{1}{2}\langle \mathbf{C}(y)x, x\rangle$$

$$= \|y\|_p^p + p\sum_{i=1}^d x_i y_i |y_i|^{p-2} - p\sum_{i=1}^d |y_i|^p + \frac{1}{2}\langle \mathbf{C}(x-y), x-y\rangle$$

$$= \|y\|_p^p + \sum_{i=1}^d p y_i |y_i|^{p-2}(x_i - y_i) + \frac{1}{2}\|x-y\|_{\mathbf{C}(y)}^2$$

$$= \|y\|_p^p + \langle \nabla f(y), x-y\rangle + \frac{1}{2}\|x-y\|_{\mathbf{C}(y)}^2.$$

$\square$

**Lemma E.13.** *Suppose $p \geq 2$. The function $f(x) = \|x\|^p$ satisfies Equation (5) with either of the two curvature mappings:*

$$(1)\ \mathbf{C}(y) = p\|y\|_2^{p-2}\mathbf{I}. \qquad (2)\ \mathbf{C}(y) = p\|y\|_2^{p-4}yy^\top.$$

*Proof.* (1). When $p = 2$, we have $\mathbf{C}(y) = 2\mathbf{I}$. Therefore, $f$ satisfies Equation (5) because $\|x\|^2$ is 2-strongly-convex.

Now suppose $p > 2$. By applying Young's Inequality we get that:

$$\frac{(\|x\|^2)^{\frac{p}{2}}}{\frac{p}{2}} + \frac{(\|y\|^{p-2})^{\frac{p}{p-2}}}{\frac{p}{p-2}} \geq \|x\|^2 \|y\|^{p-2},$$

and rearranging

$$\|x\|^p + \left(\frac{p}{2} - 1\right) \|y\|^p \geq \frac{p}{2} \|y\|^{p-2} \|x\|^2.$$

We get our result from the above inequality and by observing that $\nabla f(y) = p\|y\|_2^{p-2} y$.

$$
\begin{aligned}
\|x\|_2^p &\geq \left(1 - \frac{p}{2}\right) \|y\|_2^p + \frac{p}{2} \|y\|_2^{p-2} \|x\|_2^2 \\
&= \|y\|_2^p - p\|y\|_2^p + \frac{p}{2}\|y\|_2^{p-2}\|x\|_2^2 + \frac{p}{2}\|y\|_2^p \\
&= \|y\|_2^p + p\|y\|_2^{p-2}\langle x, y\rangle - p\|y\|_2^{p-2}\langle x, y\rangle - p\|y\|_2^p + \frac{p}{2}\|y\|_2^{p-2}\|x\|_2^2 + \frac{p}{2}\|y\|_2^p \\
&= \|y\|_2^p + p\|y\|_2^{p-2}\langle x, y\rangle - p\|y\|_2^p - p\|y\|_2^{p-2}\langle x, y\rangle + \frac{p}{2}\|y\|_2^{p-2}\langle x, x\rangle + \frac{p}{2}\|y\|_2^{p-2}\langle y, y\rangle \\
&= \|y\|_2^p + p\|y\|_2^{p-2}\langle x, y\rangle - p\|y\|_2^{p-2}\langle y, y\rangle - \langle \mathbf{C}(y)x, y\rangle + \frac{1}{2}\langle \mathbf{C}(y)x, x\rangle + \frac{1}{2}\langle \mathbf{C}(y)y, y\rangle \\
&= \|y\|_2^p + p\|y\|_2^{p-2}\langle x, y\rangle - p\|y\|_2^{p-2}\langle y, y\rangle + \frac{1}{2}\langle \mathbf{C}(y)(x - y), x - y\rangle \\
&= \|y\|_2^p + \left\langle p\|y\|_2^{p-2}y, x - y\right\rangle + \frac{1}{2}\|x - y\|_{\mathbf{C}(y)}^2 \\
&= \|y\|_2^p + \left\langle p\|y\|_2^{p-2}y, x - y\right\rangle + \frac{1}{2}\|x - y\|_{\mathbf{C}(y)}^2 \\
&= \|y\|_2^p + \langle \nabla f(y), x - y\rangle + \frac{1}{2}\|x - y\|_{\mathbf{C}(y)}^2.
\end{aligned}
$$

(2) When $p = 2$, we have $\mathbf{C}(y) = \frac{p}{\|y\|_2^2} yy^\top$. Since $\|x\|^2$ is a square of an absolutely convex function, it satisfies Equation (5) with curvature mapping $\mathbf{C}(y)$. For more details, refer to section 7 and F.

Suppose that $p > 2$. As done previously, we can use Young's Inequality to obtain:

$$\|x\|^p + \left(\frac{p}{2} - 1\right)\|y\|^p \geq \frac{p}{2}\|y\|^{p-2}\|x\|^2. \tag{34}$$

Moreover, by Cauchy-Schwarz:

$$\|x\|_2 \|y\|_2 \geq |\langle x, y\rangle| \implies \|x\|_2^2 \|y\|_2^2 \geq (\langle x, y\rangle)^2.$$

Multiplying both sides by $-\frac{p}{2}\|y\|^{p-4}$ we get,

$$-\frac{p}{2}\|x\|^2\|y\|^{p-2} \leq -\frac{p}{2}\|y\|^{p-4}(\langle x, y\rangle)^2,$$

Adding $\|x\|_2^p + \left(\frac{p}{2} - 1\right)\|y\|_2^p$ to both sides and by using Equation (34),

$$\|x\|^p - \frac{p}{2}\|y\|^{p-4}(\langle x, y\rangle)^2 + \left(\frac{p}{2} - 1\right)\|y\|^p \geq 0.$$

We can reorder the terms to obtain the result:

$$\|x\|_2^p \geq \left(1 - \frac{p}{2}\right)\|y\|_2^p + \frac{p}{2}\|y\|_2^{p-4}(\langle x, y\rangle)^2$$

$$= \|y\|_2^p - p\|y\|_2^p + \frac{p}{2}\|y\|_2^{p-4}(\langle x, y\rangle)^2 + \frac{p}{2}\|y\|_2^p$$

$$= \|y\|_2^p - p\|y\|_2^p + \frac{p}{2}\|y\|_2^{p-4}x^\top yy^\top x + \frac{p}{2}\|y\|_2^{p-4}y^\top yy^\top y$$

$$= \|y\|_2^p - p\|y\|_2^p + \frac{1}{2}\langle \mathbf{C}(y)x, x\rangle + \frac{1}{2}\langle \mathbf{C}(y)y, y\rangle$$

$$= \|y\|_2^p - p\|y\|_2^p + p\|y\|_2^{p-2}\langle x, y\rangle - p\|y\|_2^{p-2}\langle x, y\rangle + \frac{1}{2}\langle \mathbf{C}(y)x, x\rangle + \frac{1}{2}\langle \mathbf{C}(y)y, y\rangle$$

$$= \|y\|_2^p - p\|y\|_2^p + p\|y\|_2^{p-2}\langle x, y\rangle - p\|y\|_2^{p-4}x^\top yy^\top y + \frac{1}{2}\langle \mathbf{C}(y)x, x\rangle + \frac{1}{2}\langle \mathbf{C}(y)y, y\rangle$$

$$= \|y\|_2^p - p\|y\|_2^p + p\|y\|_2^{p-2}\langle x, y\rangle - \langle \mathbf{C}(y)x, y\rangle + \frac{1}{2}\langle \mathbf{C}(y)x, x\rangle + \frac{1}{2}\langle \mathbf{C}(y)y, y\rangle$$

$$= \|y\|_2^p - p\|y\|_2^p + p\|y\|_2^{p-2}\langle x, y\rangle + \langle \mathbf{C}(y)(x - y), x - y\rangle$$

$$= \|y\|_2^p - p\|y\|_2^{p-2}\langle y, y\rangle + p\|y\|_2^{p-2}\langle x, y\rangle + \frac{1}{2}\|x - y\|_{\mathbf{C}(y)}^2$$

$$= \|y\|_2^p + \left\langle p\|y\|_2^{p-2}y, x - y\right\rangle + \frac{1}{2}\|x - y\|_{\mathbf{C}(y)}^2$$

$$= \|y\|_2^p + \langle \nabla f(y), x - y\rangle + \frac{1}{2}\|x - y\|_{\mathbf{C}(y)}^2.$$

$$\square$$

**Lemma E.14.** *Suppose $p \geq 2$ and let $f : \mathbb{R}^d \to \mathbb{R}$ be defined as $f(x) = \|x\|_p^2$. Then $f$ satisfies Equation (5) with the following curvature mapping:*

$$\mathbf{C}(y) = \frac{2}{\|y\|_p^{p-2}}\,\mathrm{Diag}(|y_1|^{p-2},\,\ldots,\,|y_d|^{p-2})$$

*Proof.* Using Holder's inequality we can see that,

$$\left(\sum_{i=1}^d |x_i|^2\,|y_i|^{p-2}\right)^p \leq \left(\sum_{i=1}^d |x_i|^p\right)^2 \left(\sum_{i=1}^d |y_i|^{p-2}\right)^{p-2} = \|x\|_p^{2p}\|y\|_p^{p(p-2)}$$

We raise both sides to the power of $\frac{1}{p}$ and proceed by rearranging some terms:

$$\|x\|_p^2 \geq \frac{1}{\|y\|_p^{p-2}}\sum_{i=1}^d |x_i|^2\,|y_i|^{p-2}$$

$$= \|y\|_p^2 - \|y\|_p^2 + \frac{1}{\|y\|_p^{p-2}}\sum_{i=1}^d |x_i|^2\,|y_i|^{p-2}$$

$$= \|y\|_p^2 - \frac{1}{\|y\|_p^{p-2}}\sum_{i=1}^d |y_i|^p + \frac{1}{\|y\|_p^{p-2}}\sum_{i=1}^d |x_i|^2\,|y_i|^{p-2}$$

$$= \|y\|_p^2 - \frac{2}{\|y\|_p^{p-2}}\sum_{i=1}^d |y_i|^p + \frac{1}{\|y\|_p^{p-2}}\sum_{i=1}^d |x_i|^2\,|y_i|^{p-2} + \frac{1}{\|y\|_p^{p-2}}\sum_{i=1}^d |y_i|^p$$

$$= \|y\|_p^2 + \frac{2}{\|y\|_p^{p-2}}\sum_{i=1}^d y_i\,|y_i|^{p-2}\,x_i - \frac{2}{\|y\|_p^{p-2}}\sum_{i=1}^d |y_i|^p + \frac{1}{\|y\|_p^{p-2}}\sum_{i=1}^d |x_i|^2\,|y_i|^{p-2}$$

$$\qquad - \frac{2}{\|y\|_p^{p-2}}\sum_{i=1}^d y_i\,|y_i|^{p-2}\,x_i + \frac{1}{\|y\|_p^{p-2}}\sum_{i=1}^d |y_i|^p$$

We can arrive at our result by realizing that the last three terms are equal to $\|x - y\|^2_{\mathbf{C}(y)}$ and the middle two terms are equal to $\langle \nabla f(y), x - y \rangle$. Observe that $\frac{\partial f}{\partial y_i} = \frac{2}{\|y\|^{p-2}_p} y_i |y_i|^{p-2}$. Therefore,

$$
\|x\|^2_p \geq \|y\|^2_p + \sum_{i=1}^d \frac{2y_i |y_i|^{p-2}}{\|y\|^{p-2}_p} (x_i - y_i) + \frac{1}{\|y\|^{p-2}_p} \sum_{i=1}^d |x_i|^2 |y_i|^{p-2}
$$

$$
- \frac{2}{\|y\|^{p-2}_p} \sum_{i=1}^d y_i |y_i|^{p-2} x_i + \frac{1}{\|y\|^{p-2}_p} \sum_{i=1}^d |y_i|^p
$$

$$
= \|y\|^2_p + \langle \nabla f(y), x - y \rangle + \frac{1}{2} \langle \mathbf{C}(y)x, x \rangle - \langle \mathbf{C}(y)x, y \rangle + \frac{1}{2} \langle \mathbf{C}(y)y, y \rangle
$$

$$
= \|y\|^2_p + \langle \nabla f(y), x - y \rangle + \frac{1}{2} \langle \mathbf{C}(y)(x - y), x - y \rangle
$$

$$
= \|y\|^2_p + \langle \nabla f(y), x - y \rangle + \frac{1}{2} \|x - y\|^2_{\mathbf{C}(y)}
$$

□

**Lemma E.15.** *Suppose $p \geq 1$. Let $g : \mathbb{R}^d \to \mathbb{R}$ be $g(x) = \|x\|_p$. Then $f := g^2$ satisfies Equation (5) with the following curvature mapping:*

$$
\mathbf{C}(y) = 2\nabla g(y)\nabla g(y)^\top.
$$

*Proof.* By Lemma F.2, $g$ is absolutely convex for $p \geq 1$. Therefore, $g^2$ satisfies Equation (5) with curvature mapping $\mathbf{C}$. □

### E.4 LOWER AND UPPER BOUND EXAMPLES

**Lemma E.16.** *Let $\mathbf{G}$ be a symmetric positive semi-definite matrix. Let $f : \mathbb{R}^d \to \mathbb{R}$ where $f(x) = \|x\|^2_{\mathbf{G}}$. Then $f$ satisfies Assumption 2.1 with curvature mapping $\mathbf{C}(y) \equiv 2\mathbf{G}$ and constant $L_{\mathbf{C}} = 0$.*

*Proof.* We start by computing:

$$
0 = \|y\|^2_{\mathbf{G}} - 2\|y\|^2_{\mathbf{G}} + \|y\|^2_{\mathbf{G}}
$$

$$
= -\|x\|^2_{\mathbf{G}} + \|x\|^2_{\mathbf{G}} + \|y\|^2_{\mathbf{G}} - 2\|y\|^2_{\mathbf{G}} + \|y\|^2_{\mathbf{G}}
$$

$$
= -\|x\|^2_{\mathbf{G}} + \|y\|^2_{\mathbf{G}} - 2\langle \mathbf{G}y, y \rangle + \langle \mathbf{G}y, y \rangle + \langle \mathbf{G}x, x \rangle
$$

$$
= -\|x\|^2_{\mathbf{G}} + \|y\|^2_{\mathbf{G}} - 2\langle \mathbf{G}y, y \rangle + \langle \mathbf{G}y, y \rangle + \langle \mathbf{G}x, x \rangle
$$

$$
= -\|x\|^2_{\mathbf{G}} + \|y\|^2_{\mathbf{G}} + 2\langle \mathbf{G}y, x \rangle - 2\langle \mathbf{G}y, y \rangle + \langle \mathbf{G}y, y \rangle - 2\langle \mathbf{G}y, x \rangle + \langle \mathbf{G}x, x \rangle
$$

$$
= -\|x\|^2_{\mathbf{G}} + \|y\|^2_{\mathbf{G}} + \langle 2\mathbf{G}y, x - y \rangle + \langle \mathbf{G}y, y \rangle - 2\langle \mathbf{G}y, x \rangle + \langle \mathbf{G}x, x \rangle
$$

$$
= -\|x\|^2_{\mathbf{G}} + \|y\|^2_{\mathbf{G}} + \langle 2\mathbf{G}y, x - y \rangle + \frac{1}{2}2\|x - y\|^2_{\mathbf{G}}.
$$

Rearranging the terms we get

$$
\|x\|^2_{\mathbf{G}} = \|y\|^2_{\mathbf{G}} + \langle 2\mathbf{G}y, x - y \rangle + \frac{1}{2}2\|x - y\|^2_{\mathbf{G}}
$$

$$
= \|y\|^2_{\mathbf{G}} + \langle \nabla f(y), x - y \rangle + \frac{1}{2}\|x - y\|^2_{2\mathbf{C}(y)}.
$$

□

**Lemma E.17.** *Let $p \geq 2$. Suppose $g : \mathbb{R}^d \to \mathbb{R}$ with $g(x) = \|x\|_p^2$. Let $f := g^2$. Then $f$ satisfies Assumption 2.1 with constant $L_{\mathbf{C}} = 2(p-1)$ and either of the two curvature mappings:*

$$\mathbf{B}(y) = \frac{2}{\|y\|_p^{p-2}} \operatorname{Diag}(|y_1|^{p-2}, \ldots, |y_d|^{p-2}) \qquad \mathbf{B}(y) = 2\nabla g(y)\nabla g(y)^\top.$$

*Proof.* In Lemma E.14, we proved that $f$ satisfies inequality (5) with the abovementioned curvature mappings. Now we will show that $f$ is smooth so it satisfies inequality (6). For $p = 2$. it is clear that $f$ is 2-smooth. We focus on the case where $p > 2$.

Since $p > 2$ we have that $1 < q < 2$ where $q = \frac{p}{p-1}$. Kakade et al. (2012) proved that $h(x) = \frac{1}{2} \|x\|_q^2$ is strongly-convex with respect to the $L_q$ norm with $\mu = q - 1$. We also know that $L_p$ norms are a decreasing function of $p$. Therefore, $h(x) = \frac{1}{2} \|x\|_q^2$ is also strongly-convex with respect to the $L_2$ norm because $\|x - y\|_2^2 \leq \|x - y\|_q^2$:

$$\frac{1}{2} \|x\|_q^2 \geq \frac{1}{2} \|y\|_q^2 + \langle \nabla h(y), x - y \rangle + \frac{1}{2}\mu \|x - y\|_q^2$$

$$\geq \frac{1}{2} \|y\|_q^2 + \langle \nabla h(y), x - y \rangle + \frac{1}{2}\mu \|x - y\|_2^2$$

The Frenchel conjugate of $\frac{1}{2} \|\cdot\|_q^2$ is $\frac{1}{2} \|\cdot\|_p^2$ because the dual norm of $\|\cdot\|_q$ is $\|\cdot\|_p$. Kakade et al. (2012) showed that if $h$ is $\mu$-strongly-convex then the Frenchel conjugate of $h$ is $\frac{1}{\mu}$-smooth. Therefore, $\frac{1}{2} \|x\|_p^2$ is $L$-smooth with $L = \frac{1}{q-1} = p - 1$. Thus, $f(x) = \|x\|_p^2$ is smooth with constant $2(p-1)$. $\qquad\square$

**Lemma E.18.** *Suppose $a, b \in \mathbb{R}$ and $a \neq 0$ and $b > 0$. The function $f : \mathbb{R} \to \mathbb{R}$ defined as*

$$f(x) = \sqrt{ax^4 + b}$$

*satisfies the upper and lower bounds in Assumption 2.1 with $\mathbf{C}(y) = \frac{2ay^2}{f(y)}$ and $L_{\mathbf{C}} = \sqrt{8a}$.*

*Proof.* Observe that $x^2 + y^2 \geq 2x^2 y^2$. Multiply both sides by $ab$ we get $ab(x^2 + y^2) \geq 2abx^2 y^2$. Then we add $a^2 x^4 y^4 + b^2$ to both sides,

$$a^2 x^4 y^4 + abx^2 + aby^2 + b^2 \geq a^2 x^4 y^4 + 2abx^2 y^2 + b^2.$$

We can write this equivalently as,

$$(ax^4 + b)(ay^4 + b) \geq (b + ax^2 y^2)^2.$$

Then taking the square root of both sides,

$$\sqrt{(ax^4 + b)(ay^4 + b)} \geq b + ax^2 y^2 = ay^4 + b - ay^4 + ax^2 y^2.$$

Rearranging the terms,

$$\sqrt{ax^4 + b}\sqrt{ay^4 + b} \geq ay^4 + b - ay^4 + ax^2 y^2$$

$$= ay^4 + b - 2ay^4 + ax^2 y^2 + ay^4$$

$$= ay^4 + b + 2axy^3 - 2ay^4 + ax^2 y^2 - 2axy^3 + ay^4$$

$$= ay^4 + b + 2ay^3(x - y) + ay^2(x^2 - 2xy + y^2)$$

$$= ay^4 + b + 2ay^3(x - y) + \frac{1}{2}2ay^2(x - y)^2.$$

Divide both sides by $\sqrt{ay^4 + b}$ to obtain our result,

$$\sqrt{ax^4 + b} \geq \sqrt{ay^4 + b} + \frac{2ay^3}{\sqrt{ay^4 + b}}(x - y) + \frac{1}{2}\frac{2ay^2}{\sqrt{ax^4 + b}}(x - y)^2.$$

To compute $L_{\mathbf{C}}$, note that $f$ is $L$-smooth so we can find an upper bound on $f''$ which is given by $L_{\mathbf{C}} = \sqrt{8a}$. $\qquad\square$

**Lemma E.19.** *Suppose $\delta > 0$. Suppose $h : \mathbb{R} \to \mathbb{R}$ is such that:*

$$h(x) = \begin{cases} \frac{1}{2}x^2 & |x| \leq \delta \\ \delta(|x| - \frac{1}{2}\delta) & |x| > \delta \end{cases}.$$

*Let $f = h^2$. Then $f$ satisfies (2.1) with constant $L_{\mathbf{C}} = 2\delta^2$ and curvature mapping*

$$\mathbf{C}(y) = \begin{cases} y^2 & |y| \leq \delta \\ \delta^2 & |y| > \delta \end{cases}$$

*Notice that $L_{\mathbf{C}}$ is less than the L-smoothness constant of $f$, which is $3\delta^2$ (the tightest bound on the second derivative of $f$).*

*Proof.* First, we will prove that $f$ satisfies inequality (5). We will split the proof into four cases and prove the inequality holds in each case.

Case $|x|, |y| \leq \delta$. We know $x^4 + y^4 \geq 2x^2y^2$. We divide both sides by $4$ and rearrange the terms,

$$\frac{1}{4}x^4 \geq -\frac{1}{4}y^4 + \frac{1}{2}x^2y^2 = \frac{1}{4}y^4 - y^4 + \frac{1}{2}y^4 + \frac{1}{2}x^2y^2$$

$$= \frac{1}{4}y^4 - y^4 + \frac{1}{2}y^4 + \frac{1}{2}x^2y^2 + xy^3 - xy^3$$

$$= \frac{1}{4}y^4 - y^4 + xy^3 + \frac{1}{2}y^2(x^2 - 2xy + y^2)$$

$$= \frac{1}{4}y^4 - y^3(x - y) + \frac{1}{2}y^2(x - y)^2.$$

We have our result because $f'(y) = y^3$ for $|y| \leq \delta$.

Case $|x| \geq \delta$ and $|y| \leq \delta$. For any $|y| \leq \delta$ define $r : \mathbb{R} \to \mathbb{R}$ as the following,

$$r(x) = \frac{y^4}{4} + \frac{\delta^4}{4} + \delta^2 x^2 - \delta^3 |x| - \frac{x^2 y^2}{2}$$

First, we need to show that $r(x) \geq 0$. Suppose $x \geq \delta$. When $x = \delta$,

$$r(x) = \frac{y^4}{4} - \frac{y^2\delta^2}{2} + \frac{\delta^4}{4} = \frac{1}{4}(y^2 - \delta^2) \geq 0.$$

Therefore, for $x > \delta$, if we show that $r'(x) \geq 0$ then $r(x) \geq 0$. By a simple computation, we get that

$$r'(x) = 2\delta^2 x - \delta^3 - \frac{xy^2}{2}.$$

Since $x \geq \delta$ then obviously $x \geq \frac{2}{3}\delta$ so $\frac{3}{2}x - \delta \geq 0$. Rearranging the terms and multiplying the entire inequality by $\delta$ we get that

$$2\delta^2 x - \delta^3 - \frac{x\delta^2}{2} \geq 0.$$

It is easy to show that $-y^2 \geq -\delta^2$ because $|y| \leq \delta$. Therefore,

$$r'(x) = 2\delta^2 x - \delta^3 - \frac{xy}{2} \geq 2\delta^2 x - \delta^3 - \frac{x\delta^2}{2} \geq 0.$$

Now suppose $x \leq -\delta$. Observe that $r(-\delta) = r(\delta) \geq 0$. Thus, if we show that for $x \leq -\delta$, $r'(x) \leq 0$ then $r(x) \geq 0$. For $x \leq -\delta$, we have that

$$r'(x) = 2\delta^2 x + \delta^3 - \frac{xy^2}{2}.$$

Notice that $y^2 \leq \delta^2$ because $|y| \leq \delta$. Then we have that $2\delta^2 - \frac{y^2}{2} \geq \frac{3\delta^2}{2}$. Multiplying both sides of this inequality by $x$ we obtain,

$$x\left(2\delta^2 - \frac{y^2}{2}\right) \leq \frac{3\delta^2 x}{2} \leq \frac{3\delta^3}{2} \leq -\delta^3.$$

The inequality was reversed because $x \leq -\delta < 0$ and the second inequality also follows from $x \leq -\delta$. Rearranging the terms in this inequality we see that

$$r'(x) = 2\delta^2 x - \delta^3 - \frac{xy^2}{2} \leq 0.$$

Therefore, we have shown that $r(x) \geq 0$ for arbitrary $abs y \leq \delta$. Then by rearranging the terms in $r$, we have that

$$\delta^2 x^2 - \delta^3 |x| + \frac{\delta^4}{4} \geq -\frac{y^4}{4} + \frac{x^2 y^2}{2}.$$

The left-hand side is equal to $\delta^2(|x| - \frac{1}{2}\delta) = f(x)$. In the previous case, we showed that the right-hand side is equal to $\frac{1}{4}y^4 - y^3(x-y) + \frac{1}{2}y^2(x-y)^2$. Therefore, we have our result.

Case $|x|, |y| \geq \delta$. First, we show that the following inequality holds:

$$x^2 - 2xy + y^2 - 2\delta\left(|x| - x\frac{y}{|y|}\right) \geq 0. \tag{35}$$

For $x, y \geq \delta$ and $x, y \leq -\delta$ it is easy to show because $(x-y)^2 \geq 0$. Now suppose $x \geq \delta$ and $y \leq -\delta$. Then we must show that

$$x^2 - 2xy + y^2 - 4\delta x \geq 0.$$

Since $y \leq -\delta$ we have that $(x-y)^2 \geq (x+\delta)^2$. Therefore,

$$(x-y)^2 - 4\delta x \geq (x+\delta)^2 - 4\delta x = (x-\delta)^2 \geq 0.$$

Now suppose $x \leq -\delta$ and $y \geq \delta$. Similar to before, we need to show

$$x^2 - 2xy + y^2 + 4\delta x \geq 0.$$

Since $y \geq \delta$ we obtain $4xy \leq 4\delta x$ by multiplying both sides by $4x$ and reversing the inequality because $x \leq -\delta < 0$. Therefore,

$$(x-y)^2 + 4\delta x \geq (x-y)^2 + 4xy = (x+y)^2 \geq 0.$$

As a result, we have shown inequality (35) holds. We can rewrite the inequality as

$$\frac{x^2}{2} - xy + \frac{y^2}{2} - \delta|x| + \delta x\frac{y}{|y|} \geq 0.$$

Moving some terms to the right-hand side we get,

$$x^2 - \delta|x| + \frac{\delta^2}{4} \geq -\frac{-y^2}{2} + \frac{\delta^2}{4} + xy + \frac{x^2}{2} - \delta\frac{xy}{|y|}$$

$$= \left(|y| - \frac{1}{2}\delta\right)^2 + 2xy - \delta\frac{xy}{|y|} - 2y^2 + \delta|y| + \frac{x^2}{2} - xy + \frac{y^2}{2}.$$

Recall, that $f'(y) = 2\delta^2\left(|y| - \frac{1}{2}\delta\right)\frac{y}{|y|}$. Observe that we can factor the left-hand side of the inequality and after multiplying both sides by $\delta^2$ we get our result:

$$\delta^2\left(|x| - \frac{1}{2}\delta\right)^2 \geq \delta^2\left(|y| - \frac{1}{2}\delta\right)^2 + 2\delta^2 xy - \delta^2\frac{xy}{|y|} - 2\delta^2 y^2 + \delta^3|y| + \frac{\delta^2}{2}x^2 - \delta^2 xy + \frac{\delta^2}{2}y^2$$

$$= \delta^2\left(|y| - \frac{1}{2}\delta\right)^2 + 2\delta^2\left(|y| - \frac{1}{2}\delta\right)\frac{y}{|y|}(x-y) + \frac{1}{2}\delta^2(x-y)^2$$

The case where $|x| \leq \delta, |y| \geq \delta$ is similar to the previous cases. Using some elementary calculus, one can show that

$$\frac{x^4}{4} + \frac{\delta^2 y^2}{2} - \frac{\delta^4}{4} - \delta^2 xy - \frac{\delta^2 x^2}{2} + \delta^3 x \geq 0.$$

Rearranging the terms above directly leads to the result.

Now we show that $f$ satisfies inequality (6) with $L_{\mathbf{C}} = 2\delta^2$. In the case where $|y| \geq \delta$, $L_{\mathbf{C}} + \mathbf{C}(y) = 3\delta^2$ is the $L$-smoothness constant of $f$ so the inequality holds. We consider the case where $|y| \leq \delta$.

Case $|x|, |y| \leq \delta$. Then $xy \leq |xy| \leq \delta^2$ so $x^2 + xy \leq 2\delta^2$. Adding $y^2$ to both sides and multiplying by $(x - y)^2$ we obtain,

$$(x - y)^2(y^2 + 2\delta^2) \geq (x - y)^2(x^2 + xy + y^2)$$
$$= (x^3 - y^3)(x - y).$$

By Lemma E.8 we have our result.

Case $|x| \geq \delta, |y| \leq \delta$. We must show that the following inequality holds:

$$\frac{y^4}{4} - \frac{x^2 y^2}{2} + 2\delta^2 xy - \delta^2 y^2 - \delta^3 |x| + \frac{\delta^4}{4} \leq 0.$$

We leave out the details of the calculations. The proof is similar to the same case for showing the lower bound. We can define a polynomial in $x$ for arbitrary $|y| \leq \delta$. Then we show that this polynomial is less than 0 for $x \geq \delta$ by computing the value at $\delta$ and show that the derivative is negative for $x \geq \delta$ We proceed similarly for $x \leq -\delta$. By rearranging and manipulating the terms in the above inequality we can arrive at our result. These calculations are similar to the previous cases so we exclude them for brevity. □

# F ABSOLUTELY CONVEX FUNCTIONS

Discussing the theory constructions derived from Assumption 2.1, we introduced a stand-alone class of functions, satisfying an absolute convexity condition. In this section, we derive more properties and examples. Let us remind that a function $\phi : \mathbb{R}^d \to \mathbb{R}$ is absolutely convex if and only if:

$$\phi(y) \geq |\phi(x) + \langle \nabla\phi(x), y - x \rangle|, \qquad \forall x, y \in \mathbb{R}^d. \tag{36}$$

Our first statement is a Lemma that establishes calculus in the spirit of Lemma 5.1 in the main text.

**Lemma F.1.** *Let $\phi, \phi_1, \phi_2 : \mathbb{R}^d \to \mathbb{R}$ be absolutely convex, and let $\mathbf{A} \in \mathbb{R}^{d \times m}$, $b \in \mathbb{R}^d$ and $\alpha \geq 0$. Then*

  *(i) $\phi + \alpha$ is absolutely convex.*

  *(ii) $\alpha\phi$ is absolutely convex.*

  *(iii) $\phi_1 + \phi_2$ is absolutely convex.*

  *(iv) $\phi(\mathbf{A}x + b)$ is absolutely convex.*

*Proof.* We prove each statement:

  (i) $\psi(x) := \phi(x) + \alpha \geq |\phi(y) + \langle \nabla\phi(y), x - y \rangle| + \alpha \geq |\phi(y) + \langle \nabla\phi(y), x - y \rangle + \alpha| = |\psi(y) + \langle \nabla\psi(y), x - y \rangle|$.

  (ii) $\psi(x) := \alpha\phi(x) \geq \alpha\left(|\phi(y) + \langle \nabla\phi(y), x - y \rangle|\right) \geq \alpha|\phi(y) + \langle \nabla\phi(y), x - y \rangle| = |\alpha\phi(y) + \langle \alpha\nabla\phi(y), x - y \rangle| = |\psi(y) + \langle \nabla\psi(y), x - y \rangle|$.

  (iii) $\psi(x) := \phi_1(x) + \phi_2(x) \geq |\phi_1(y) + \langle \nabla\phi_1(y), x - y \rangle| + |\phi_2(y) + \langle \nabla\phi_2(y), x - y \rangle| \geq |\phi_1(y) + \langle \nabla\phi_1(y), x - y \rangle + \phi_2(y) + \langle \nabla\phi_2(y), x - y \rangle| = |\psi(y) + \langle \nabla\psi(y), x - y \rangle|$.

  (iv) $\psi(x) = \phi(\mathbf{A}x + b) \geq |\phi(\mathbf{A}y + b) + \langle \nabla\phi(\mathbf{A}y + b), \mathbf{A}x + b - (\mathbf{A}y + b) \rangle| = |\phi(\mathbf{A}y + b) + \langle \mathbf{A}^\top\nabla\phi(\mathbf{A}y + b), x - y \rangle| = |\phi(\mathbf{A}y + b) + \langle A^\top\nabla\phi(\mathbf{A}y + b), x - y \rangle| = |\psi(y) + \langle \nabla\psi(y), x - y \rangle|$.

$\square$

## F.1 EXAMPLES

**Lemma F.2.** *Let $p \geq 1$. Then $\phi : \mathbb{R}^d \to \mathbb{R}$ where $\phi(x) = \|x\|_p$ is absolutely convex.*

*Proof.* We already know that $\phi$ is convex so we show that

$$-\phi(y) - \langle \nabla\phi(y), x - y \rangle \leq \phi(x),$$

where $\frac{\partial \phi(y)}{\partial y_i} = \frac{y_i|y_i|^{p-2}}{\|y\|_p^{p-1}}$ for $x, y \in \mathbb{R}^d$.

Observe that

$$\langle \phi(y), x - y \rangle = \sum_{i=1}^{d} \frac{y_i|y_i|^{p-2}}{\|y\|_p^{p-1}}(x_i - y_i).$$

We make use of Holder's inequality which is stated below for $r, s \geq 0$,

$$\left(\sum_{i=1}^{d} |x_i|^r |y_i|^s\right)^{r+s} \leq \left(\sum_{i=1}^{d} |x_i|^{r+s}\right)^r \left(\sum_{i=1}^{d} |y_i|^{r+s}\right)^s.$$

For any $x, y \in \mathbb{R}^d$ and with $r = 1$ and $s = p - 1$ we have that,

$$\left(\sum_{i=1}^{d} |x_i||y_i|^{p-1}\right)^p \leq \left(\sum_{i=1}^{d} |x_i|^p\right) \left(\sum_{i=1}^{d} |y_i|^p\right)^{p-1}.$$

Simplifying this expression we obtain,

$$\sum_{i=1}^{d} |x_i||y_i|^{p-1} \leq \left( \sum_{i=1}^{d} |x_i|^p \right)^{\frac{1}{p}} \left( \sum_{i=1}^{d} |y_i|^p \right)^{\frac{p-1}{p}}$$

$$= \|x\|_p \left( \left( \sum_{i=1}^{d} |y_i|^p \right)^{\frac{1}{p}} \right)^{p-1}$$

$$= \|x\|_p \|y\|_p^{p-1}.$$

We can obtain a lower bound on the term,

$$\sum_{i=1}^{d} |x_i||y_i|^{p-1} = \sum_{i=1}^{d} |x_i||y_i||y_i|^{p-2} = \sum_{i=1}^{d} |x_iy_i||y_i|^{p-2}$$

$$\geq \sum_{i=1}^{d} -(x_iy_i)|y_i|^{p-2}$$

$$= -\sum_{i=1}^{d} x_iy_i|y_i|^{p-2}.$$

Therefore,

$$\|x\|_p \|y\|_p^{p-1} \geq -\sum_{i=1}^{d} x_iy_i|y_i|^{p-2}.$$

Now add $\|y\|_p^p = \sum_{i=1}^{d} |y_i|^p$ to both sides of the inequality we get

$$\|x\|_p \|y\|_p^{p-1} + \|y\|_p^p \geq \sum_{i=1}^{d} |y_i|^p - \sum_{i=1}^{d} x_iy_i|y_i|^{p-2} = \sum_{i=1}^{d} y_i^2|y_i|^{p-2} - \sum_{i=1}^{d} x_iy_i|y_i|^{p-2}$$

$$= \sum_{i=1}^{d} \left( y_i^2|y_i|^{p-2} - x_iy_i|y_i|^{p-2} \right) = \sum_{i=1}^{d} y_i|y_i|^{p-2}(y_i - x_i)$$

$$= -\sum_{i=1}^{d} y_i|y_i|^{p-2}(x_i - y_i).$$

Now divide both sides of this inequality by $\|y\|_p^{p-1}$,

$$\|x\|_p + \|y\|_p \geq -\sum_{i=1}^{d} \frac{y_i|y_i|^{p-2}}{\|y\|_p^{p-1}}(x_i - y_i).$$

By rearranging the terms we get our desired inequality

$$-\|y\|_p - \sum_{i=1}^{d} \frac{y_i|y_i|^{p-2}}{\|y\|_p^{p-1}}(x_i - y_i) \leq \|x\|_p.$$

$\square$

**Lemma F.3.** *There exists an absolutely convex function* $\phi : \mathbb{R} \to \mathbb{R}$ *such that the derivative of* $f := \phi^2$ *is not Lipschitz continuous. Namely,*

$$\phi(x) = \begin{cases} \frac{3}{2}|x| + \frac{1}{2} & |x| \geq 1 \\ x^{\frac{3}{2}} + 1 & 0 \leq x < 1 \\ (-x)^{\frac{3}{2}} + 1 & -1 < x < 0 \end{cases}.$$

*Proof.* Firstly, by a simple computation we can show that $\phi'' \geq 0$ so $\phi$ is convex. Observe that $|f'| \leq \frac{3}{2}$ and $x_\star = 0$. Also notice that $\phi$ is bounded below by $\frac{3}{2}|x|$. Therefore, by Lemma F.16, $\phi$ is absolutely convex.

From a brief computation, we can obtain,

$$
f'(x) = \begin{cases} \frac{3}{2}(3|x|+1)\frac{x}{|x|} & |x| \geq 1 \\ 3(x^{\frac{3}{2}}+1)x^{\frac{1}{2}} & 0 \leq x < 1 \\ 3((-x)^{\frac{3}{2}}+1)(-x)^{\frac{1}{2}} & -1 < x < 0 \end{cases}.
$$

Now suppose by contradiction that $h'$ is Lipschitz continuous. Then there exists an $L > 0$ such that for all $x, y \in \mathbb{R}^d$,

$$
\frac{|f'(x) - f'(y)|}{|x - y|} \leq L.
$$

Specifically, this holds for $0 < x < 1$ and $y = 0$ so we have

$$
\frac{|f'(x) - f'(y)|}{|x - y|} = \frac{|f'(x)|}{|x|} = \frac{\left|3x^2 + 3x^{\frac{1}{2}}\right|}{|x|}
$$

$$
= \frac{3x^2 + 3x^{\frac{1}{2}}}{x} = 3x + \frac{3}{\sqrt{x}} \leq L.
$$

We can find an $x$ small enough such that $\frac{3}{\sqrt{x}} > L$. Therefore, the inequality cannot hold. As a consequence, $f'$ is not Lipschitz continuous. $\qquad\square$

**Lemma F.4.** *Let $\delta > 0$. Then $f, \phi : \mathbb{R} \to \mathbb{R}$ defined below are absolutely convex.*

$$
f(x) = \delta\sqrt{1 + \frac{x^2}{\delta^2}}, \qquad \phi(x) = \begin{cases} \frac{1}{2}x^2 + \frac{\delta^2}{2} & |x| \leq \delta \\ \delta(|x| - \frac{\delta}{2}) + \frac{\delta^2}{2} & x > \delta \end{cases},
$$

*Note that $f$ is the pseudo-Huber loss function and $\phi$ is the Huber loss function.*

*Proof.* Both the Huber loss and pseudo-Huber loss functions are well-known examples of convex loss functions. The minimizer of $f$ and $\phi$ is $x_\star = 0$.

From a simple computation we obtain that $|f'(x)| \leq 1$ for all $x \in \mathbb{R}$,

$$
f'(x) = \frac{x}{\delta\sqrt{1 + \frac{x^2}{\delta^2}}} \leq \frac{x}{\delta\frac{x}{\delta}} = 1.
$$

Also, we know that for all $x \in \mathbb{R}$,

$$
f(x) = \delta\sqrt{1 + \frac{x^2}{\delta^2}} \geq \delta\sqrt{\frac{x^2}{\delta^2}} = \sqrt{x^2} = |x|.
$$

Therefore, by Lemma F.16, $f$ is absolutely convex.

By computing the derivative of $\phi$, we can show that $|\phi'(x)| \leq \delta$ for all $x \in \mathbb{R}$. Now we show that $\phi(x) \geq \delta|x|$.

When $x > \delta$, we can simply observe that $\phi(x) = \delta|x|$. Now consider the case where $x \leq \delta$. We have that,

$$
(|x| - \delta)^2 \geq 0.
$$

Expanding the square we get,

$$
x^2 - 2\delta|x| + \delta^2 \geq 0.
$$

Rearranging the terms and dividing by 2 we see that,

$$
\phi(x) = \frac{1}{2}x^2 + \frac{\delta^2}{2} \geq \delta|x|.
$$

Hence, by Lemma F.16, $\phi$ is absolutely convex. $\qquad\square$

**Lemma F.5.** *Let $a > 0$ and $b \geq 0$. Then $\phi : \mathbb{R} \to \mathbb{R}$ where $\phi(x) = \sqrt{ax^2 + b}$ is absolutely convex.*

*Proof.* Notice that $x_\star = 0$. Also by a simple computation we know can show that $f$ is convex,

$$\phi''(x) = \frac{ab}{(ax^2 + b)^{\frac{3}{2}}} \geq 0.$$

We can compute an upper bound on the derivative of $f$,

$$\phi'(x) = \frac{ax}{\sqrt{ax^2 + b}} \leq \frac{ax}{\sqrt{ax^2}} = \sqrt{a}.$$

Then,

$$\phi(x) = \sqrt{ax^2 + b} \geq \sqrt{ax^2} = \sqrt{a}\sqrt{x^2} = \sqrt{a}\,|x|\,.$$

Since $\phi(x) \geq \sqrt{a}\,|x|$, by Lemma F.16, $f$ is absolutely convex. $\qquad\square$

**Lemma F.6.** *The function $\phi : \mathbb{R} \to \mathbb{R}$, defined as follows*

$$\phi(x) = \begin{cases} x + 1 + \frac{1}{x+1} & x \geq 0 \\ 1 - x + \frac{1}{1-x} & x < 0 \end{cases}$$

*is absolutely convex.*

*Proof.* Observe that $x_\star = 0$. By a simple computation, we can show that $\phi''(x) \geq 0$ for all $x \in \mathbb{R}$,

$$\phi''(x) = \begin{cases} \frac{2}{(x+1)^3} & x \geq 0 \\ \frac{2}{(1-x)^3} & x < 0 \end{cases}.$$

Similarly to the previous examples, we compute an upper bound on $\phi'$,

$$\phi'(x) = \begin{cases} 1 - \frac{1}{(x+1)^2} & x \geq 0 \\ \frac{1}{(1-x)^2} - 1 & x < 0 \end{cases}.$$

It is clear that $|\phi'(x)| \leq 1$ for all $x \in \mathbb{R}$. It is easy to shoiw $\phi(x) \geq |x|$. For $x \geq 0$, we have

$$\phi(x) = x + 1 + \frac{1}{x+1} \geq x.$$

For $x < 0$, we have

$$\phi(x) = 1 - x + \frac{1}{1 - x} \geq -x.$$

Therefore, by Lemma F.16, $\phi$ is absolutely convex. $\qquad\square$

### F.2 Functions with zero minimum

Absolutely convex functions have some interesting properties when their minimum is $0$.

**Lemma F.7.** *If $\phi$ is absolutely convex, then the following statements are equivalent:*

1. $\phi(0) = 0$,

2. $\phi(x) = \langle \nabla\phi(x), x \rangle$,

3. $\phi$ is homogeneous of degree 1.

*Proof.* We establish three implications:

$(i) \Rightarrow (ii)$ Pick any $y$. If $\phi(0) = 0$, then using Equation (36) with $x = 0$ leads to $|\phi(y) + \langle \nabla\phi(y), -y \rangle| \leq 0$, which implies $\phi(y) = \langle \nabla\phi(y), y \rangle$.

$(ii) \Rightarrow (iii)$ We start by substituting $\phi(y) = \langle \nabla\phi(y), y \rangle$ and $\phi(x) = \langle \nabla\phi(x), x \rangle$ into $\phi(x) \geq \phi(y) + \langle \nabla\phi(y), x - y \rangle$ to get,

$$\phi(x) = \langle \nabla\phi(x), x \rangle \geq \langle \nabla\phi(y), y \rangle + \langle \nabla\phi(y), x - y \rangle = \langle \nabla\phi(y), x \rangle\,.$$

This means that,
$$\phi(x) = \max_{g \in Q} \langle g, x \rangle,$$
where $Q := \{ \nabla \phi(y) \ : \ y \in \mathbb{R}^d \}$. As a consequence, for any $t \geq 0$ and $x \in \mathbb{R}^d$ we get
$$\phi(tx) = \max_{g \in Q} \langle g, tx \rangle = t \max_{g \in Q} \langle g, x \rangle = t\phi(x).$$

$(iii) \Rightarrow (i)$ Choose any $x \in \mathbb{R}^d$ and $t = 0$. Then $\phi(0) = \phi(tx) = t\phi(x) = 0\phi(x) = 0$.    □

**Lemma F.8.** *Let* $\phi : \mathbb{R}^d \to \mathbb{R}$ *be absolutely convex. Suppose there exists an* $x_\star \in \mathbb{R}^d$ *such that* $\phi(x_\star) = 0$*. Then*

$$\phi(x) = \langle \nabla \phi(x), x - x_\star \rangle.$$

*Proof.* From absolute convexity, we have that

$$0 \leq |\phi(x) + \langle \nabla \phi(x), x_\star - x \rangle| \leq \phi(x_\star) = 0.$$

So
$$\phi(x) + \langle \nabla \phi(x), x_\star - x \rangle = 0.$$

Simply rearranging we get our result,
$$\phi(x) = -\langle \nabla \phi(x), x_\star - x \rangle = \langle \nabla \phi(x), x - x_\star \rangle.$$

□

**Lemma F.9.** *Let* $\phi : \mathbb{R} \to \mathbb{R}$ *be absolutely convex. Suppose there exists an* $x_\star \in \mathbb{R}$ *such that* $\phi(x_\star) = 0$*. Also, suppose that* $\phi$ *is differentiable everywhere but at* $x_\star$*. Then it must be that*

$$\phi(x) = m |x - x_\star| = \begin{cases} -m(x - x_\star) & x < x_\star \\ m(x - x_\star) & x > x_\star \end{cases} \tag{37}$$

*for some* $m \in \mathbb{R}_{\geq 0}$*.*

*Proof.* We have that $f(x) = \phi(x + x_\star)$ is absolutely convex. It is also homogeneous of degree one since $f(0) = 0$. Define $U_1 = \{ x \in \mathbb{R} \mid x > 0 \}$ and $U_2 = \{ x \in \mathbb{R} \mid x < 0 \}$.

By homogeneity, for any $t > 0$ and $x \in U_1$ we have that $f(tx) = tf(x)$. Differentiating both sides with respect to $x$ we get $f'(tx) = f'(x)$. This means that for any $x \in U_1$ we have that $f'(x) = m_1$ for some $m_1 \in \mathbb{R}$. Since $U_1$ is a connected open set, we have that $f(x) = m_1 x$ for $x \in U_1$.

By similar reasoning, for $x \in U_2$, $f'(x) = m_2$ for some $m_2 \in \mathbb{R}$. Also then, $f(x) = m_2 x$ for $x \in U_2$.

Now consider $x \in U_1$ and $y \in U_2$. By absolute convexity we know that
$$m_1 x = f(x) \geq |f(y) + f'(y)(x - y)| = |m_2 y + m_2(x - y)| = |m_2 x| \geq |m_2||x|.$$

Then $m_1 \geq |m_2| \frac{x}{|x|} = |m_2|$ which comes from the fact that $x \in U_1$ so $x > 0$ and thus $\frac{x}{|x|} = 1$. Similarly,
$$m_2 y = f(y) \geq |f(x) + f'(x)(y - x)| = |m_1 x + m_1(y - x)| = |m_1 y| \geq |m_1||y|.$$

Then $|m_1| \leq m_2 \frac{y}{|y|} = -m_2$ because $y \in U_2$ so $y < 0$ and thus $\frac{y}{|y|} = -1$.

Since $0 \leq |m_1| \leq -m_2$ we get $m_2 \leq 0$. Also because $|m_2| \leq m_1$ it must be that $m_2 = -m_1$ where $m_1 \geq 0$.

For brevity, we set $m = m_1$ and so we have that

$$f(x) = \begin{cases} -mx & x < 0 \\ mx & x > 0 \end{cases}.$$

By definition we have that $\phi(x) = f(x - x_\star)$. Therefore, we obtain our desired result

$$\phi(x) = \begin{cases} -m(x - x_\star) & x - x_\star < 0 \\ m(x - x_\star) & x - x_\star > 0 \end{cases}.$$

$\square$

**Lemma F.10.** *Suppose that $\phi : \mathbb{R}^d \to \mathbb{R}$ is absolutely convex and $\phi(0) = 0$. Then $\phi$ is sub-additive,*

$$\phi(x + y) \leq \phi(x) + \phi(y).$$

*Proof.* By Lemma F.7, we know $\phi$ is positively homogeneous of degree one. Also because $\phi$ is convex we have that for any $0 \leq \alpha \leq 1$,

$$\phi(\alpha x + (1 - \alpha)y) \leq \alpha \phi(x) + (1 - \alpha)\phi(y).$$

Selecting $\alpha = \frac{1}{2}$,

$$\phi\left(\frac{1}{2}x + \frac{1}{2}y\right) \leq \frac{1}{2}\phi(x) + \frac{1}{2}\phi(y). \tag{38}$$

By homogeneity of $\phi$,

$$\frac{1}{2}\phi(x + y) \leq \phi\left(\frac{1}{2}x + \frac{1}{2}y\right).$$

By combining the previous inequality with inequality (38) and multiplying by 2 we get our result.

$\square$

**Lemma F.11.** *Suppose that $\phi : \mathbb{R}^d \to \mathbb{R}$ is absolutely convex and $\phi(0) = 0$. Then the epigraph of $\phi$ is a convex cone.*

*Proof.* Now we show that $\operatorname{epi}\phi$ is a convex cone. Suppose $(x, \mu_1) \in \operatorname{epi}\phi$ and $(y, \mu_2) \in \operatorname{epi}\phi$. Suppose $\alpha \geq 0$ and $\beta \geq 0$. Then

$$\phi(\alpha x + \beta y) \leq \phi(\alpha x) + \phi(\beta y) = \alpha \phi(x) + \beta \phi(y) \leq \alpha \mu_1 + \beta \mu_2.$$

The first inequality is from sub-additivity and the first equality is from homogeneity, Therefore, $(\alpha x + \beta y, \alpha \mu_1 + \beta \mu_2) \in \operatorname{epi}\phi$ so $\operatorname{epi}\phi$ is a convex cone. $\square$

**Lemma F.12.** *Suppose that $\phi : \mathbb{R}^d \to \mathbb{R}$ is absolutely convex and $\phi(0) = 0$. Then $\phi$ is even, i.e.*

$$\phi(x) = \phi(-x), \qquad \forall x \in \mathbb{R}^d. \tag{39}$$

*Proof.* By absolute convexity we have that for any $x, y \in \mathbb{R}^d$

$$\begin{aligned}
\langle \nabla \phi(x), x \rangle = \phi(x) &\geq |\phi(y) + \langle \nabla \phi(y), x - y \rangle| \\
&= |\phi(y) + \langle \nabla \phi(y), x \rangle + \langle \nabla \phi(y), -y \rangle| \\
&= |\phi(y) + \langle \nabla \phi(y), x \rangle - \langle \nabla \phi(y), y \rangle| \\
&= |\phi(y) + \langle \nabla \phi(y), x \rangle - \phi(y)| \\
&= |\langle \nabla \phi(y), x \rangle|. \tag{40}
\end{aligned}$$

Similarly,

$$\phi(y) = \langle \nabla \phi(y), y \rangle \geq |\langle \nabla \phi(x), y \rangle|. \tag{41}$$

Now subtitute $y = -x$ into (40),

$$\phi(x) \geq |\langle \nabla \phi(-x), x \rangle| = |\langle \nabla \phi(-x), -x \rangle| = |\phi(-x)| = \phi(-x),$$

where the last equality is because absolutely convex functions are non-negative. Substituting $y = -x$ into (41),

$$\phi(-x) \geq |\langle \nabla \phi(x), -x \rangle| = |\langle \nabla \phi(x), x \rangle| = |\phi(x)| = \phi(x).$$

Therefore, $\phi(x) = \phi(-x)$. $\square$

### F.3 REAL VALUED FUNCTIONS

Simple absolutely convex functions from the real numbers to the real numbers are an instructive playground to understand how to finalize the generalized proofs. Below, we report some properties of such sub-class. In particular, we prove bounded subgradients in Lemma F.15 and a useful result for validating examples in Lemma F.16.

**Lemma F.13.** *Let $f : \mathbb{R} \to \mathbb{R}$ defined as follows, $f(x) = |a + b(x - x_0)|$ for some $a, b, x_0 \in \mathbb{R}$ with $b \neq 0$. Then for any $c > 0$. There exists $x_1, x_2 \in \mathbb{R}$ with $x_2 \geq x_1$, $f(x_1) = f(x_2) = c$, $|x_2 - x_1| = \frac{2c}{|b|}$ and $f'(x_2) \geq 0$ and $f'(x_1) \leq 0$.*

*Proof.* By simply solving the equation $f(x) = c$, we get that

$$x_2 = \frac{c - a}{b} + x_0;$$

$$x_1 = \frac{-c - a}{b} + x_0.$$

Thus,

$$|x_2 - x_1| = \left| \frac{c - a}{b} + x_0 - \frac{-c - a}{b} - x_0 \right| = \left| \frac{c - a + c + a}{b} \right| = \left| \frac{2c}{b} \right|.$$

Suppose without loss of generality that $b > 0$. Then $f'(x_2) = cb > 0$ and $f'(x_1) = -cb < 0$. □

**Lemma F.14.** *Suppose $\phi : \mathbb{R} \to \mathbb{R}$ is a convex function that is not constant with minimizer $x_\star$. Suppose $\phi$ lower bounded by $f(x) = |a + b(x - x_0)|$ for $a, b, x_0 \in \mathbb{R}$ and $b \neq 0$. For any $c > \phi(x_\star)$ there exists an $x'$ such that $\phi(x') = c$ and $|x' - x_\star| \leq \frac{2c}{|b|}$.*

*Proof.* By Lemma F.13, there exists $x_1, x_2 \in \mathbb{R}$ such that $|x_2 - x_1| = \frac{2c}{|b|}$ and $f(x_1) = f(x_2) = c$.

We also know that $x_2 > x_1$ and that $f'(x_2) \geq 0$ and $f'(x_1) \leq 0$. We will show that $x_1 < x_\star < x_2$. Suppose by contradiction that $x_\star \geq x_2$. Since $f$ is a linear function with slope $f'(x_2)$ on the interval $[x_2, \infty)$ we get

$$f(x_\star) = f(x_2) + f'(x_2)(x_\star - x_2) \geq f(x_2) = c > \phi(x_\star),$$

which is a contradiction because $f$ is supposed to be a lower bound on $\phi$. The first inequality follows from the fact that $f'(x_2)(x_\star - x_2) \geq 0$. A similar argument follows if the assumption $x_\star \leq x_1$ is made.

So $c = f(x_1) \leq \phi(x_1)$ and $\phi(x_\star) < c$. By the Intermediate Value Theorem, there exists $x' \in (x_1, x_\star)$ such that $\phi(x') = c$.

Thus $x_1 < x' < x_\star < x_2$. Therefore, $|x' - x_\star| \leq |x_2 - x_1| = \frac{2c}{|b|}$ with $\phi(x') = c$. □

**Lemma F.15.** *Suppose $\phi : \mathbb{R} \to \mathbb{R}$ is absolutely convex and has a minimizer $x_\star$. Then there exists an $M \in \mathbb{R}$ such that $|\phi'(x)| \leq M$ for all $x \in \mathbb{R}$ i.e. $|\phi'|$ is bounded.*

*Proof.* If $\phi$ is constant then its derivative is bounded so we consider the case where $\phi$ is not constant.

We will do a proof by contradiction. Let $c \in \mathbb{R}$ such that $c > \phi(x_\star)$. Let $\epsilon = \frac{c - \phi(x_\star)}{2}$. Note that $|\phi(x_\star) - c| > \epsilon$.

By continuity of $\phi$ at $x_\star$ there must be a $\delta > 0$ such that if $|x - x_\star| \leq \delta$ then $|\phi(x) - \phi(x_\star)| \leq \epsilon$.

Suppose that $|\phi'|$ is unbounded. Therefore, there exists a sequence of numbers $y_n \in \mathbb{R}$ such that

$$\lim_{n \to \infty} |\phi'(y_n)| = \infty$$

For any $n$, let $f_n(x) = |\phi(y_n) + \phi'(y_n)(x - y_n)|$. Since $\phi$ is absolutely convex, $f_n$ must be a lower bound on $\phi$. By Lemma F.14 there exists an $x_n$ such that $\phi(x_n) = c$ and $|x_n - x_\star| \leq \frac{2c}{|\phi'(y_n)|}$ for any $y_n$.

Since $|\phi'|$ is unbounded we can choose a $N$ such that $\frac{2c}{|\phi'(y_N)|} \leq \delta$. Thus $|x_N - x_\star| \leq \delta$. Therefore, $|\phi(x_\star) - \phi(x_N)| = |\phi(x_\star) - c| \leq \epsilon$.

But this contradicts the fact that $|\phi(x_\star) - c| > \epsilon$. $\qquad\square$

**Lemma F.16.** *Suppose $\phi : \mathbb{R} \to \mathbb{R}$ is convex and has a minimizer at $x_\star = 0$. Suppose $\phi'$ is bounded. If $\phi$ can be lower bounded by $h(x) = m\,|x|$ where $|\phi'(y)| \leq m$ for all $y \in \mathbb{R}$ then $\phi$ is absolutely convex.*

*Proof.* Suppose without loss of generality $y < x_\star = 0$. Since $\phi$ is convex we have that $\phi'(y) \leq 0$ by monotoncity of $|\phi'|$. Since $m$ is a bound on $\phi'$ we get

$$m \geq -\phi'(y) \geq 0. \tag{42}$$

Since $h$ is a lower bound on $\phi$ we also know that .

$$\phi(y) \geq -my \geq 0. \tag{43}$$

Now let $f(x) = |\phi(y) + \phi'(y)(x - y)|$. Denote $x^v$ as the point where $f(x^v) = 0$. On the interval $(-\infty, x^v]$, $f$ is equal to the line tangent to $\phi$ at point $y$. So by convexity, $f$ is a lower bound on $\phi$ on the interval $(-\infty, x^v]$. It remains to show that $f$ lower bounds $\phi$ on the interval $[x^v, \infty)$.

First, we show that $x^v = -\frac{\phi(y)}{\phi'(y)} + y \geq x_\star = 0$. We can take the reciprocal of inequality (42) to obtain, $\frac{1}{m} \leq -\frac{1}{\phi'(y)}$. Multiply this inequality by (43) to get, $-y \leq -\frac{\phi(y)}{\phi'(y)}$. We can do this because the terms on both sides of the inequalities are positive. Rearranging we can see that $x^v \geq 0$.

Suppose $x \in [x^v, \infty)$. On this interval, $f$ is a line with slope $-\phi'(y)$ passing through the point $(x^v, 0)$. Thus we can rewrite $f$ as $f(x) = -\phi'(y)(x - x^v)$. Since $x^v \geq 0$ we know that $x - x^v \leq x$. Multiply this inequality by $m \geq -\phi'(y) \geq 0$ to get that $-\phi'(y)(x - x^v) \leq mx$. Since $h(x) = mx$ is a lower bound on $\phi$ we have that $-\phi'(y)(x - x^v) \leq \phi(x)$. Therefore, $f$ is a lower bound on $\phi$ for arbitrary $y < 0$ so we are done.

A similar argument can be made for $y \geq 0$, the signs will be flipped at each step. $\qquad\square$

A direction of potential interest for future developments is how to "absolutely-convexify" a given function. Below, we prove the propotypical case of functions from $\mathbb{R}$ to $\mathbb{R}$. In words, any convex function lifted high enough is absolutely convex.

**Lemma F.17.** *Suppose $f : \mathbb{R} \to \mathbb{R}$ is a convex function. Then $\phi(x) = f(x) + \beta$ is non-negative for $x \in [a, b]$ with $\alpha = max\{|f'(a)|, |f'(b)|\}$ and $\beta = \frac{\alpha(b-a)}{2} - \frac{f(a)+f(b)}{2}$.*

*Proof.* It is sufficient to show that $\phi(x) \geq 0$ for $x \in [a, b]$:

$$\begin{aligned}
\phi(x) &= f(x) + \frac{\alpha(b - a)}{2} - \frac{f(a) + f(b)}{2} \\
&= \frac{f(x) - f(a) + \alpha(x - a)}{2} + \frac{f(x) - f(b) - \alpha(x - b)}{2} \\
&\geq \frac{D_f(x, a)}{2} + \frac{D_f(x, b)}{2} \\
&\geq 0.
\end{aligned}$$

$\qquad\square$

**Lemma F.18.** *Suppose $f : \mathbb{R} \to \mathbb{R}$ is a convex function. Then for any interval $[a, b]$ there exists $\beta$, such that $\phi(x) = f(x) + \beta$ is absolutely convex on $[a, b]$.*

*Proof.* As $\phi$ is convex by convexity of $f$, it is sufficient to show that for every $x, y \in [a, b]$:

$$-\phi(x) \leq \phi(y) + \phi'(y)(x - y).$$

From Lemma F.17 we have $\phi(x) \geq 0$, so it is sufficient to consider the case when $\phi'(y)(x - y) \leq 0$.

Having $\alpha, \beta$ from Lemma F.17.

**Case 1:** $\phi'(y) \leq 0$ and $x \geq y$.

From the convexity of $\phi$ it follows that:

$$\phi(y) + \phi'(y)(x - y) - (\phi(a) + \phi'(a)(x - a)) = D_\phi(y, a) + (\phi'(y) - \phi'(a))(x - y)$$
$$\geq D_\phi(y, a)$$
$$\geq 0.$$

Hence, $-f'(a) \leq \alpha$ by construction, implying:

$$\phi(a) - \alpha(x - a) \leq \phi(y) + \phi'(y)(x - y).$$

It remains to show that:

$$\phi(x) + \phi(a) - \alpha(x - a) = f(x) + f(a) + 2\beta - \alpha(x - a)$$
$$= f(x) - f(b) + \alpha(b - a) - \alpha(x - a)$$
$$= f(x) - f(b) - \alpha(x - b)$$
$$\geq D_f(x, b)$$
$$\geq 0.$$

**Case 2:** $\phi'(y) \geq 0$ and $x \leq y$.

In analogy with the previous case:

$$\phi(b) + \alpha(x - b) \leq \phi(y) + \phi'(y)(x - y).$$

Therefore, it is sufficient to show that:

$$\phi(x) + \phi(b) + \alpha(x - b) = f(x) + f(b) + 2\beta + \alpha(x - b)$$
$$= f(x) - f(a) + \alpha(b - a) + \alpha(x - b)$$
$$= f(x) - f(a) + \alpha(x - a)$$
$$\geq D_f(x, a)$$
$$\geq 0.$$

$\square$

It is useful to remind the flowing standard result for sub-gradients. The proof is in the referenced book.

**Lemma F.19** (Lebourg Mean Value Theorem (Clarke, 1990))**.** *Suppose $\phi : \mathbb{R} \to \mathbb{R}$ is Lipschitz on any open set containing the line segment $[x, y]$. Then there exists an $a \in (x, y)$ such that*

$$\phi(x) - \phi(y) \in \langle \partial\phi(a), x - y \rangle.$$

**Lemma F.20.** *Suppose $\phi : \mathbb{R} \to \mathbb{R}$ is absolutely convex. Let $M \in \mathbb{R}$ be the bound on the subgradient of $\phi$, i.e. $|\phi'(x)| \leq M$. Fix a $y \in \mathbb{R}$. Then for any $x \geq y$ we have that*

$$\phi(y) + M(x - y) \geq \phi(x).$$

*Similarly, for any $x \leq y$,*

$$\phi(y) - M(x - y) \geq \phi(x).$$

*Proof.* We prove the first claim. Suppose $x \geq y$.

Since $\phi$ is convex, it is locally Lipschitz i.e. Lipschitz on $[y, x]$. By Lebourg's MVT we know there exists a $c \in (y, x)$ such that $\phi(x) - \phi(y) = g(x - y)$ where $g \in \partial\phi(c)$. Note that $g \leq M$. So, $g(x - y) \leq M(x - y)$ because $x - y \geq 0$. Therefore,

$$\phi(x) - \phi(y) \leq M(x - y).$$

Rearranging this expression we obtain $\phi(x) \leq \phi(y) + M(x_0 - y)$.

The proof of the second claim follows the same format and uses the fact that $g \geq -M$ instead. $\square$

**Lemma F.21.** *Suppose $\phi : \mathbb{R} \to \mathbb{R}$ is absolutely convex. Let $M \in \mathbb{R}$ be such that $|\phi'| \leq M$. Then the following limits exist and are equal*

$$\lim_{x \to \infty} |\phi'(x)| = \lim_{x \to -\infty} |\phi'(x)|.$$

*Proof.* First, we show that $\lim_{x \to \infty} |\phi'(x)|$ exists.

Let $\{x_n\}$ be an arbitrary sequence such that $x_n \to \infty$. Since $\phi$ is convex the sequence $\{|\phi'(x_n)|\}$ is monotonically increasing. Also, it is bounded above. Therefore, by the Monotone Convergence Theorem there exists an $L_1 \in \mathbb{R}$

$$\lim_{n \to \infty} |\phi'(x_n)| = L_1 \text{ and } L_1 \geq |\phi'|.$$

Since $x_n$ is arbitrary it must be that $\lim_{x \to \infty} |\phi'(x)| = L_1$. A similar argument can demonstrate that there exists an $L_2 \in \mathbb{R}$ with $|L_2| \geq |\phi'|$ and $\lim_{x \to -\infty} |\phi'(x)| = |L_2|$.

Now we show that $L_1 = |L_2|$. We will do this by contradiction. Suppose without loss of generality that $L_1 > |L_2|$ and that $L_2 < 0$.

There exists an $\epsilon > 0$ such that $L_1 - \epsilon > |L_2|$. Now there also exists an $N$ such that for any $y > N$ we have that $|\phi'(y) - L_1| < \epsilon$. So $L_1 - \epsilon < \phi'(y)$.

By absolutely convexity of $\phi$, function $f(x) = |\phi(y) + \phi'(y)(x - y)|$ is a lower bound on $\phi$. Let $x_v$ be the value where $f(x_v) = 0$.

Define $l(x) = -\phi'(y)(x - x_v)$, which is the line passing through the point $(x_v, 0)$ with slope $-\phi'(y)$. Observe that for $x \in (-\infty, x_v]$, $l(x) = f(x)$ so $l$ is a lower bound on $\phi$ in that interval. Define $h_1(x) = -(L_1 - \epsilon)(x - x_v)$ to be the line passing through $(x_v, 0)$ with slope $-(L_1 - \epsilon)$. For $x \leq x_v$, $h_1(x) < l(x)$ because $L_1 - \epsilon < \phi'(y)$. Therefore, $h_1(x) < \phi(x)$ for $x \in (-\infty, x_v]$.

Define $h_2(x) = \phi(x_v) + L_2(x - x_v)$. Since $L_2 < 0$, by Lemma F.20, the function $h_2$ is an upper bound on $\phi$ for any $x < x_v$.

By calculation we can determine an $x_i$ such that $h_1(x_i) = h_2(x_i)$ where

$$x_i = \frac{x_v(L_2 + L_1 - \epsilon) - \phi(x_v)}{L_2 + L_1 - \epsilon}.$$

Now we show that $x_i \leq x_v$. Note that $\phi$ is absolutely convex so $-\phi(x_0) \leq 0$. By adding $(L_2 + L_1 - \epsilon)x_v$ to both sides of this inequality we obtain

$$x_v(L_2 + L_1 - \epsilon) - \phi(x_0) \leq x_v(L_2 + L_1 - \epsilon).$$

Dividing by $L_2 + L_1 - \epsilon$ we get

$$x_i = \frac{x_v(L_2 + L_1 - \epsilon) - \phi(x_v)}{L_2 + L_1 - \epsilon} \leq x_v.$$

Let $x < x_i$. So $h_1(x) = h_1(x_i) - (L_1 - \epsilon)(x - x_i)$ and $h_2(x) = h_1(x_i) + L_2(x - x_i)$. Observe that $-(L_1 - \epsilon) < L_2$ because $L_2 < 0$ and $L_1 - \epsilon > |L_2|$. Multiplying both sides of this inequality by $x - x_i$ which is less than 0 and then adding $h(x_i)$ to both sides again we see that $h_1(x) > h_2(x)$.

Therefore, for any $x < x_i < x_v$, we have that $h_1(x) > h_2(x)$. This is a contradiction because on the interval $(-\infty, x_v)$, $h_1$ is a lower bound so $\phi(x) \geq h_1(x)$ and $h_2$ is an upper bound so $h_2(x) \geq \phi(x)$. □

**Lemma F.22.** *Suppose $\phi : \mathbb{R} \to \mathbb{R}$ is absolutely convex. Suppose it has a minimum point $x_\star$. Suppose there exists a $y_1, y_2 \in \mathbb{R}$ such that $y_1 < x_\star < y_2$ and for every $y \leq y_1$, $\phi'(y) = m_1$ and for every $y' \geq y_2$, $\phi'(y') = m_2$. Then $m_1 = m_2$.*

*Proof.* Note that since $\phi$ is convex it has monotonicly increasing derivative. Therefore, $m_1 < 0$ since $y < x_\star$ and $m_2 > 0$ since $y' > x_\star$. Let $f(x) = |\phi(y_1) + m_1(x - y_1)|$ be the tangent cone to

$y_1$. Then define line $h(x) = |m_1| \left(x + \frac{\phi(y_1)}{m_1} - y_1\right)$ which is a line that has the same slope as $f$ and intersects the vertex of $f$. Note that $h$ is a lower bound on $\phi$ by absolute convexity. $\qquad \square$

**Lemma F.23.** *Suppose $\phi : \mathbb{R} \to \mathbb{R}$ be absolutely convex and assume $\phi^* = \inf_{x \in \mathbb{R}} \phi(x) < \infty$. Then there exists an $x_\star \in \mathbb{R}$ such that $\phi(x_\star) \le \phi(x)$ for any $x \in \mathbb{R}$.*

*Proof.* Suppose $x \in \mathbb{R}$ is arbitrary. Choose a $y \in \mathbb{R}$ and by absolute convexity we have that $|\phi'(y) + \phi'(y)(x - y)| \le \phi(x)$. We can rearrange the terms on the left side and take the limit to see that

$$\lim_{|x| \to \infty} |\phi(y) - \phi'(y)y + \phi'(y)x| = \infty.$$

since only the last term which is linear depends on $x$. Therefore, we have that $\lim_{|x| \to \infty} \phi(x) = \infty$.

This demonstrates that $\phi$ is coercive. Now let $x_n$ be a sequence such that $\phi(x_n) \to f^*$. Suppose that $\lim_{n \to \infty} |x_n| = \infty$. Then by coercivity, we get that $\lim_{|x_n| \to \infty} \phi(x_n) = infty$ which is a contradiction with the fact that $\phi^* \le \infty$. Thus it must be that $\lim_{n \to \infty} |x_n| = r$ for some $r \in \mathbb{R}$. Let $B_r = \{x \in \mathbb{R} \mid |x| \le r\}$ which is compact because it is closed and bounded. Since $x_n \in B_r$ and every sequence in a compact set has a convergent subsequence, so there exists an $x_\star \in B_r$ s.t. $x_{n_k} \to x_\star$. By continuity of $\phi$ (because it is a convex function) we obtain $f^* = \lim_{k \to \infty} f(x_{n_k}) = f(x_\star)$. $\qquad \square$

### F.4 MULTIVARIABLE FUNCTIONS

Having analyzed the easy case, we move to general instances of absolutely convex functions. In particular, we prove that gradients of absolutely convex functions are bounded. The first statement is a rewriting of Lemma 7.1 in the main text.

**Lemma F.24.** *Suppose $\phi : \mathbb{R}^d \to \mathbb{R}$ is absolutely convex and has a minimizer $x_\star$. Then there exists a $M \in \mathbb{R}$ such that $\|\nabla\phi(x)\|_2 \le M$ for all $x \in \mathbb{R}^d$.*

*Proof.* If $\phi$ is the constant function then its gradient is bounded so we consider the case where $\phi$ is not constant.

We will do a proof by contradiction. Let $c > \phi(x_\star)$. Let $\epsilon = \frac{c - \phi(x_\star)}{2}$. Note that $|\phi(x_\star) - c| > \epsilon$. Suppose $\delta > 0$. Observe that since $\phi$ is convex on $\mathbb{R}^d$ it is continuous on $\mathbb{R}^d$ and in particular it is continuous at $x_\star$. Therefore, there exists a $\delta > 0$ such that if $|x_\star - x| \le \delta$ then $|\phi(x_\star) - \phi(x)| \le \epsilon$.

Suppose that $|\nabla\phi|$ is unbounded. So, there exists a sequence of points $y_n \in \mathbb{R}^d$ such that

$$\lim_{n \to \infty} \|\nabla\phi(x)\|_2 = \infty.$$

Let $f_n(x) = |\phi(y_n) + \langle \nabla\phi(y_n), x - y_n \rangle|$. Since $\phi$ is absolutely convex, $f_n$ must be a lower bound on $\phi$. We can proceed similarly to the proof of bounded gradients in $\mathbb{R}$ (Lemma F.15) by considering the restriction of $f$ and $\phi$ to specific lines. This allows us to find a sequence of points $x_n$ that lie on those lines and $x_n \to x_\star$.

Define $L_n$ to be the line that passes through $x_\star$ in the direction of $\nabla\phi(y_n)$. Let $\phi\big|_{L_n} : \mathbb{R} \to \mathbb{R}$ be the restriction of $\phi$ to the line $L_n$ and similarly, $f_n\big|_{L_n} : \mathbb{R} \to \mathbb{R}$ be the restriction of $f_n$ to $L_n$. Note that the function $f_n\big|_{L_n}$ is of the form $|a + b(x - x_0)|$ where $b = \|\nabla\phi(y_n)\|^2$ for some $a, x_0 \in \mathbb{R}$. Let $\bar{x}^* \in \mathbb{R}$ be the minimizer of $\phi\big|_{L_n}$. Then by Lemma, there exists a point, $\bar{x}_n \in \mathbb{R}$ such that $\phi\big|_{L_n}(\bar{x}_n) = c$ and $|\bar{x}^* - \bar{x}_n| \le \frac{2c}{\|\nabla\phi(y_n)\|^2}$. Observe that $\bar{x}^*$ corresponds to $x_\star$. Also, $\bar{x}_n$ can be mapped to a point $x_n \in \mathbb{R}^d$ which lies on the line $L_n$ and $\phi(x_n) = c$ and $|x_\star - x_n| \le \frac{2c}{\|\nabla\phi(y_n)\|^2}$. This holds for each $y_n$.

Since $\|\nabla\phi(y_n)\|_2$ is unbounded we can find an $N$ such that $|x_\star - x_N| < \delta$. Therefore, $|\phi(x_\star) - \phi(x_n)| = |\phi(x_\star) - c| \le \epsilon$. However, this contradicts the fact that $|\phi(x_\star) - c| > \epsilon$. $\qquad \square$

**Lemma F.25.** *The maximum of a constant and an absolutely convex function is absolutely convex.*

*Proof.* Let $\alpha \in \mathbb{R}$ and $f$ be absolutely convex and $g := \max\{f, \alpha\}$. We split the argument in some sub-steps.

**(Trivial case)** Since absolutely convex functions are always positive, it follows that if $\alpha \leq 0$ then $g(x) = \max\{f(x), \alpha\} \equiv f(x)$ and $f = g$ is absolutely convex.

**(Second case)** Let $\alpha > 0$. Since $f$ is absolutely convex, it is convex and $g$ is by construction. Therefore, the positive side of the inequality in absolute convexity needs not to be verified. It reamains to show that:

$$\text{wts} \quad g(y) \geq -g(x) - \langle \nabla g(y), x - y \rangle \quad \forall x, y \iff g(y) + g(x) + \langle \nabla g(y), x - y \rangle \geq 0 \quad \forall x, y. \tag{44}$$

For convenience, we will show the last version is positive for different choices of $x, y$. Recall that $f$ is always positive so for arbitrary $(x, y)$ there are four regions identified by the strips $[0, \alpha); [\alpha, \infty)$ iover which the values $f(x), f(y)$ can fall.

Additionally, recognize that for $z \in \mathbb{R}^d$ we have $\nabla g(z) = \nabla \max\{f(z), a\} = \mathbb{1}_{f(z) > a} \nabla f(z)$. Let us treat all the cases in separate ways.[4]

If $f(x) \leq \alpha$ and $f(y) < \alpha$ we have the expression $2\alpha \geq 0$ by construction.

If $f(x) > \alpha$ and $f(y) < \alpha$ we have the expression:

$$f(x) + \alpha > 0; \tag{45}$$

again, by construction.

If $f(x) < \alpha$ and $f(y) > \alpha$ we have $a + f(y) + \langle \nabla f(y), x - y \rangle > f(x) + f(y) + \langle \nabla f(y), x - y \rangle \geq 0$ since we assumed $f$ is absolutely convex.

If $f(x) \geq \alpha$ and $f(y) > \alpha$ one has:

$$f(x) + f(y) + \langle \nabla f(y), x - y \rangle \geq 0, \tag{46}$$

which follows by the assumed strong convexity of $f$. $\qquad\square$

remark Let $x_\star = \arg\min f$, for an absolutely convex function $f$. Observe that one can always use, for any $y \neq x_\star, x \neq x_\star$:

$$f(x) \geq |f(y) + \langle \nabla f(y), x_\star - y \rangle|. \tag{47}$$

**Lemma F.26.** *Suppose $v \in \mathbb{R}^d$. A function $\phi : \mathbb{R}^d \to \mathbb{R}$ is absolutely convex if and only if the function $f : \mathbb{R} \to \mathbb{R}$ defined as $f(t) = \phi(x + tv)$ is absolutely convex for all $x \in \mathbb{R}^d$.*

*Proof.*

($\Rightarrow$) Suppose $f(t)$ is an absolutely convex function for any $x, v \in \mathbb{R}^d$. We already know that $\phi$ will be convex so we only need to show that for all $y, z \in \mathbb{R}^d$,

$$-\phi(x) \leq \phi(y) + \langle \nabla \phi(y), z - y \rangle.$$

Note that $f'(t) = \langle \nabla \phi(x + tv), v \rangle$. Select $x = y$ and $v = z - y$. Since $f$ is absolutely convex, the following inequality will hold,

$$\begin{aligned} -f(1) &\leq f(0) + f'(0)(1 - 0) = f(0) + f'(0) \\ &= \phi(y) + \langle \nabla \phi(y), z - y \rangle. \end{aligned}$$

We have our result because $f(1) = \phi(y + v) = \phi(z)$

($\Leftarrow$) Suppose $\phi$ is absolutely convex. Let $x, v \in \mathbb{R}^d$ be arbitrary. We know already that $f$ is convex. So we just need to show that for any $s, t \in \mathbb{R}$ we have that

$$-f(t) \leq f(s) + f'(s)(t - s).$$

---

[4]In principle, at $z = a$ there is a singularity, we avoid doing this computation since the non-differentiable definition of absolute convexity is satisfied for the max function.

By absolute convexity of $\phi$ we know that,

$$
\begin{aligned}
-\phi(x + tv) &\leq \phi(x + sv) + \langle \nabla\phi(x + sv), x + tv - (x + sv) \rangle \\
&= \phi(x + sv) + \langle \nabla\phi(x + sv), tv - sv \rangle \\
&= \phi(x + sv) + \langle \nabla\phi(x + sv), v \rangle (t - s) \\
&= f(s) + f'(s)(t - s).
\end{aligned}
$$

Since $f(t) = \phi(x + tv)$ we have our result. $\qquad\square$

## G    EXTRA EXPERIMENTS

### G.1    REGRESSION WITH SQUARED HUBER LOSS

In this experiment we optimize the function

$$f(x) = \frac{1}{n} \sum_{i=1}^{n} h_\delta^2(a_i x - b_i),$$

where $a_i \in \mathbb{R}^d$, $b_i \in \mathbb{R}$ are the data samples associated with a regression problem, and $h_\delta$ is the Huber loss function. We run the experiments with $\mathbf{C}(x)$ for absolutely convex functions 7.2.

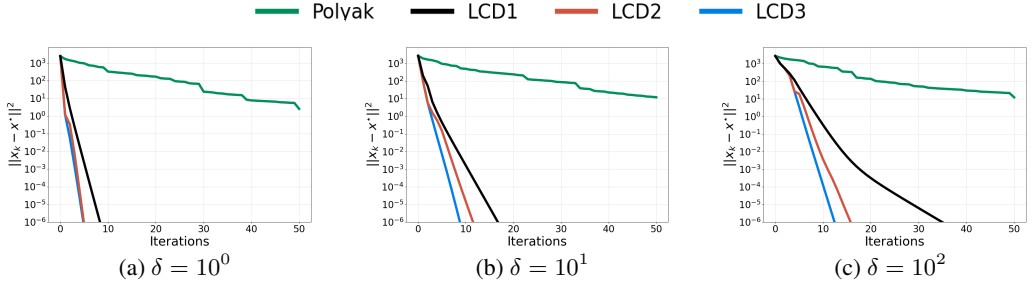

(a) $\delta = 10^0$    (b) $\delta = 10^1$    (c) $\delta = 10^2$

Figure 5: Regression on `housing` dataset.

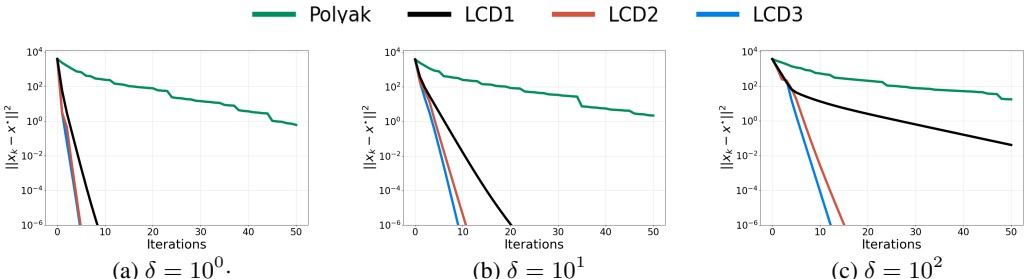

(a) $\delta = 10^0 \cdot$    (b) $\delta = 10^1$    (c) $\delta = 10^2$

Figure 6: Regression on `mpg` dataset.

Figures 5–6 show that the algorithms using the $\mathbf{C}(x)$ matrix perform much better than the Polyak method. We observe very fast convergence of both LCD3 and LCD2, regardless of $\delta$. Contrary, as $\delta$ increases LCD1 loses in comparison with the other two matrix methods. The most likely reason is increasing part of the objective, which is quartic, as it requires extra adaptiveness on the smoothness constant.

### G.2    RIDGE REGRESSION

We consider the following objective function:

$$f(x) = \frac{1}{n} \sum_{i=1}^{n} (a_i x - b_i)^2 + \lambda \|x\|^2$$

where $a_i \in \mathbb{R}^d$, $b_i \in \mathbb{R}$ are the data samples associated with a regression problem. By $L$ we understand the smoothness constant of a linear regression instance, excluding the regularizer.

In Figures 7–8, $\mathbf{C}(x)$ is associated with the regularizer, and it becomes a multiple of $\mathbf{I}$. As discussed in the main text, in this case LCD2 has closed-form solution, which coincides with LCD3. The LCD1 algorithm becomes GD. We can see, similar behavior to logistic regression with $L_2$ regularizer that is consistent improvement of LCD2 over the Polyak's method.

Figures 9–10 show the results with $\mathbf{C}(x) = \frac{2}{n}\mathbf{A}^\top\mathbf{A}$, which is a lower bound on the main part of the objective. In this circumstance, LCD1 becomes the Newton's method, and converges in one step. As anticipated in the main text, LCD3 can diverge. Finally, LCD2 performs in a very consistent way, and converges in exactly 15 steps across all the setups.

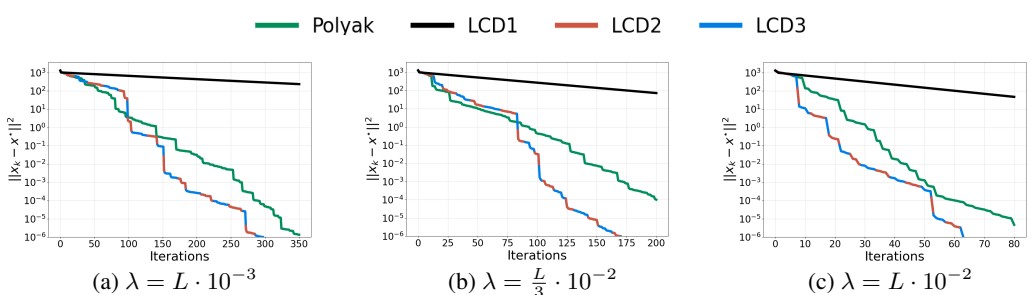

Figure 7: Ridge regression on `housing` dataset; $\mathbf{C}(x) = 2\lambda$.

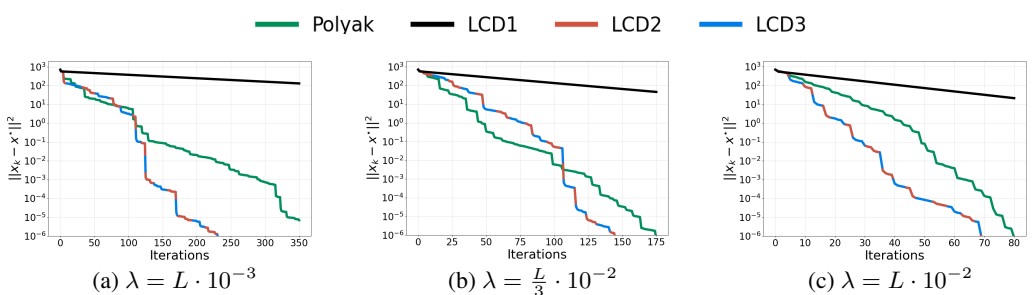

Figure 8: Ridge regression on `mpg` dataset; $\mathbf{C}(x) = 2\lambda$.

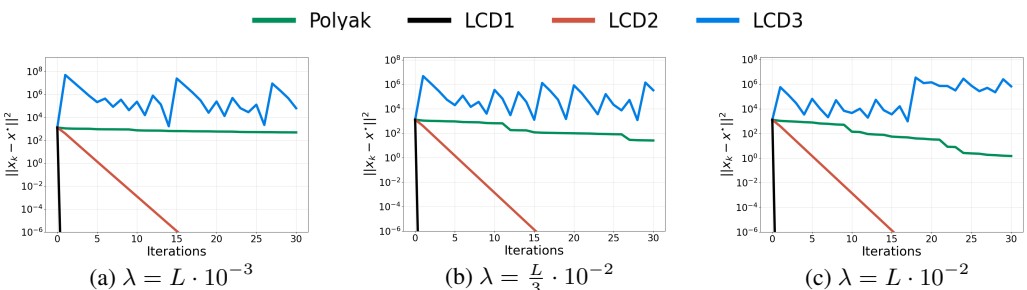

Figure 9: Ridge regression on `housing` dataset; $\mathbf{C}(x) = \frac{2}{n}\mathbf{A}^\top\mathbf{A}$.

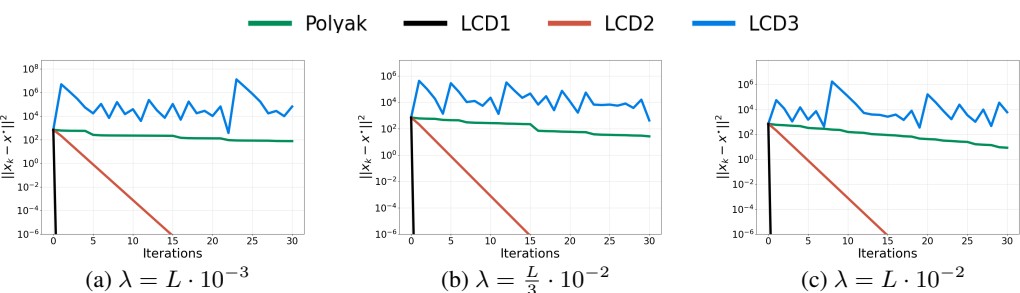

Figure 10: Ridge regression on `mpg` dataset; $\mathbf{C}(x) = \frac{2}{n}\mathbf{A}^\top\mathbf{A}$.

