# OpenReview forum: "Local Curvature Descent: Squeezing More Curvature out of Standard and Polyak Gradient Descent"
_ICLR.cc/2025/Conference — Submitted to ICLR 2025_

### Official Review · Reviewer_inYC · 2024-10-28

**Soundness:** 3
**Presentation:** 3
**Contribution:** 2
**Rating:** 5
**Confidence:** 4

**Summary:**

The paper explores methods for enhancing gradient descent through adaptive step sizes. The basic idea is to exploit "local curvature" without resorting to fully second-order methods, which require computationally expensive Hessians, i.e., using extra knowledge the idea is to interpolate between pure first order and second order. To this end, the authors introduce three algorithms or more precisely step-size estimation strategies: LCD1, LCD2, and LCD3. Each one optimizes step size based on curvature properties of the objective function.

The authors establish theoretical convergence rates for LCD1 and LCD2, aligning with those for standard and Polyak step size gradient descent methods, which they intend to extend. The authors also claim improved empirical performance.

**Strengths:**

The paper expands the class of objective functions for which we can obtain solid convergence guarantees. Moreover, the step size strategies (LCD 2 and LCD 3 which generalize Polyak steps) can provide some form of adaptivity. The computational results indicate that there might be empirical performance improvements as well.

**Weaknesses:**

I have some concern regarding the novelty of the proposed method/approaches. In particular, the proposed concept of local curvature and the analysis seems to be very similar to "relative smoothness" and the associated analysis, see e.g.,

https://pubsonline.informs.org/doi/10.1287/moor.2016.0817

and

https://arxiv.org/pdf/1911.08510

(just to name two examples - there is much more)

Given these similarities the authors should discuss and relate their work to that of relative smoothness and also clearly delineate their own contribution from what is already known or implied by relative smoothness. I would like to ask the authors to clarify this in the rebuttal:

1. How does the local curvature assumption differ from or extend relative smoothness?

2. Do the convergence guarantees for LCD1 and LCD2 improve upon what could be derived using relative smoothness?

3. Are there problem settings where the local curvature approach provides benefits over relative smoothness?


Independent of the above, the paper feels a little incremental, technical (the paper is very long), and niche in the sense that it provides "yet another" step size strategy for a broader class of objectives. The results are nice but not very surprising.

**Questions:**

see above -

---

> ### Author Response · Authors · 2024-12-03
>
> Dear Reviewer,
>
> Thank you for taking the time to review our work!
>
> > How does the local curvature assumption differ from or extend relative smoothness?
>
> Other works such as [1] propose the notion of relative smoothness based on the Bregmann divergence of some arbitrary strictly convex and continuously differentiable kernel function $h$. The motivation for our assumption is to leverage local curvature information to develop adaptive algorithms with matrix-valued step sizes without going the route of fully-fledged second-order methods. Hence, we focus on deriving and analyzing algorithms that utilize a local curvature matrix.
>
> Specifically, we believe that relative smoothness does not fully capture our assumption. The variable nature of the norm does not allow for the term $\frac{1}{2}||x-y||^2_{C(y) + LI}$ in our assumption to be expressed as the Bregman divergence of some kernel function. For example, taking the Euclidean kernel $h(x) = \frac{1}{2}||x||^2_2$ recovers the classical $L$-smoothness assumption,
>
> $$
> f(x) \leq f(x) + \left<\nabla f(y),x-y \right> + \frac{L}{2}||x-y||^2_2.
> $$
>
> Let's say we want the norm to be induced by a matrix that can vary with $y$ because our assumption is of the following form,
>
> $$
> f(x) \leq f(x) + \left<\nabla f(y),x-y \right> + \frac{1}{2}||x-y||^2_{C(y) + LI}.
> $$
>
> In this case, a simple kernel function is not immediate. If we try to use $h(x) = ||x||^2_{C(x)}$, the expression for $D_h(x,y)$ does not simplify to $\frac{1}{2}||x-y||^2_{C(y)}$. In fact, we need to differentiate through $C(x)$ to calculate the Bregman divergence so the expression for $D_h(x,y)$ quickly becomes much more complicated than the terms in our assumption.
>
> Furthermore, we would like to emphasize that Assumption 2.1 in our work introduces both an upper bound and lower bound on $f$. Of course, the relative smoothness assumption pertains to the upper bound on $f$.
>
> > Do the convergence guarantees for LCD1 and LCD2 improve upon what could be derived using relative smoothness?
>
> Our assumptions and algorithms are not covered by the relative smoothness framework due to the variable nature of the norm. Thus a direct comparison cannot be made. However, works such as [1] derive an $O(1/k)$ rate for the Bregman Gradient algorithm. The Bregman Gradient algorithm which is derived from the relative smoothness assumption is analogous to LCD1 which is derived from the upper bound in Assumption 2.1. LCD1 also achieves an $O(1/k)$ rate.
>
> > Are there problem settings where the local curvature approach provides benefits over relative smoothness?
>
> The analysis of LCD1 and LCD2 leads to better constants compared to their classical counterparts. Also, GD with Polyak step and GD with step size $\frac{1}{L}$ have the same theoretical convergence rate. However, there is a great benefit in using the Polyak step size because it converges much faster empirically. The same reasoning applies to LCD2, an extension of the Polyak step size. Although LCD2 has the same theoretical rate as GD (up to constants), by utilizing local curvature information and adaptive step size, LCD2 exhibits superior empirical performance compared to both Polyak step size and constant step GD.
>
> For machine learning problems, diagonal curvature matrices may be available as demonstrated by our examples. This results in computationally cheap update steps for our algorithms and improved empirical performance shown by our experiments. The performance of algorithms derived from relative smoothness would depend on the kernel function that is used in the algorithm. Certain kernel functions may result in updated steps that cannot be computed efficiently in practice.
>
> [1] Bregman First Order Methods: https://arxiv.org/pdf/1911.08510

---

### Official Review · Reviewer_NSv5 · 2024-11-01

**Soundness:** 3
**Presentation:** 3
**Contribution:** 1
**Rating:** 3
**Confidence:** 5

**Summary:**

This paper proposes three variants of descent methods for solving convex optimization problems that satisfy Assumptions 2.1, specifically (5) and (6). Conditions (5) and (6) define a class of convex problems with a curvature matrix C(y) and a smoothness parameter Lc​. These parameters, C(y) and Lc​, are used to compute the step sizes for the three methods. The first method generalizes gradient descent with a fixed step size; the second method generalizes gradient descent with the Polyak step size; and the third method is a heuristic variant of the second method. Both the first and second methods are proved to converge at a rate of O(1/k). Experimental results demonstrate the convergence of the methods, with Methods 2 and 3 notably outperforming the Polyak step-size approach, highlighting the benefits of incorporating curvature information in algorithm design. The paper also provides example problems for which the curvature matrix C(y) and Lc​ are computable.

**Strengths:**

- The methods illustrate how readily available curvature information for certain problems can be used to design algorithms that improve performance.
- Sections 6 and 7 provide problem examples and proofs, which help identify the class of problems for which these algorithms are applicable.

**Weaknesses:**

- The setting is very limited, namely, convex optimization problems for which the scaling matrix C(y) is known and available.
- It is claimed that Assumption 2.1 defines a new class of functions, but that is not true.  These ideas are all related to notions of generalized convexity and generalized distance functions.  In this respect, the extensions of the analyses of gradient descent, etc. are relatively straightforward.

**Questions:**

Prior to equation (22), what is Step 3 of LCD2?  There does not appear to be a "step 3"?  This seems to follow easily from the projection theorem onto closed and convex sets.

---

> ### Author Response · Authors · 2024-12-03
>
> Dear Reviewer,
>
> Thank you for the time spent analyzing the content of our work!
>
> > The setting is very limited, namely, convex optimization problems for which the scaling matrix C(y) is known and available.
>
> We understand the reviewer's concern but politely disagree. The new assumptions are weaker than well-studied strong convexity and smoothness. Also, we show a few interesting examples satisfying our assumptions, which are not strongly convex or smooth. Interestingly, our assumptions recover convexity and smoothness, another important set-up in optimization, as a __special case__.
>
> > It is claimed that Assumption 2.1 defines a new class of functions, but that is not true. These ideas are all related to notions of generalized convexity and generalized distance functions.
>
> Other works such as [1-3] that have examined assumptions like relative convexity and relative smoothness have limitations that are addressed in our work. For example, [1] only considers fixed positive definite matrices $M$ and $G$ to define a smoothness and convexity assumption:
>
> For all $x, y \in \mathbb{R}^d$,
>
> $
> \begin{equation}
> f(x) \leq f(x) + \left<\nabla f(y),x-y \right> + \frac{1}{2}||x-y||_{M}
> \end{equation}
> $
>
> $
> f(x+h) \geq f(x) + \left<\nabla f(y),x-y \right> + \frac{1}{2}||x-y||_{G}
> $
>
> In our assumption, the curvature matrix is allowed to vary with $y$ which makes it much more flexible. A simple example of a function that satisfies  Assumption 2.1 but not this assumption is the square of the Huber loss.
>
> Our assumption is more general than the relative smoothness/convexity assumption proposed in [2]. They only consider Hessian matrices $H(x) = \nabla^2 f(x)$.  Our goal is to develop adaptive matrix-valued step sizes without going the route of fully-fledged second-order methods. Assumption 2.1 does not restrict that the matrix must be the Hessian. This gives us the flexibility to use some local curvature information when it is available to design powerful adaptive step sizes.
>
> In response to Reviewer inYC, we address the relationship between Assumption 2.1 and relative smoothness using Bregmann divergence in detail. To summarize, the variable nature of the norm in our assumption does not allow it to be expressed using the Bregmann divergence of some kernel function.
>
> If we restricted to fixed matrices $M$ and $G$ then we can use $h_1(x) = \frac{1}{2}||x||^2_M$ and $h_2(x) = \frac{1}{2}||x||_{G}^2$ as kernel functions to derive the assumption in [1]. Of course, as we mentioned this assumption is more restrictive than Assumption 2.1. The variable nature of the matrix norm captures local curvature information which is crucial to our algorithms and separates our assumption from relative smoothness/convexity.
>
> > Prior to equation (22), what is Step 3 of LCD2? There does not appear to be a "step 3"? This seems to follow easily from the projection theorem onto closed and convex sets.
>
> Apologies for the confusion. This does follow from the projection theorem. The "Step 3" part is a typo. It is meant to refer to the projection onto the localization set. We have fixed this in an updated version.
>
> [1] SDNA: https://arxiv.org/pdf/1502.02268
>
> [2] Randomized Sub-space Newton: https://arxiv.org/pdf/1905.10874
>
> [3] Bregman First Order Methods: https://arxiv.org/pdf/1911.08510

---

### Official Review · Reviewer_hk65 · 2024-11-02

**Soundness:** 2
**Presentation:** 2
**Contribution:** 1
**Rating:** 3
**Confidence:** 4

**Summary:**

Convexity and L-smoothness are standard assumptions in optimization literature which are useful for easier analysis of optimization algorithms and determining the right algorithm parameters. These global conditions may not always take local differences in the curvature into account. This paper proposes new analogues of the assumptions to incorporate certain kinds of local curvature information. The paper also proposes modifications of gradient descent using matrix valued step sizes to take advantage of the modified assumptions.

**Strengths:**

The paper is clearly written and is easy to read. It is interesting to see that Newton's method and gradient descent can be seen as special cases of the same framework under the assumption proposed by the authors.

**Weaknesses:**

The first-order convexity condition and L-smoothness imply that $$ f(y) + \langle \nabla f(y), x-y\rangle \leq f(x) \leq f(y) + \langle \nabla f(y), x-y\rangle  + \frac L2 || x-y||^2.$$ The authors' assumption 2.1 modifies these inequalities by adding the term $\frac12 || x-y||^2_{\mathbf C}$ to the lower bound and upper bound for $f(x)$ provided by these inequalities (since $\frac12 || x-y||^2_{\mathbf C+L\mathbf I} = \frac12 || x-y||^2_{\mathbf C} +  \frac L2 || x-y||^2$). Thus, assumption 2.1 seems to be more general than L-smoothness but it is less general than convexity.

In the standard analyses of GD, L-smoothness guarantees a sufficient decrease with each step (with the right step size) and convexity ensures that that decrease pushes us towards the minimizer. The two inequalities balance each other in a crucial way. This work exploits that tradeoff. However, it is not surprising that if the same term is added to both the lower bound and upper bound of $f(x)$ then they will cancel each other out and the standard convergence proofs will still go through.

Furthermore, if $f$ is twice differentiable, assumption 2.1 is actually equivalent to
$$\mathbf C(x) \preceq \nabla^2 f(x) \preceq \mathbf C(x) + L_C \mathbf I.$$ This follows from the same kind of standard arguments used to show that convex functions have positive semidefinite Hessians. Having observed this second order condition, many of the remarks that the authors make follow directly. The case when $\mathbf C = 0$ is the standard case with convexity and L-smoothness, and the case when $L_C=0$ is the realm of second order methods like Newton's method. My impression is that to give any useful advantage over standard GD, the map $\mathbf C(x)$ will have to approximate the Hessian $\nabla^2 f(x)$. But then the algorithms provided do not seem to be very useful unless there is a good way to approximate the Hessian.

LCD1 actually just seems to be a version of Newton's method where the Hessian is overestimated by its upper bound $\mathbf C(x) + L_C \mathbf I$ to make it more stable. The convergence rate provided for LCD1 is the same as GD (unless $L_C = 0$, in which case it is pure Newton's method anyway), which does not provide any new insights either. The authors present LCD2 as a generalization of Polyak's step size, but the step size $\beta_k$ in that case does not even have a closed form. Computing $\beta_k$ itself requires an optimization problem to be solved at each step, and the benefits of doing that are not clear. LCD3 has a closed form step-size but there are no convergence results provided for it, so it's not clear how well it performs.

The assumption could still have been justified with examples of interesting functions that satisfy assumption 2.1 in non-trivial ways. Unfortunately, that does not seem to be the case. One of the curvature matrices specified for each of the examples 6.1-6.4 is just the Hessian. For examples 6.2 and 6.4, $\nabla f(x) \nabla f(x)^\top$ is proposed as another candidate for the curvature matrix, but these kinds of approximations of the Hessian are already covered by quasi-Newton methods like Berndt–Hall–Hall–Hausman algorithm.

The experiments are also only performed on these trivial examples and authors compare their proposed step sizes only against Polyak step size. The first experiment is on a strongly convex and L-smooth function, which is covered by the classical assumptions, and the optimal method for which would have been a momentum-based algorithm like Nesterov's accelerated gradient descent. The second experiment chooses the Hessian as the curvature matrix, reducing it to the case where second order methods would perform better.

Overall, the assumption proposed in the paper does to offer many new theoretical insights nor do the algorithms proposed offer practical advantages over existing algorithms.

**Questions:**

- Can the authors comment on the second order characterization of assumption 2.1?
- What is the time complexity of LCD2 compared to Polyak step size? Specifically, how does the computation of $\beta_k$ affect the complexity?

---

> ### Author Response · Authors · 2024-12-03
>
> Dear Reviewer,
>
> Thank you for taking the time to engage with out work!
>
> > The assumption could still have been justified with examples of interesting functions that satisfy assumption 2.1 in non-trivial ways. Unfortunately, that does not seem to be the case
>
> Our class of functions includes any strongly-convex function (a common assumption in optimization) as a special case when $\mathbf{C}(x) = \mu \mathbf{I}$. We provide additional examples of functions that are not strongly-convex but satisfy our assumption with a non-trivial $\mathbf{C}(x)$ curvature matrix including the square of absolutely convex functions. Therefore, many functions relevant to optimization are included in this new class.
>
> None of our examples have a curvature matrix that is equal to the Hessian (other than convex quadratics). Of course, the curvature matrices are related to the Hessian but that is not surprising since they represent some local curvature information and $C(x) \preceq \nabla^2f(x)$.
>
> > The experiments are also only performed on these trivial examples and authors compare their proposed step sizes only against Polyak step size.
>
> Experiments are tailored to the assumptions we consider. Our goal is to build a deep understanding of this setup before entering stochastic or non-convex regions
>
> > Can the authors comment on the second order characterization of assumption 2.1?
>
> We would like to clarify that Assumption 2.1 is not equivalent to
>
> $
> \begin{equation}
> C(x) \preceq \nabla^2f(x) \preceq  C(x) + LI.
> \end{equation}
> $
>
> Only one direction is true. If Assumption 2.1 is satisfied then this characterization is true. The other direction does not work. For example, of course, $\nabla^2 f(x) \preceq \nabla^2f(x) \preceq  \nabla^2 f(x)  + LI$. But setting $C = \nabla^2 f(x)$ does not work for many examples. For instance, consider the square of the Huber loss with the curvature matrix as the Hessian. Then the lower bound fails for $x = 0$ and $y = 0.5$.
>
> > What is the time complexity of LCD2 compared to Polyak step size? Specifically, how does the computation of affect the complexity?
>
> We can use Newton's root-finding method to compute $\beta_k$. The main cost is computing the eigendecomposition of $C(x_k)$ at the beginning of each iteration of LCD2. This has complexity $O(n^3)$ but in practice it is faster than computing the inverse. Of course, when $C(x_k)$ is a diagonal matrix we don't need to compute the eigendecomposition. We can take a few steps of Newton's method for a good approximation of $B_k$. Empirically, we observed that LCD2 converged faster than Polyak even for wall-clock time.

---

### Official Review · Reviewer_cRYj · 2024-11-03

**Soundness:** 3
**Presentation:** 4
**Contribution:** 3
**Rating:** 6
**Confidence:** 3

**Summary:**

- The paper introduces a new class of functions called convex and smooth with local structure. Many interesting examples are provided that satisfy the proposed assumption, and the calculus for the proposed class is presented. For this class, the authors proposed three new algorithms LCD1, LCD2, and LCD3. Theorem 4.1, 4.2 shows $\frac{L_C ||x_0 - x_*||^2}{2k}$ for LCD1 which is a generalization of gradient descent, and LCD2 which is a generalization of Polyak stepsizes. It is a good paper, it is well-written, easy to follow, and motivated by examples.

**Strengths:**

1. The proof of Theorems 4.1 and 4.2 is correct.
2.  A wide range of examples is provided alongside a general framework for constructing convex and smooth functions with local curvatures.
3. The theoretical results are motivated by examples and supported by experiments.

**Weaknesses:**

1. All three methods require a knowledge of $C(x)$. In the examples, such $C(x)$ is provided; however, if one wants to run these methods, one must find such $C(x)$, which can be a non-trivial task. Meanwhile, if $C(x)$ is not known in advance, one can still use GD  with line-search or Polyak stepsizes, or normalized gradient descent. Meaning, to use LCD1, LCD2, and LCD3, the knowledge of $C(x)$ is necessary.

**Questions:**

1. Is there any relation between convex and smooth local curvature functions with functions defined by Bregman divergence? If yes, is there a connection between Mirror Descent and LCD1?
2. GD with Polyak stepsizes also converge for the problem (1) under the proposed assumption, as well as the normalized gradient method, however, with a slower convergence rate. It would be beneficial to add a comparison of existing methods not only in experiments but also in the main section.
3. How the choice of $C(x)$ in experiments affects the performancece if the methods?
4. In the first experiment with $L_2$ regularization, GD also can be used; it would be interesting to see if LCD1 outperforms GD.

Typos.
- should it be $\mathbb{R}^d \rightarrow \mathbb{R}^d$ ?
- there is a problem with reference on lines 550-552
- in the proof of Lemma B.1, the last two lines of the first equation are repeated
- lines 756-757 ...the term $||x_k - x_*||^2_{C(x_k)}$ ... "positive" is missing.
- lines 1026-1027 remove "any".

---

> ### Author Response · Authors · 2024-12-03
>
> Dear Reviewer,
>
> Thank you for taking the time to review our work!
>
> > Is there any relation between convex and smooth local curvature functions with functions defined by Bregman divergence? If yes, is there a connection between Mirror Descent and LCD1?
>
> In response to Reviewer inYC, we address the relationship between Assumption 2.1 and relative smoothness using Bregman divergence in detail. To summarize, the variable nature of the norm in our assumption does not allow it to be expressed using the Bregman divergence of some kernel function.
>
> >GD with Polyak stepsizes also converge for the problem (1) under the proposed assumption, as well as the normalized gradient method, however, with a slower convergence rate. It would be beneficial to add a comparison of existing methods not only in experiments but also in the main section.
>
> Our results include that GD with Polyak step size converges at an $O(1/k)$ rate which is the same rate obtained by LCD1 and LCD2. Empirically, we found that LCD2 and LCD3 have superior performance compared to GD with Polyak step size.
>
> > How the choice of C(x) in experiments affect the performance if the methods?
>
> Intuitively, we found that when $C(x)$ "represents" more information about the optimization problem, the better the LCD methods perform. For example, in experiments where the $C(x)$ matrix is derived from the regularization term, LCD2 performs better than Polyak as the regularization weight $\lambda$ increases. Of course, we found that using the $C(x)$ which results in a tighter bound performs better.
>
> > In the first experiment with regularization, GD also can be used; it would be interesting to see if LCD1 outperforms GD.
>
> In the plots for this experiment, we did not include GD because it performed worse than LCD1 and made no progress.
>
> Thank you for finding the typos. We have fixed those in an updated version.

---

### Author Response · Authors · 2024-12-03

Dear Reviewers,

We would like to thank you for your feedback and comments on our paper. We deeply appreciate the time and effort you spent reviewing it. To improve the paper, we present the main ideas of our rebuttal below.

General Comments
The reviewers highlighted the following strengths of the paper:

* The proposed class generalizes the classical class of convex smooth functions;
* We show that our assumptions differ from relative smoothness/convexity
* The proposed methods achieve strong performance on problems with special structures that satisfy the new assumption;
* The theoretical foundation includes convergence proofs for LCD1 and LCD2;
the empirical evaluation of standard machine learning tasks is thorough;
* The paper develops LDC3, a practical preconditioned Polyak stepsize method.

Furthermore, we address reviewers' concerns about the practical application of our class, emphasizing its connection to smoothness and (strong) convexity, the most studied set-ups in mathematical optimization


We believe that this rebuttal period has been useful to strengthen the quality and understanding of our work. We hope that the reviewers find our responses satisfactory for further engagement.

Thank you for your time.

Sincerely,

The author(s)

---

### Meta-Review · Area_Chair_DiNp · 2024-12-19

**Metareview:**

The paper studies settings of smooth convex optimization where additional information about "local curvature" of the objective is readily available. The paper proposes three variations of gradient descent -- generalizing GD with fixed step size and with Polyak step size, and a variable-metric heuristic based on Polyak step size. For the first two step size strategies, the paper proves $1/k$ convergence rate. The paper further provides examples to justify the availability of local curvature information and experiments to demonstrate the effectiveness of the proposed strategies. While the paper is of interest to ICLR and more broadly the optimization community, the contributions felt somewhat specialized and possibly incremental. Thus, at this point, the paper was judged to be below the acceptance threshold.

**Additional Comments On Reviewer Discussion:**

The reviews all had similar concerns, which can be grouped into: (1) the strong assumptions about the knowledge of local curvature, (2) the need for solving another optimization problem to determine the step size in one of the strategies, and (3) the niche quality of the presented results. The authors did try to address these concerns to the extent possible; however, it seems like the reviews did not miss anything crucial.

---

### Decision · Program_Chairs · 2025-01-22

Reject